# Predicting the morphology of ice particles in deep convection using the super-droplet method: development and evaluation of SCALE-SDM 0.2.5-2.2.0, -2.2.1, and -2.2.2

Shin-ichiro Shima[1,2], Yousuke Sato[3,2], Akihiro Hashimoto[4], and Ryohei Misumi[5]

[1]Graduate School of Simulation Studies, University of Hyogo, Kobe, Japan
[2]RIKEN Center for Computational Science, Kobe, Japan
[3]Faculty of Science, Hokkaido University, Sapporo, Japan
[4]Meteorological Research Institute, Japan Meteorological Agency, Tsukuba, Japan
[5]National Research Institute for Earth Science and Disaster Resilience, Tsukuba, Japan

**Correspondence:** Shin-ichiro Shima (s_shima@sim.u-hyogo.ac.jp)

**Abstract.** The super-droplet method (SDM) is a particle-based numerical scheme that enables accurate cloud microphysics simulation with lower computational demand than multi-dimensional bin schemes. Using SDM, a detailed numerical model of mixed-phase clouds is developed in which ice morphologies are explicitly predicted without assuming ice categories or mass-dimension relationships. Ice particles are approximated using porous spheroids. The elementary cloud microphysics processes considered are advection and sedimentation; immersion/condensation and homogeneous freezing; melting; condensation and evaporation including cloud condensation nuclei activation and deactivation; deposition and sublimation; and coalescence, riming, and aggregation. To evaluate the model's performance, a 2D large-eddy simulation of a cumulonimbus was conducted, and the life cycle of a cumulonimbus typically observed in nature was successfully reproduced. The mass-dimension and velocity-dimension relationships the model predicted show a reasonable agreement with existing formulas. Numerical convergence is achieved at a super-particle number concentration as low as $128/\mathrm{cell}$, which consumes 30 times more computational time than a two-moment bulk model. Although the model still has room for improvement, these results strongly support the efficacy of the particle-based modeling methodology to simulate mixed-phase clouds.

## 1 Introduction

Mixed-phase clouds, which are clouds comprising droplets and ice particles, appear under multiple atmospheric conditions, from the tropics to the poles, and throughout the year (Shupe et al., 2008). Accurately simulating the evolution of droplets and ice particles in mixed-phase clouds is crucial to understanding cloud dynamics, precipitation formation, water transport, radiative properties, aerosol-cloud interaction, cloud electrification, and lightning. These features are all crucial to many environmental and societal issues, such as climate change and variability, numerical weather prediction, weather modification, and icing on infrastructure (e.g., wind turbines and power lines) and aircraft (e.g., Korolev et al., 2017).

Through its 70-year history, numerical models of cloud microphysics became increasingly sophisticated (e.g., Khain et al., 2015; Khain and Pinsky, 2018; Grabowski et al., 2019; Morrison et al., 2020). However, recent model inter-comparison studies

revealed that the models do not show any sign of converging toward the truth. Even the most sophisticated models do not correspond well, and the divergence in model results is as large in sophisticated models as it is in simple models (VanZanten et al., 2011; Xue et al., 2017). Mixed-phase cloud microphysics modeling is particularly challenging because we still lack a sufficient scientific understanding of mixed-phase cloud microphysics, and an algorithm appropriate for mixed-phase cloud microphysics does not exist. This study aims to address the second problem.

Every numerical model is an approximation of a phenomenon's mathematical model, which is a theoretical description that should express the system's behavior accurately. We apply a numerical scheme to construct a numerical model, which we use to produce an approximate solution of the phenomenon's underlying mathematical model for given spatiotemporal boundary conditions. This general philosophy of simulation is well documented, e.g., in Stevens and Lenschow (2001).

There are several types of cloud microphysics numerical models that are based on different levels of theoretical descriptions.

The first of these is the bulk model, which is the most widely used cloud microphysics model type (see, e.g., Khain et al., 2015; Morrison and Milbrandt, 2015; Khain and Pinsky, 2018; Grabowski et al., 2019; Morrison et al., 2020, for a review). Bulk models consider only the particle population's statistical features and are thus based on macroscopic descriptions of cloud microphysics. They solve a mathematical model that is closed in the lower moments of the distribution function of cloud droplets, rain droplets, and ice particle categories (e.g., mass and number mixing ratios). The basic premise of bulk models is that the distribution function can be determined by the lower moments, but such a universal relationship is unknown. In other words, in bulk models, to predict the time evolution of a chosen set of moments, their time derivatives are approximated by some functions of the moments being predicted, but this is not generally possible (see, e.g., Beheng, 2010). It would be also informative to note the analogy and difference between the Navier–Stokes equation and bulk models (Morrison et al., 2020), which highlights the difficulty in deriving bulk models. Therefore, for cloud microphysics, a more bottom-up approach to construct more accurate and reliable numerical models would be desired.

Kinetic description provides a more detailed microscopic mathematical model of cloud microphysics, with the evolution and motion of individual aerosol, cloud, and precipitation particles being explicitly considered. Assuming that particles are locally well-mixed, particle collisions are regarded as a stochastic process. Each particle is characterized by its position and internal state, the latter of which is specified by variables known as attributes, such as size, mass, ratio of the ice crystal's minor axis to the major axis (hereafter called "aspect ratio"), velocity, and chemical composition.

Mixed-phase cloud microphysics are far more complicated than those of liquid-phase clouds, with various ice crystal formation mechanisms, diffusional growth by deposition/sublimation, diverse ice particle morphologies, ice melting and shedding, riming and wet growth, aggregation, spontaneous/collisional breakup of ice particles, and rime splintering at play (e.g., Pruppacher and Klett, 1997; Hashino and Tripoli, 2007, 2008, 2011a, b; Khvorostyanov and Curry, 2014; Khain and Pinsky, 2018). Although our scientific understanding is not yet sufficient, it is plausible that mixed-phase cloud microphysics could be accurately described under a kinetic description framework. Indeed, direct comparison with laboratory data suggests that a kinetic description could express ice particle morphology evolution accurately (Jensen and Harrington, 2015). This is crucial because ice particle morphology significantly influences the fall speed, growth by diffusion and collision, and radiative properties of ice

particles. Because of their direct correspondence to elementary processes, it should also be easier to refine kinetic descriptions using laboratory measurements.

Two numerical scheme types exist for kinetic descriptions, namely bin schemes and particle-based schemes.

The development of bin schemes started independently of bulk models in the 1950s (e.g., Mason and Ramanadham, 1954; Hardy, 1963; Srivastava, 1967). For a review, see, e.g., Khain et al. (2015), Khain and Pinsky (2018), Grabowski et al. (2019), and Morrison et al. (2020).

Particle-based cloud microphysics modeling is a new approach that has emerged since the mid-2000s (e.g., Paoli et al., 2004; Jensen and Pfister, 2004; Shirgaonkar and Lele, 2006; Andrejczuk et al., 2008, 2010; Shima et al., 2009; Sölch and Kärcher, 2010; Riechelmann et al., 2012; Brdar and Seifert, 2018; Seifert et al., 2019; Jaruga and Pawlowska, 2018; Grabowski and Abade, 2017; Abade et al., 2018; Grabowski et al., 2018; Hoffmann et al., 2019). During particle-based modeling's early development, calculating the coalescence process was a numerical challenge. Shima et al. (2009), Andrejczuk et al. (2010), Sölch and Kärcher (2010), and Riechelmann et al. (2012) proposed different algorithms, and among those four schemes, the super-droplet method (SDM) developed by Shima et al. (2009) provides a computationally efficient Monte Carlo algorithm (Unterstrasser et al., 2017; Dziekan and Pawlowska, 2017). Several other coalescence algorithms were proposed in different research areas such as the weighted flow algorithm for aerosol dynamics (DeVille et al., 2011); O'Rourke's method (1981), and the no-time counter method (Schmidt and Rutland, 2000) for spray combustion; and Ormel and Spaans's method (2008) and Johansen et al.'s method (2012) for astrophysics. Li et al. (2017) confirmed that the performance of SDM is better than Johansen et al.'s method (2012), but direct comparison with other algorithms remains to be assessed.

The essential difference between bin schemes and particle-based schemes lies in the representation of particles. Bin schemes adopt an Eulerian approach and the particle distribution function is approximated using a finite number of control volumes (histogram). The time evolution is solved using a finite volume method or a finite difference method. In contrast, particle-based schemes rely on a Lagrangian approach and the population of real particles is approximated by using a population of weighted samples, sometimes referred to as super-droplets or super-particles. As discussed in Grabowski et al. (2019), bin schemes face problems that are challenging to overcome such as numerical diffusion, computational cost, and the breakdown of the Smoluchowski equation (Smoluchowski, 1916; Alfonso and Raga, 2017; Dziekan and Pawlowska, 2017). However, SDM could resolve, or at least mitigate, those problems.

Therefore, SDM and similar particle-based schemes should be more suitable for mixed-phase cloud microphysics simulations than bin schemes. Mainly because of computational costs, it is practically impossible to apply bin schemes to the most comprehensive form of kinetic description, which inevitably involves multiple attributes to express each particle's internal state. Instead, many existing bin models solve a simplified kinetic description that uses particle distribution functions with a one-dimensional attribute space approximation. For example, most rely on artificially separated categories of ice particles, with predefined mass-dimension and area-dimension relationships in each category. Another approach is adopted in the SHIPS model developed by Hashino and Tripoli (2007, 2008, 2011a, b), which is a bin model that solves sophisticated and comprehensive kinetic descriptions and does not use ice categories or mass-dimension relationships. However, to justify using the one-dimensional particle distribution function, they rely on the "implicit mass sorting assumption", stating that different solid

hydrometeor species do not belong to the same bin because they are naturally sorted by mass. Such simplifications could be a significant source of errors. SDM and similar particle-based schemes could directly simulate comprehensive kinetic descriptions with lower computational demand.

This study's primary objective is to assess particle-based modeling methodology's capability to simulate mixed-phase clouds. Therefore, we develop and evaluate the performance of a detailed numerical mixed-phase cloud model using SDM, wherein ice particle morphologies are explicitly predicted.

We first construct a mixed-phase cloud microphysics mathematical model, which is based on kinetic description. The fluid dynamics of moist air is described by the compressible Navier–Stokes equation, and aerosol, cloud, and precipitation particles are represented by point particles. Following Chen and Lamb (1994a, b) and Misumi et al. (2010), ice particles are approximated using porous spheroids. The elementary cloud microphysics processes considered in the model are advection and sedimentation; immersion/condensation and homogeneous freezing; melting; condensation and evaporation including the cloud condensation nuclei (CCN) activation and deactivation; deposition and sublimation; and coalescence, riming, and aggregation. We base the mathematical models used for those elementary processes on revised versions of existing formulas. Additionally, our model does not rely on ice categories or predefined mass-dimension relationships. For simplicity and owing to the lack of appropriate algorithms, we do not consider spontaneous/collisional breakup or rime splintering. We then develop a numerical model called SCALE-SDM to solve the mathematical model. Mixed-phase cloud microphysics is solved using the SDM. The fluid dynamics of moist air is solved by adopting a forward temporal integration scheme to both horizontal and vertical directions using a finite volume method with an Arakawa-C staggered grid. To evaluate our model's performance, we conduct a two-dimensional (2D) simulation of an isolated cumulonimbus, and find that our model well reproduces the life cycle of a cumulonimbus typically observed in nature. The mass-dimension and velocity-dimension relationships our model predicts show a reasonable agreement with existing formulas based on laboratory measurements and field observations. We also investigate the simulation's numerical convergence and confirm that our model can produce an accurate approximate solution with lower computational demand than multi-dimensional bin schemes. We then explore the possibility of further refining and sophisticating the model; however, advancing our understanding of mixed-phase cloud microphysics is beyond the scope of this study.

Several previous works are closely relevant to this study. Chen and Lamb (1994a, b) developed a detailed multi-dimensional bin model, which Misumi et al. (2010) extended and added ice volume as a new particle attribute. We follow that strategy and approximate ice particles as porous spheroids; however, their kinetic description is more detailed than ours because they also considered spontaneous/collisional breakup, shedding, rime splintering, and surface chemical reactions. They solved the model using a multi-dimensional bin scheme; hence, their numerical model carries a high computational cost. Hashino and Tripoli (2007, 2008, 2011a, b) further extended Chen and Lamb (1994a, b)'s kinetic description to account for polycrystals that can form below $-20\,°C$. They solve the mathematical model using a one-dimensional bin scheme; however, careful validation is needed to justify their implicit mass sorting assumption. Paoli et al. (2004), Jensen and Pfister (2004), and Shirgaonkar and Lele (2006) separately developed a particle-based model for ice-phase clouds, but neither the evolution of ice particle morphologies nor the aggregation of ice particles were considered in their models. Sölch and Kärcher (2010) also developed a particle-based

model for ice-phase clouds, but that model relies on ice categories and mass-dimension relationships. Brdar and Seifert (2018) developed McSnow, the first particle-based model for mixed-phase clouds. McSnow is a multidimensional expansion of the P3 bulk model (Morrison and Milbrandt, 2015; Milbrandt and Morrison, 2016), and thus free from ice categories; however, it still relies on mass-dimension relationships. Further, a kinetic approach is applied to ice particles, but not to droplets or aerosol
particles.

In this study we demonstrate that a large-eddy simulation of a cumulonimbus that predicts ice particle morphologies without assuming ice categories or mass-dimension relationships is possible if we use SDM.

The organization of the remainder of this paper is as follows. In Secs. 2–4, our mixed-phase cloud mathematical model is described in detail. The cloud microphysics model is based on kinetic description and is coupled with moist air fluid dy-
namics. Note that this model is an expansion of Shima et al. (2009)'s warm cloud model. In Sec. 5, we develop a numerical model called SCALE-SDM by applying SDM. To evaluate SCALE-SDM's performance, we conduct a 2D simulation of an isolated cumulonimbus. Sec. 6 presents the design of the numerical experiments, and in Sec. 7, the overall properties of the simulated cumulonimbus and ice particle morphologies are analyzed. The numerical convergence characteristics of the model are investigated in Sec. 8. In Sec. 9, possible improvements of the model are discussed, and a summary and conclusions are
presented in Sec. 10. Lastly, lists of symbols and abbreviations are provided in Appendices A and B, respectively. Note that a comprehensive table of contents is provided as PDF bookmarks.

## 2  Attributes of atmospheric particles

### 2.1  Notion of a particle

Let us represent aerosol, cloud, and precipitation particles as point particles. The particle state is then characterized by two
types of variables: position $\boldsymbol{x}$ and attributes $\boldsymbol{a}$. Attributes consist of several variables representing the particle's internal state, and the attributes considered in this study are $\boldsymbol{a} = \{r, \{m_\alpha^{\mathrm{sol}}\}, \{m_\beta^{\mathrm{insol}}\}, T^{\mathrm{fz}}, a, c, \rho^{\mathrm{i}}, m^{\mathrm{rime}}, n^{\mathrm{mono}}, \boldsymbol{v}\}$, i.e., liquid water amount, masses of soluble substances, masses of insoluble substances, freezing temperature, equatorial radius, polar radius, apparent density, rime mass, number of monomers, and velocity.

In this study, for simplicity, partially frozen/melted particles are not considered. We assume that each particle completely
freezes or melts instantaneously (see Secs. 4.1.4 and 4.1.5). Therefore, either the equivalent droplet radius $r$ or ice particle attributes $\{a, c, \rho^{\mathrm{i}}\}$ are always zero in our model. Furthermore, we assume that all particles contain soluble substances and are always deliquescent even when the humidity is low (see Sec. 4.1.6). Further, as a crude representation of "pre-activation", we do not allow the complete sublimation of an ice particle (see Sec. 4.1.7). Therefore, $r$ and $\{a, c, \rho^{\mathrm{i}}\}$ cannot be simultaneously zero.
In the remainder of this section, we provide a detailed explanation of each attribute.

## 2.2 Liquid water amount

The amount of liquid water contained in a particle is expressed by the volume-equivalent sphere's radius $r$. That is, the volume of water in a particle is $(4/3)\pi r^3$.

## 2.3 Masses of soluble and insoluble substances

Let $m_\alpha^{\text{sol}}$, $\alpha = 1, 2, \ldots, N^{\text{sol}}$ be the masses of soluble substances contained in the particle, and let $m_\beta^{\text{insol}}$, $\beta = 1, 2, \ldots, N^{\text{insol}}$ be the masses of insoluble substances.

## 2.4 Freezing temperature and ice nucleation active surface site

We only consider homogeneous freezing and condensation/immersion freezing in this study because these are dominant in mixed-phase clouds (e.g., Cui et al., 2006; De Boer et al., 2011; Murray et al., 2012).

Based on the "singular hypothesis" (Levine, 1950), we consider that each insoluble particle has its own freezing temperature $T^{\text{fz}}$, and that a supercooled droplet freezes as soon as the ambient temperature $T$ decreases below $T^{\text{fz}}$. The freezing process is described in detail in Sec. 4.1.4.

Particle's $T^{\text{fz}}$ is directly connected to the ice nucleation active surface site (INAS) density concept (e.g., Fletcher, 1969; Connolly et al., 2009; Niemand et al., 2012; Hoose and Möhler, 2012).

An INAS is a localized structure, such as lattice mismatches, cracks, and hydrophilic sites, on an insoluble substance's surface that catalyzes ice formation at temperatures lower than a specific temperature. INAS density $n_{\text{S}}(T)$ gives the accumulated number of INAS per unit surface area of the insoluble substance. Therefore, $n_{\text{S}}(T)$ is a function that increases as $T$ decreases. The freezing temperature $T^{\text{fz}}$ corresponds to the highest temperature at which the first INAS appears on the insoluble substance's surface. Let $A^{\text{insol}}$ be the insoluble substance's surface area. Then, the probability that $T^{\text{fz}}$ is larger than $T$ can be calculated as $P(T^{\text{fz}} > T) = 1 - \exp[-A^{\text{insol}} n_{\text{S}}(T)]$. The probability density function of $T^{\text{fz}}$ then becomes

$$p(T) = -\frac{dP(T^{\text{fz}} > T)}{dT} = -A^{\text{insol}} \frac{dn_{\text{S}}}{dT} e^{-A^{\text{insol}} n_{\text{S}}}. \tag{1}$$

We can determine $T^{\text{fz}}$ by selecting a random number that follows this probability distribution.

For mineral dust, biogenic substances, and soot, we can use the INAS density formulas of Niemand et al. (2012), Wex et al. (2015), and Ullrich et al. (2017), respectively. If a particle consists of multiple insoluble substances, we assume that $T^{\text{fz}}$ is the highest of all.

It is possible that a single INAS does not appear until $-38\,^\circ$C, meaning that the particle is ice nucleation (IN) inactive and will not freeze by immersion/condensation freezing but only by homogeneous freezing. To account for this, we set $T^{\text{fz}} = -38\,^\circ$C. If a particle contains only soluble substances, we also set $T^{\text{fz}} = -38\,^\circ$C.

There are various ice nucleation pathways (e.g., Kanji et al., 2017); however, in this study we do not consider other ice nucleation pathways, such as deposition nucleation, deliquescent freezing, pore freezing, and contact freezing. The possibility of extending our model to incorporate these mechanisms is discussed in Sec. 9.3.1.

## 2.5 Porous spheroid approximation of ice particles

Ice particles have diverse morphologies such as columns, hexagonal plates, dendrites, rimed crystals, graupel, hailstones, and aggregates (e.g., Magono and Lee, 1966; Kikuchi et al., 2013). Following the strategies of Chen and Lamb (1994a, b), Misumi et al. (2010), and Jensen and Harrington (2015), let us approximate each ice particle as a porous spheroid, which is characterized by three variables, namely equatorial radius $a$, polar radius $c$, and apparent density $\rho^{\mathrm{i}}$. That is, the ice particle's apparent volume is $V = (4\pi/3)a^2c$, and its mass can be evaluated as $m = \rho^{\mathrm{i}}V$. The two radii $a$ and $c$ represent the ice particle's spatial extent and $\rho^{\mathrm{i}}$ represents its internal structure. Let us define the aspect ratio as $\phi := c/a$. A spheroid is considered a prolate spheroid if $\phi > 1$, and columns could be approximated by prolate spheroids. In contrast, plates and dendrites are approximated by oblate spheroids, i.e., $\phi < 1$. If an ice particle is hollowed out or intricately branched, $\rho^{\mathrm{i}}$ becomes smaller than the ice crystal's true density $\rho^{\mathrm{i}}_{\mathrm{true}} \approx 916.8\,\mathrm{kg/m^3}$.

## 2.6 Rime mass and number of monomers

Following Brdar and Seifert (2018) we introduce two additional ice particle attributes, namely rime mass $m^{\mathrm{rime}}$ and number of monomers $n^{\mathrm{mono}}$. Rime mass $m^{\mathrm{rime}}$ records the mass of ice a particle has obtained through the riming process. The number of monomers $n^{\mathrm{mono}}$ is an integer representing the number of primary ice crystals in the particle. In this study, $m^{\mathrm{rime}}$ and $n^{\mathrm{mono}}$ are used only for analyzing the simulation results. Unlike the McSnow model of Brdar and Seifert (2018), this study's time evolution equations do not depend on $m^{\mathrm{rime}}$ or $n^{\mathrm{mono}}$, as will be detailed in Sec. 4.1.

## 2.7 Velocity

We approximate that each particle is always moving at its terminal velocity. Therefore, a particle's velocity $\boldsymbol{v}$ is a diagnostic attribute.

## 2.8 Effective number of attributes

In summary, particle attributes consist of $\boldsymbol{a} = \{r, \{m^{\mathrm{sol}}_\alpha\}, \{m^{\mathrm{insol}}_\beta\}, T^{\mathrm{fz}}, a, c, \rho^{\mathrm{i}}, m^{\mathrm{rime}}, n^{\mathrm{mono}}, \boldsymbol{v}\}$. We need the mass of insoluble substances $\{m^{\mathrm{insol}}_\beta, \beta = 1, 2, \ldots, N^{\mathrm{insol}}\}$ (and corresponding INAS densities) to specify freezing temperature $T^{\mathrm{fz}}$. However, as described in Sec. 4.1, time evolution equations do not depend on $\{m^{\mathrm{insol}}_\beta\}$. Rime mass $m^{\mathrm{rime}}$ and the number of monomers $n^{\mathrm{mono}}$ do not affect time evolution either. Particle velocity $\boldsymbol{v}$ is a diagnostic attribute. Therefore, the attributes directly relevant to time evolution are reduced to $\{r, \{m^{\mathrm{sol}}_\alpha\}, T^{\mathrm{fz}}, a, c, \rho^{\mathrm{i}}\}$. Compared to the warm cloud SDM model of Shima et al. (2009), we have introduced four new attributes.

## 3 Variables for moist air

We only consider dry air and water vapor for the gas phase and ignore other trace gases. In this section, we introduce several variables that describe the state of moist air: wind velocity $\boldsymbol{U} = (U, V, W)$, density of dry air $\rho_{\mathrm{d}}$, density of water vapor

$\rho_{\mathrm{v}}$, density of moist air $\rho := \rho_{\mathrm{d}} + \rho_{\mathrm{v}}$, specific humidity $q_{\mathrm{v}} := \rho_{\mathrm{v}}/\rho$, mass of dry air per unit mass of moist air $q_{\mathrm{d}} := \rho_{\mathrm{d}}/\rho$, temperature $T$, pressure $P$, and potential temperature of moist air $\theta := T/\Pi := T/(P/P_0)^{R/c_{\mathrm{p}}}$. Here, $P_0 = 1000$ hPa is a reference pressure; $R_{\mathrm{d}}$, $R_{\mathrm{v}}$, and $R := q_{\mathrm{d}}R_{\mathrm{d}} + q_{\mathrm{v}}R_{\mathrm{v}}$ are the gas constants of dry air, water vapor, and moist air, respectively; and $c_{\mathrm{pd}}$, $c_{\mathrm{pv}}$, and $c_{\mathrm{p}} := q_{\mathrm{d}}c_{\mathrm{pd}} + q_{\mathrm{v}}c_{\mathrm{pv}}$ are the isobaric specific heats of dry air, water vapor, and moist air, respectively. To
simplify notation, we introduce a variable representing the state of moist air: $\boldsymbol{G} := \{\boldsymbol{U}, \rho, q_{\mathrm{v}}, \theta, P, T\}$.

## 4   Time evolution equations of mixed-phase clouds

In this section, we describe our model's time evolution equations, first from cloud microphysics and then moist air fluid dynamics. Our model is detailed; however, it still falls short in completely describing mixed-phase cloud microphysics. To keep the model description concise, discussions on the shortcomings and how to overcome them are left for Sec. 9.

### 4.1   Cloud microphysics

Let us assign a unique index $i$ to each particle. This section explains the time evolution equations of particles $\{\{\boldsymbol{x}_i(t), \boldsymbol{a}_i(t)\}, i = 1, 2, \ldots, N_{\mathrm{r}}^{\mathrm{wp}}\}$. Here, $N_{\mathrm{r}}^{\mathrm{wp}}$ represents the total number of particles accumulated over the whole period. However, because of coalescence, precipitation, and other processes, some particles might not exist all the time, thus, we let $I_{\mathrm{r}}(t)$ be the set of particle indices existing in the domain at time $t$.

### 4.1.1   Advection and sedimentation

Particle $i$'s motion equation is

$$\frac{d}{dt}(m_i \boldsymbol{v}_i) = \boldsymbol{F}_i^{\mathrm{drg}} - m_i g \hat{\boldsymbol{z}}, \quad \frac{d\boldsymbol{x}_i}{dt} = \boldsymbol{v}_i, \tag{2}$$

where $m_i$ is the particle's mass, $\boldsymbol{F}_i^{\mathrm{drg}}$ is the force of drag from moist air, $g$ is Earth's gravity, and $\hat{\boldsymbol{z}}$ is the unit vector in the $z$ axis direction. Note that $-\boldsymbol{F}_i^{\mathrm{drg}}$ gives the reaction force acting on moist air. The momentum of moist air changes as described
in Eqs. (73) and (81).

If terminal velocity is reached, the motion equation becomes

$$\boldsymbol{v}_i = \boldsymbol{U}_i - \hat{\boldsymbol{z}} v_i^\infty, \quad \frac{d\boldsymbol{x}_i}{dt} = \boldsymbol{v}_i, \tag{3}$$

where $\boldsymbol{U}_i := \boldsymbol{U}(\boldsymbol{x}_i)$ is the $i$-th particle's ambient wind velocity, and $v_i^\infty$ is the terminal velocity, which is a function of attributes $\boldsymbol{a}_i$ and the state of the ambient air $\boldsymbol{G}_i$.

In this study, we assume that terminal velocity is always achieved instantaneously; however, this is a simplification. For example, the relaxation time of large droplets is a few seconds (Fig. 3 of Wang and Pruppacher, 1977) though that of micrometer-sized droplets is approximately $10^{-5}$ s (see, e.g., Eq. (1) of Chen et al., 2018, and the discussion that follows). The acceleration of particles can be considered by explicitly solving the motion equation (see, e.g., Naumann and Seifert, 2015), but extremely small time steps would be required for small particles.

The next two subsections explain the formulas used to calculate droplet and ice particle terminal velocities.

### 4.1.2 Droplet terminal velocity

To calculate droplet terminal velocity, we use the formula of Beard (1976): $v_i^\infty = v_{\text{Beard}}^\infty(\min(r_i, 3.5\text{mm}); \rho_i, P_i, T_i)$, where $\rho_i := \rho(\boldsymbol{x}_i)$ and $P_i := P(\boldsymbol{x}_i)$ are the density and pressure of ambient moist air, respectively. This formula applies to droplets with radii smaller than $3.5\,\text{mm}$. If we use the formula for droplets larger than this, the fall speed becomes unrealistically fast. Therefore, we use the fall speed of a droplet with a $3.5\,\text{mm}$ radius for droplets larger than the size limit.

### 4.1.3 Ice particle terminal velocity

For ice particle terminal velocity, we use the formula of Böhm (1989, 1992c, 1999): $v_i^\infty = v_{\text{Böhm}}^\infty(m_i, \phi_i, d_i, q_i; \rho_i, T_i)$, where $d_i$ is the characteristic length, and $q_i$ is the area ratio.

In Böhm's theory, $d_i$ is defined by $2a_i$, and $q_i$ is defined by the area ratio regarding circumscribed ellipse $q_i^{\text{ce}} := A_i / A_i^{\text{ce}}$, where $A_i$ is the projected area perpendicular to the flow direction, and $A_i^{\text{ce}}$ is the area of the circumscribed ellipse of $A_i$, i.e., the area of the smallest ellipse that completely contains $A_i$.

However, in this study, we start from a slightly different definition of $d_i$ and $q_i$, which we adopted mistakenly:

$$d_i = D_i := 2\max(a_i, c_i), \quad q_i = q_i^{\text{cc}} := A_i / A_i^{\text{cc}}, \tag{4}$$

where $D_i$ is the maximum dimension, $q_i^{\text{cc}}$ is the area ratio regarding circumcircle, and $A_i^{\text{cc}}$ is the area of the circumcircle of $A_i$, i.e., the area of the smallest circle that completely contains $A_i$.

Consequently, Eq. (4) underestimates the fall speeds of columnar ice particles. Nevertheless, based on the assessment detailed in Sec. 9.2, we will confirm that this difference does not change the results of our simulation significantly, and hence, we conclude that this flaw causes only a minor impact on this study. We also note that in Sec. 9.2 we will develop and release a fixed version of the model, SCALE-SDM 0.2.5-2.2.2.

In our model, we assume that ice particles are falling with their maximum dimension perpendicular to the flow direction. Therefore, the circumcircle area becomes $A_i^{\text{cc}} = \pi \max(a_i, c_i)^2$. The projected area $A_i$ can be roughly evaluated by the area of the circumscribed ellipse $A_i^{\text{ce}} = \pi a_i \max(a_i, c_i)$; however, we must subtract pores and indentations at boundaries from $A_i^{\text{ce}}$. We assume that the ratio $A_i / A_i^{\text{ce}}$ is a power of the volume fraction $\rho_i^{\text{i}} / \rho_{\text{true}}^{\text{i}}$, and that the exponent $\kappa$ is a function of the aspect ratio $\phi_i$:

$$A_i = A_i^{\text{ce}} \left( \frac{\rho_i^{\text{i}}}{\rho_{\text{true}}^{\text{i}}} \right)^{\kappa(\phi_i)}. \tag{5}$$

Based on the following arguments, we propose a value $\kappa$ of the form

$$\kappa(\phi_i) = \exp(-\phi_i). \tag{6}$$

Following Jensen and Harrington (2015), we assume $\kappa \to 1$ as $\phi_i \to 0$, and $\kappa \to 0$ as $\phi_i \to \infty$. $\phi_i \ll 1$ means that the ice particle is thin and extends horizontally. Therefore, we can expect that the structure is uniform along the vertical axis and that the ratio $A_i / A_i^{\text{ce}}$ is equal to the volume fraction $\rho_i^{\text{i}} / \rho_{\text{true}}^{\text{i}}$. Thus, $\kappa(\phi_i = 0) = 1$. At the other extreme, $\phi_i \gg 1$ indicates that

the ice particle is columnar. Such ice crystals typically hollow inward along their basal face; therefore, the volume fraction $\rho_i^{\mathrm{i}}/\rho_{\mathrm{true}}^{\mathrm{i}}$ will not affect the ratio $A_i/A_i^{\mathrm{ce}}$. Thus, $\kappa(\phi_i \to \infty) = 0$.

For $\phi_i \approx 1$, Jensen and Harrington (2015) argued that $(\rho_i^{\mathrm{i}}/\rho_{\mathrm{true}}^{\mathrm{i}})^\kappa = 1$, i.e., $\kappa = 0$. However, this cannot be justified for aggregates with low apparent densities. Thus, we estimate $\kappa$ through a dimensional analysis. We assume that the power laws $m_i \propto D_i^\beta$ and $A_i \propto D_i^{\beta/s}$ hold. Thus, by the definition of apparent density, $\rho_i^{\mathrm{i}} = m_i/((4/3)\pi a_i^2 c_i) \propto D_i^{\beta-3}$. From Eq. (5), $D_i^{\beta/s} = D_i^2 D_i^{(\beta-3)\kappa}$. Hence, $\kappa = (2s - \beta)/\{s(3 - \beta)\}$ holds. Schmitt and Heymsfield (2010) estimated that $(\beta, s) = (2.22, 1.30)$ for aggregates observed during the Cirrus Regional Study of Tropical Anvils and Cirrus Layers–Florida Area Cirrus Experiment (CRYSTAL-FACE) field project. Therefore, $\kappa = 0.375$ for CRYSTAL-FACE aggregates. They also estimated that $(\beta, s) = (2.20, 1.25)$ for aggregates observed during an Atmospheric Radiation Measurement (ARM) field project, which results in $\kappa = 0.300$.

The $\kappa$ given by Eq. (6) yields $\kappa(0) = 1$, $\kappa(1) = 0.368$, and $\kappa(\infty) = 0$, which agree with the aforementioned estimation.

### 4.1.4 Immersion/condensation and homogeneous freezing

As explained in Sec. 2.4, a supercooled droplet freezes when the ambient temperature drops below its freezing temperature. This section provides a more precise description of when and how freezing occurs in our model.

We consider that the $i$-th particle freezes immediately when the following three conditions are all satisfied: (1) The particle is a droplet, i.e., $r_i > 0$; (2) the ambient water vapor is supersaturated over liquid water, i.e., $e_i > e_s^{\mathrm{w}}(T_i)$; and (3) the ambient temperature is lower than the particle's freezing temperature, i.e., $T_i < T_i^{\mathrm{fz}}$. Here, $e_i := e(\boldsymbol{x}_i)$ and $T_i := T(\boldsymbol{x}_i)$ are the ambient vapor pressure and temperature of the $i$-th particle, respectively, and $e_s^{\mathrm{w}}(T)$ is the saturation vapor pressure over a planar liquid water surface at temperature $T$.

We assume that the resulting ice crystal is spherical, with the true ice crystal density $\rho_{\mathrm{true}}^{\mathrm{i}}$. Therefore, attributes are initiated as follows: $r_i' = 0$, $a_i' = c_i' = r_i(\rho^{\mathrm{w}}/\rho_{\mathrm{true}}^{\mathrm{i}})^{1/3}$, $\rho_i^{\mathrm{i}\prime} = \rho_{\mathrm{true}}^{\mathrm{i}}$, $n_i^{\mathrm{mono}\prime} = 1$, and $m_i^{\mathrm{rime}\prime} = 0$. The primed variables here denote values after the update, and $\rho^{\mathrm{w}}$ is the density of liquid water. $\{m_{\alpha i}^{\mathrm{sol}}\}$, $\{m_{\beta i}^{\mathrm{insol}}\}$, and $T_i^{\mathrm{fz}}$ remain unchanged.

When freezing occurs, each particle releases latent heat of fusion to the moist air, as described in Eqs. (74), (79), and (80).

### 4.1.5 Melting

When ambient temperature rises above $0\,^\circ\mathrm{C}$, we consider that melting occurs immediately. Thus, the attributes are updated as follows: $r_i' = (a_i^2 c_i \rho_i^{\mathrm{i}}/\rho^{\mathrm{w}})^{1/3}$ and $a_i' = c_i' = \rho_i^{\mathrm{i}\prime} = n_i^{\mathrm{mono}\prime} = m_i^{\mathrm{rime}\prime} = 0$. $\{m_{\alpha i}^{\mathrm{sol}}\}$, $\{m_{\beta i}^{\mathrm{insol}}\}$, and $T_i^{\mathrm{fz}}$ remain unchanged. When melting occurs, each particle absorbs latent heat of fusion from the moist air, as indicated in Eqs. (74), (79), and (80).

### 4.1.6 Condensation and evaporation

Following, e.g., Rogers and Yau (1989), the time evolution equation describing droplet growth by condensation/evaporation can be derived as follows.

The growth rate is identical to vapor flux at the droplet surface. If the diffusion of vapor around the droplet is in a quasi-steady state, we obtain

$$\frac{dm_i}{dt} = 4\pi r_i D_{\mathrm{v}}(\rho_{\mathrm{v}i} - \rho_{\mathrm{v}i}^{\mathrm{sfc}}). \tag{7}$$

Here, $D_{\mathrm{v}}$ is water vapor's diffusivity in air, $\rho_{\mathrm{v}i} := \rho_{\mathrm{v}}(\boldsymbol{x}_i)$ is the ambient moist air's water vapor density, and $\rho_{\mathrm{v}i}^{\mathrm{sfc}}$ is water vapor density at the surface of the droplet.

If we further assume that thermal diffusion is also in a quasi-steady state, and that surface temperature $T_i^{\mathrm{sfc}}$ and ambient temperature $T_i$ are close to each other, i.e., $(T_i^{\mathrm{sfc}} - T_i)/T_i \ll 1$, Eq. (7) can be reduced to

$$r_i \frac{dr_i}{dt} = \frac{1}{\rho^{\mathrm{w}}(F_{\mathrm{k}}^{\mathrm{w}} + F_{\mathrm{d}}^{\mathrm{w}})} \left\{ S_i^{\mathrm{w}} - \frac{e_{\mathrm{s}i}^{\mathrm{w,eff}}}{e_{\mathrm{s}}^{\mathrm{w}}(T_i)} \right\}, \tag{8}$$

where $S_i^{\mathrm{w}} := e_i/e_{\mathrm{s}}^{\mathrm{w}}(T_i)$ is the ambient saturation ratio over liquid water, and

$$F_{\mathrm{k}}^{\mathrm{w}} = \left( \frac{L_{\mathrm{v}}}{R_{\mathrm{v}} T_i} - 1 \right) \frac{L_{\mathrm{v}}}{k T_i}, \quad F_{\mathrm{d}}^{\mathrm{w}} = \frac{R_{\mathrm{v}} T_i}{D_{\mathrm{v}} e_{\mathrm{s}}^{\mathrm{w}}(T_i)}, \tag{9}$$

where $L_{\mathrm{v}}$ is the latent heat of vaporization, $k$ is the thermal conductivity of moist air, and $e_{\mathrm{s}i}^{\mathrm{w,eff}}$ is the effective saturation vapor pressure regarding the $i$-th droplet's surface. Following Köhler's theory (Köhler, 1936), an approximate formula of $e_{\mathrm{s}i}^{\mathrm{w,eff}}$ can be derived as

$$\frac{e_{\mathrm{s}i}^{\mathrm{w,eff}}}{e_{\mathrm{s}}^{\mathrm{w}}(T_i)} = 1 + \frac{a(T_i)}{r_i} - \frac{b(\{m_{\alpha i}^{\mathrm{sol}}\})}{r_i^3}, \tag{10}$$

where $a \approx 3.3 \times 10^{-5}\,\mathrm{cm\,K}/T_i$, $b \approx 4.3\,\mathrm{cm}^3 \sum_\alpha I_\alpha m_{\alpha i}^{\mathrm{sol}}/M_\alpha^{\mathrm{sol}}$, $I_\alpha$ is the van't Hoff factor, which represents the degree of ionic dissociation, and $M_\alpha^{\mathrm{sol}}$ is the molecular weight of the solute $\alpha$. The second and third terms of Eq. (10) account for curvature and solute effects, respectively.

The growth of a droplet by condensation/evaporation is governed by Eqs. (8)-(10) in our model. When a droplet or an ice particle falls through the air, the flow around it enhances the diffusional growth, a phenomenon known as the ventilation effect. It does not essentially affect the growth of droplets smaller than $50\,\mu\mathrm{m}$ in radius (see Sec. 13.2.3 of Pruppacher and Klett (1997)). Therefore, for simplicity, we do not consider the ventilation effect on droplets in this study. Notably, Eqs. (8)-(10) also describe the respective activation and deactivation of cloud droplets from and to aerosol particles (see, e.g., Arabas and Shima, 2017; Hoffmann, 2017; Abade et al., 2018).

Vapor and latent heat couplings to moist air through condensation and evaporation are calculated by Eqs. (71), (72), (74), (76), (77), and (79).

### 4.1.7 Deposition and sublimation

The shapes of ice crystals formed by depositional growth exhibit strong dependencies on temperature and, to a lesser extent, supersaturation (e.g., Nakaya, 1954; Hallett and Mason, 1958; Kobayashi, 1961). The former is known as the primary growth habit and the latter as the secondary growth habit. The primary growth habit determines the preferred growth direction, i.e.,

columnar or planar, and the secondary growth habit determines the mode of growth, i.e., whether the columnar crystal becomes solid or hollow, and whether the planar crystal becomes plate-like, sectored, or dendritic. In this study, we use the model of Chen and Lamb (1994a) with various modifications.

The mass growth rate can be derived similarly to Eqs. (7) and (8):

$$\frac{dm_i}{dt} = 4\pi C D_{\mathrm{v}}(\rho_{vi} - \rho_{vi}^{\mathrm{sfc}})\bar{f}_{\mathrm{vnt}} = 4\pi C \frac{S_i^{\mathrm{i}} - 1}{F_{\mathrm{k}}^{\mathrm{i}} + F_{\mathrm{d}}^{\mathrm{i}}}\bar{f}_{\mathrm{vnt}}, \tag{11}$$

where $S_i^{\mathrm{i}} := e_i/e_{\mathrm{s}}^{\mathrm{i}}(T_i)$ is the ambient saturation ratio over ice, and $e_{\mathrm{s}}^{\mathrm{i}}(T)$ is the saturation vapor pressure over ice at temperature $T$,

$$F_{\mathrm{k}}^{\mathrm{i}} = \left(\frac{L_{\mathrm{s}}}{R_{\mathrm{v}}T_i} - 1\right)\frac{L_{\mathrm{s}}}{kT_i}, \quad F_{\mathrm{d}}^{\mathrm{i}} = \frac{R_{\mathrm{v}}T_i}{D_{\mathrm{v}}e_{\mathrm{s}}^{\mathrm{i}}(T_i)}, \tag{12}$$

where $L_{\mathrm{s}}$ is the latent heat of sublimation, $C = C(a_i, c_i)$ is the electric capacitance of the spheroid, and $\bar{f}_{\mathrm{vnt}}$ is the particle-averaged ventilation coefficient.

The exact form of capacitance $C(a_i, c_i)$ is given by Chen and Lamb (1994a). $C \approx (2a_i + c_i)/3$ gives a good approximation for $\phi_i \approx 1$.

The coefficient $\bar{f}_{\mathrm{vnt}}$ accounts for the ventilation effect, i.e., the enhancement of diffusional growth by air flow. Hall and Pruppacher (1976) suggested that $\bar{f}_{\mathrm{vnt}}$ could be described by

$$\bar{f}_{\mathrm{vnt}} = b_1 + b_2 X^\gamma, \tag{13}$$

where $(b_1, b_2, \gamma) = (1.0, 0.14, 2)$ for $X \leq 1$, $(b_1, b_2, \gamma) = (0.86, 0.28, 1)$ for $X > 1$, $X = N_{\mathrm{Sc}}^{1/3}(N_{\mathrm{Re}i}^{\mathrm{i}})^{1/2}$, $N_{\mathrm{Sc}} = \mu/(\rho D_{\mathrm{v}})$ is the Schmidt number, $N_{\mathrm{Re}i}^{\mathrm{i}} = \rho v_i^\infty D_i/\mu$ is the Reynolds number of ice particle $i$, and $\mu$ is the dynamic viscosity of moist air.

Note that $m_i$ in Eq. (11) can become zero through sublimation over a finite time. However, in this study, we prohibit complete sublimation, and instead, we impose a limiter to $dm_i$ as follows:

$$dm_i = \max(dm_i, m_{\mathrm{min}}^{\mathrm{i}} - m_i), \tag{14}$$

where $m_{\mathrm{min}}^{\mathrm{i}}$ is an arbitrary small mass taken from the mass of a spherical ice particle with a radius of $1\,\mathrm{nm}$ and the true ice density $\rho_{\mathrm{true}}^{\mathrm{i}}$. This is a crude representation of pre-activation (see, e.g., Marcolli, 2017, for a review). Each particle keeps the memory of ice activation until the ambient temperature rises above $0\,^\circ\mathrm{C}$. A particle with $m_{\mathrm{min}}^{\mathrm{i}}$ ice grows immediately after the ambient air is supersaturated over ice, irrespective of its freezing temperature $T_i^{\mathrm{fz}}$.

In Chen and Lamb's (1994a) model, the primary growth habit is expressed by an empirical function known as the inherent growth ratio $\Gamma(T)$, which modulates the $c$-axis to $a$-axis growth rate ratio:

$$\frac{dc_i}{da_i} = \Gamma(T_i)f_{\mathrm{vnt}}\frac{c_i}{a_i} =: \Gamma^*\frac{c_i}{a_i}, \tag{15}$$

where $f_{\mathrm{vnt}}$ is the primary growth habit's ventilation coefficient, and $\Gamma^*$ is the effective inherent growth ratio, including the ventilation effect.

For purely diffusional growth, $dc_i/da_i = c_i/a_i$ holds; therefore, the aspect ratio does not change, i.e., $d\phi_i = 0$. $\Gamma(T)$ represents the lateral redistribution of vapor on the ice crystal surface through kinetic processes. We use the $\Gamma(T)$ proposed by Chen and Lamb (1994a), but set $\Gamma(T) = 1$ for $D < 10\,\mu\text{m}$, as observations suggest that ice crystals are quasi-spherical if $D < 60\,\mu\text{m}$ (Baran, 2012; Korolev and Isaac, 2003; Lawson et al., 2008). Additionally, the $\Gamma(T)$ provided in Chen and Lamb (1994a) is

5 for temperatures between $-30\,^{\circ}\text{C}$ and $0\,^{\circ}\text{C}$. For lower temperatures, we simply assume

$$\Gamma(T) = \Gamma(-30\,^{\circ}\text{C}) \approx 1.28, \quad \text{for } T < -30\,^{\circ}\text{C}. \tag{16}$$

The ventilation coefficient $f_{\text{vnt}}$ represents the preferential enhancement of vapor flux toward the ice crystal's major axis because of the air flow around it. Chen and Lamb (1994a) derived a $f_{\text{vnt}}$ of the form

$$f_{\text{vnt}} = \frac{b_1 + b_2 X^\gamma \left(c_i/C\right)^{1/2}}{b_1 + b_2 X^\gamma \left(a_i/C\right)^{1/2}}. \tag{17}$$

The secondary growth habit is expressed by deposition density $\rho_{\text{dep}}$, which represents the apparent density of the ice fraction newly created by deposition. Then, the change in ice particle volume $dV_i$ is given by

$$dV_i = \frac{dm_i}{\rho_{\text{dep}}}, \quad \text{for } dm_i \geq 0 \text{ (deposition)}. \tag{18}$$

Deposition density $\rho_{\text{dep}}$ can be expressed as

$$\rho_{\text{dep}} = \begin{cases} \rho_{\text{true}}^{\text{i}}, & \text{for } \Gamma(T_i) < 1 \ \wedge \ a_i < 100\,\mu\text{m}; \\ \rho_{\text{dep}}^{\text{CL94}}, & \text{otherwise.} \end{cases} \tag{19}$$

Here, following Jensen and Harrington (2015), we assume that planar crystal branching does not occur if the equatorial radius $a_i$ is smaller than $100\,\mu\text{m}$. $\rho_{\text{dep}}^{\text{CL94}}$ is an empirical formula of deposition density proposed by Chen and Lamb (1994a),

$$\rho_{\text{dep}}^{\text{CL94}} = \rho_{\text{true}}^{\text{i}} \exp\left[-\frac{3\max(\Delta\rho_i - 0.05\,\text{g}\,\text{m}^{-3}, 0)}{\Gamma(T_i)\,\text{g}\,\text{m}^{-3}}\right], \tag{20}$$

where $\Delta\rho_i := \rho_{\text{vi}} - \rho_{\text{vi}}^{\text{sfc}}$. From Eq. (11), $\Delta\rho_i$ becomes

$$\Delta\rho_i = \frac{S_i^{\text{i}} - 1}{D_{\text{v}}(F_{\text{k}}^{\text{i}} + F_{\text{d}}^{\text{i}})}. \tag{21}$$

Here, following Miller and Young (1979), we limit $\rho_{\text{vi}}$ by water saturation and replace the $\Delta\rho_i$ in (20) with

$$(\Delta\rho_i)^{\downarrow} = \frac{\min\left(S_i^{\text{i}}, e_{\text{s}}^{\text{w}}(T_i)/e_{\text{s}}^{\text{i}}(T_i)\right) - 1}{D_{\text{v}}(F_{\text{k}}^{\text{i}} + F_{\text{d}}^{\text{i}})}. \tag{22}$$

For sublimation, the particle volume change $dV_i$ is given by

$$dV_i = \frac{dm_i}{\rho_{\text{sbl}}}, \quad \text{for } dm_i < 0 \text{ (sublimation)}, \tag{23}$$

where sublimation density $\rho_{\text{sbl}}$ represents the apparent density of the ice fraction removed by sublimation. For simplicity, we assume that the ice particle's apparent density will not be changed through sublimation, i.e.,

$$\rho_{\text{sbl}} = \rho_i^{\text{i}}. \tag{24}$$

We can now calculate the attributes at time $t + dt$. The apparent density becomes

$$\rho_i^{\text{i}}(t + dt) = \frac{m_i + dm_i}{V_i + dV_i}, \tag{25}$$

where $dm_i$ is given in Eqs. (11) and (14), and $dV_i$ is given in Eqs. (18) and (23).

From Eq. (15) and the definition of volume $V_i = (4\pi/3)a_i^2 c_i$, after $dt$, the two radii become

$$a_i(t + dt) = a_i \exp\left(\frac{d\log V_i}{2 + \Gamma^*}\right), \tag{26}$$

$$c_i(t + dt) = c_i \exp\left(\frac{\Gamma^* d\log V_i}{2 + \Gamma^*}\right). \tag{27}$$

Applying those equations to a small ice particle's sublimation creates an extremely small planar or columnar ice particle. However, observations suggest that ice crystals are quasi-spherical if $D < 60\,\mu\text{m}$ (Baran, 2012; Korolev and Isaac, 2003; Lawson et al., 2008). Therefore, we regard the ice particle as spherical with the true ice density if the minor axis predicted by Eqs. (26) and (27) is smaller than $1\,\mu\text{m}$. That is, if $\min\{a_i(t + dt), c_i(t + dt)\} < 1\,\mu\text{m}$,

$$\rho_i^{\text{i}\prime}(t + dt) = \rho_{\text{true}}^{\text{i}}, \tag{28}$$

$$a_i'(t + dt) = c_i'(d + dt) = \left(\frac{m_i + dm_i}{(4\pi/3)\rho_i^{\text{i}\prime}(t + dt)}\right)^{\frac{1}{3}}, \tag{29}$$

where primed variables indicate values after correction.

For simplicity we assume that the rime mass fraction $m_i^{\text{rime}}/m_i$ does not change through sublimation, following Brdar and Seifert (2018):

$$m_i^{\text{rime}}(t + dt) = \begin{cases} m_i^{\text{rime}}, & \text{for } dm_i \geq 0; \\ m_i^{\text{rime}} \dfrac{m_i + dm_i}{m_i}, & \text{for } dm_i < 0. \end{cases} \tag{30}$$

Vapor and latent heat couplings to moist air through deposition and sublimation are calculated by Eqs. (71), (72), (74), (76), (78), and (79).

In this section, we detailed the deposition and sublimation model used in SCALE-SDM; however, there is significant room for improvement. For example, as we will discuss in Sec. 9.1.4, using $\Gamma(T)$ for sublimation is questionable. Instead, we propose using $\Gamma(T) = 1$ for sublimation (Eq. (110)), and validate this correction in Sec. 9.1.5. Furthermore, in Sec. 9.2, to prohibit the creation of unnaturally slender ice particles, we will propose to impose a limiter to the effective inherent growth ratio $\Gamma^*$ (Eq. (123)). Several other issues of our deposition/sublimation model, such as the representation of polycrystals, will be discussed in Sec. 9.3.5.

### 4.1.8 Stochastic description of coalescence, riming, and aggregation

Particle coalescence, riming, and aggregation can be considered a stochastic process. Following Gillespie (1972), consider a region with volume $\Delta V$. If $\Delta V$ is sufficiently small, we can consider that particles within this region are well-mixed, e.g., by atmospheric turbulence (see, e.g., Shima et al., 2009; Dziekan and Pawlowska, 2017). Then, all particle pairs in the volume can collide and coalesce/rime/aggregate during an infinitesimal time interval $dt$. The probability that a particle pair $j$ and $k$ inside $\Delta V$ will collide and coalesce/rime/aggregate within an infinitesimal time interval $(t, t + dt)$ is given by

$$P_{jk} = K(\boldsymbol{a}_j, \boldsymbol{a}_k; \boldsymbol{G}) \frac{dt}{\Delta V}, \tag{31}$$

where the function $K(\boldsymbol{a}_j, \boldsymbol{a}_k; \boldsymbol{G})$ is called the collision-coalescence/riming/aggregation kernel, and $\boldsymbol{G}$ denotes the state of the moist air in $\Delta V$.

In this study, we consider coalescence, riming, and aggregation induced by differential gravitational settling of particles because this mechanism is dominant in mixed-phase clouds.

### 4.1.9 Coalescence between two droplets

First, we consider droplet coalescence, which accounts for the formation of rain droplets from cloud droplets (autoconversion), the collection of cloud droplets by rain droplets (accretion), and the coalescence of two rain droplets (selfcollection).

The collision-coalescence kernel is given by

$$K_{\text{coal}} = E_{\text{coal}}(r_j, r_k) \pi (r_j + r_k)^2 |v_j^\infty - v_k^\infty|, \tag{32}$$

where $E_{\text{coal}}(r_j, r_k)$ is the collection efficiency of collision-coalescence, which can be decomposed into $E_{\text{coal}} = E_{\text{coal}}^{\text{collis}} E_{\text{coal}}^{\text{coal}}$. Here, collision efficiency $E_{\text{coal}}^{\text{collis}}$ considers the effect that a smaller droplet is swept aside by the flow around a larger droplet, or a droplet being caught in the wake of a similarly sized droplet collides on the downstream side. We adopt the collision efficiency used in Seeßelberg et al. (1996) and Bott (1998). Here, Davis (1972) and Jonas (1972) are used for small droplets, and Hall (1980) for larger droplets, with modifications to the collector droplet radius range $70\,\mu\text{m}$–$300\,\mu\text{m}$ to incorporate the wake effect suggested by Lin and Lee (1975). Not all the collisions end up with coalescence. Rebound or breakup (fragmentation) could also occur. Coalescence efficiency $E_{\text{coal}}^{\text{coal}}$ represents the fraction of collisions that result in permanent coalescence. In this study, we assume $E_{\text{coal}}^{\text{coal}} = 1$ for simplicity.

If coalescence takes place, droplets $j$ and $k$ then merge into a single droplet. Thus, we keep $j$ and remove $k$ from the system. The attributes of the new droplet $j$ can be calculated as follows:

$$r_j' = (r_j^3 + r_k^3)^{\frac{1}{3}}, \tag{33}$$

$$m_{\alpha j}^{\text{sol}\prime} = m_{\alpha j}^{\text{sol}} + m_{\alpha k}^{\text{sol}}, \quad \alpha = 1, 2, \ldots, N^{\text{sol}} \tag{34}$$

$$m_{\beta j}^{\text{insol}\prime} = m_{\beta j}^{\text{insol}} + m_{\beta k}^{\text{insol}}, \quad \beta = 1, 2, \ldots, N^{\text{insol}} \tag{35}$$

$$T_j^{\text{fz}\prime} = \max(T_j^{\text{fz}}, T_k^{\text{fz}}), \tag{36}$$

where primed values indicate the resultant droplet. Here, we assumed that the resultant particle's $T_j^{\text{fz}\prime}$ is given by $\max(T_j^{\text{fz}}, T_k^{\text{fz}})$, i.e., the higher freezing temperature of the two constituent particles. We also assume that the same applies to riming and aggregation.

Let us emphasize that the stochastic model introduced in this section describes the underlying mathematical model of the coalescence process, not the Monte Carlo algorithm of SDM that solves the stochastic process numerically. In the preceding paragraph, droplet $k$ was removed from the system because both $j$ and $k$ are real particles. On the contrary, in the SDM, the number of super-particles is (almost always) conserved through coalescence (Shima et al., 2009).

### 4.1.10 Riming between an ice particle and a droplet

Riming usually refers to the collection of small supercooled droplets by a larger ice particle, but we also include the collection of small ice particles by a larger droplet. The latter case could be regarded as a type of contact freezing. However, ice particles grow preferentially when ice particles and supercooled droplets coexist (Wegener-Bergeron-Findeisen mechanism). Therefore, we can expect that the latter case happens less frequently in mixed-phase clouds.

Hereafter we assume, without loss of generality, that particle $j$ is an ice particle and particle $k$ is a droplet. The collision-riming kernel is expressed as

$$K_{\text{rime}} = E_{\text{rime}} A_{\text{g}} |v_j^\infty - v_k^\infty|, \tag{37}$$

where $E_{\text{rime}}$ is the collision-riming collection efficiency and $A_{\text{g}}$ is the geometric cross-sectional area of $j$ and $k$.

Figure 1 of Wang and Ji (2000) defines $A_{\text{g}}$ for riming, but calculating it rigorously for porous spheroid models is impossible. Thus, we approximate $A_{\text{g}}$ by

$$A_{\text{g}} = \pi(a_j + r_k)\{\max(a_j, c_j) + r_k\} - (A_j^{\text{ce}} - A_j), \tag{38}$$

i.e., the indentation of the ice particle $(A_j^{\text{ce}} - A_j)$ is subtracted from the area of an ellipse with semi-axes $(a_j + r_k)$ and $\{\max(a_j, c_j) + r_k\}$. Therefore, if $r_k \ll a_j, c_j$, then $A_{\text{g}} \approx A_j$. At the other extreme, if $r_k \gg a_j, c_j$, then $A_{\text{g}} \approx \pi(a_j + r_k)\{\max(a_j, c_j) + r_k\}$.

To evaluate collision-riming collection efficiency $E_{\text{rime}}$, we combine formulas proposed by Beard and Grover (1974) and Erfani and Mitchell (2017).

If $v_j^\infty < v_k^\infty$, we consider droplet $k$ as the collector and adopt the formula of Beard and Grover (1974):

$$E_{\text{rime}} = E_{\text{BG74}}(p^{\text{i/w}}, N_{\text{Re}k}^{\text{w}}, N_{\text{St}}^{\text{i/w}}), \tag{39}$$

where $p^{\text{i/w}} := r_j^{\text{i}}/r_k$, $r_j^{\text{i}} := (a_j^2 c_j)^{1/3}$, $N_{\text{Re}k}^{\text{w}} = \rho v_k^\infty 2 r_k / \mu$ is the Reynolds number of droplet $k$, $N_{\text{St}}^{\text{i/w}} = (p^{\text{i/w}})^2 \rho_j^{\text{i}} N_{\text{Re}k}^{\text{w}} C_{\text{SC}} / (9\rho)$ is the Stokes impaction parameter when droplet $k$ is collecting an ice particle, and $C_{\text{SC}}$ is the Cunningham slip correction factor.

If $v_j^\infty \geq v_k^\infty$, we consider ice particle $j$ to be the collector. For spherical ice particle $\phi_j \approx 1$, we again use the formula of Beard and Grover (1974), but replace the Stokes impaction parameter $N_{\text{St}}^{\text{i/w}}$ with the mixed Froude number $N_{\text{mFr}}$ following Hall (1980), Rasmussen and Heymsfield (1985), and Heymsfield and Pflaum (1985). For columnar and planar ice particles,

we use formulas $E_{\text{EM17}}^{\text{clm}}$ and $E_{\text{EM17}}^{\text{pln}}$ from Erfani and Mitchell (2017), which were obtained by fitting the numerical results of Wang and Ji (2000). For the intermediate case, we calculate an average weighted by the aspect ratio $\phi_j$. For $\phi_j \leq 1$ (planar),

$$E_{\text{rime}} = \phi_j E_{\text{BG74}}(p^{\text{w/i}}, N_{\text{Re}j}^{\text{i}}, N_{\text{mFr}}) $$
$$+ (1 - \phi_j) E_{\text{EM17}}^{\text{pln}}(N_{\text{Re}j}^{\text{i}}, N_{\text{mFr}}). \tag{40}$$

For $\phi_j > 1$ (columnar),

$$E_{\text{rime}} = \frac{1}{\phi_j} E_{\text{BG74}}(p^{\text{w/i}}, N_{\text{Re}j}^{\text{i}}, N_{\text{mFr}})$$
$$+ \left(1 - \frac{1}{\phi_j}\right) E_{\text{EM17}}^{\text{clm}}(N_{\text{Re}j}^{\text{clm}}, N_{\text{mFr}}). \tag{41}$$

Here, $p^{\text{w/i}} := 1/p^{\text{i/w}} = r_k/r_j^{\text{i}}$, $N_{\text{mFr}} = (v_j^\infty - v_k^\infty)v_k^\infty/(gD_j/2)$, and $N_{\text{Re}j}^{\text{clm}} = \rho v_j^\infty 2a_j/\mu$ is the Reynolds number based on the width of column $2a_j$. Note that there is a typo in Eq. (19) of Erfani and Mitchell (2017), i.e., the two case conditions are opposite.

If riming takes place, the ice particle $j$ and droplet $k$ merge and instantaneously freeze into a single ice particle. Thus, we keep $j$ and remove $k$ from the system.

If $\max(a_j, c_j) < r_k$, we assume that the resultant ice particle is spherical with the true ice density:

$$\rho_j^{\text{i}\prime} = \rho_{\text{true}}^{\text{i}}, \tag{42}$$

$$a_j' = c_j' = \left(\frac{m_j + m_k}{(4\pi/3)\rho_{\text{true}}^{\text{i}}}\right)^{\frac{1}{3}}, \tag{43}$$

$$m_j^{\text{rime}\prime} = m_j^{\text{rime}} + m_k, \tag{44}$$

$$n_j^{\text{mono}\prime} = n_j^{\text{mono}}, \tag{45}$$

$$m_{\alpha j}^{\text{sol}\prime} = m_{\alpha j}^{\text{sol}} + m_{\alpha k}^{\text{sol}}, \quad \alpha = 1, 2, \ldots, N^{\text{sol}}, \tag{46}$$

$$m_{\beta j}^{\text{insol}\prime} = m_{\beta j}^{\text{insol}} + m_{\beta k}^{\text{insol}}, \quad \beta = 1, 2, \ldots, N^{\text{insol}}, \tag{47}$$

$$T_j^{\text{fz}\prime} = \max(T_j^{\text{fz}}, T_k^{\text{fz}}), \tag{48}$$

where primed values indicate the resultant ice particle.

If $\max(a_j, c_j) \geq r_k$, we preserve the ice particle's maximum dimension, i.e., $D_j' = D_j$, until the ice particle becomes quasi-spherical. This accounts for the gradual growth of an unrimed ice crystal to a graupel particle with a quasi-spherical shape. This filling-in simplification was introduced by Heymsfield (1982), and is used in various models (e.g., Chen and Lamb, 1994b; Morrison and Grabowski, 2008, 2010; Jensen and Harrington, 2015; Morrison and Milbrandt, 2015). As graupels have an aspect ratio of approximately 0.8 (Heymsfield, 1978), we preserve the minor dimension if $0.8 < \phi_j \leq 1/0.8 = 1.25$, which mimics graupel's tumbling. When an accreted droplet freezes, the air will be trapped inside. Let rime density $\rho_{\text{rime}}$ be the

frozen droplet's apparent density. Then, for $\phi_j \leq 0.8$ (planar) and $1.0 < \phi_j \leq 1.25$ (columnar but quasi-spherical),

$$\rho_j^{i\prime} = \frac{m_j + m_k}{V_j + m_k/\rho_{\text{rime}}}, \tag{49}$$

$$a_j' = a_j, \tag{50}$$

$$c_j' = \frac{V_j + m_k/\rho_{\text{rime}}}{(4\pi/3)a_j^2}. \tag{51}$$

Other attributes are updated using Eqs. (44)–(48). For $\phi_j > 1.25$ (columnar) and $0.8 < \phi_j \leq 1.0$ (planar but quasi-spherical),

$$\rho_j^{i\prime} = \frac{m_j + m_k}{V_j + m_k/\rho_{\text{rime}}}, \tag{52}$$

$$a_j' = \left\{ \frac{V_j + m_k/\rho_{\text{rime}}}{(4\pi/3)c_j} \right\}^{\frac{1}{2}}, \tag{53}$$

$$c_j' = c_j. \tag{54}$$

Other attributes are updated using Eqs. (44)–(48). Following Chen and Lamb (1994b), we use the formula of Heymsfield and Pflaum (1985) to calculate rime density $\rho_{\text{rime}}$:

$$\rho_{\text{rime}} = \max\{\min\{\rho_{\text{rime}}^{\text{HP85}}(Y), 0.91\,\text{g cm}^{-3}\}, 0.1\,\text{g cm}^{-3}\}, \tag{55}$$

where $Y := (-r_k v_{\text{imp}}/T_j^{\text{sfc}})/(\mu\text{m ms}^{-1}/{}^\circ\text{C})$, $v_{\text{imp}}$ is impact velocity, and $T_j^{\text{sfc}}$ is the surface temperature of ice particle $j$,

$$\rho_{\text{rime}}^{\text{HP85}}(Y) = \begin{cases} (\text{g cm}^{-3}) \exp\left(B_2 + B_3 Y + B_4 Y^2 + B_5 Y^3\right), \\ \qquad \text{for } T_j^{\text{sfc}} > -5^\circ\text{C} \wedge Y > 1.6; \tag{56} \\ AY^{B_1}, \quad \text{otherwise}, \tag{57} \end{cases}$$

$A = 0.30\,\text{g cm}^{-3}$, $B_1 = 0.44$, $B_2 = -0.03115$, $B_3 = -1.7030$, $B_4 = 0.9116$, and $B_5 = -0.1224$.

Impact velocity can be calculated using the formula of Rasmussen and Heymsfield (1985): $v_{\text{imp}} = |v_j^\infty - v_k^\infty| \max\{f_{\text{RH85}}(N_{\text{Re}j}^{\text{i}}, N_{\text{St}}^{\text{w/i}}), 0\}$, where $N_{\text{St}}^{\text{w/i}} = (p^{\text{w/i}})^2 \rho^{\text{w}} N_{\text{Re}j}^{\text{i}}/(9\rho)$ is the Stokes impaction parameter when an ice particle collects a droplet. Because the $f_{\text{RH85}}$ given in Rasmussen and Heymsfield (1985) becomes slightly negative around $0.1 < N_{\text{St}}^{\text{w/i}} < 1.0$, we impose a limiter to ensure it is positive. Surface temperature $T_j^{\text{sfc}}$ can be evaluated as

$$T_j^{\text{sfc}} = T_j + \frac{L_s D_v}{k} \Delta\rho_j, \tag{58}$$

where $\Delta\rho_j$ is given in Eq. (21). This equation is derived under an assumption of quasi-steady vapor and thermal diffusion.

When riming occurs, the frozen droplet releases the latent heat of fusion to the moist air as described in Eqs. (74), (79) and (80).

As we will discuss in Sec. 9.1.1, the rime density formula of Heymsfield and Pflaum (1985) must be revised slightly. We propose to replace the $Y$ in Eq. (56) (not in Eq. (57)) with $Y^\downarrow = \min(Y, 3.5)$ (Eq. (107)), because the rime density derived from Eq. (56) becomes too small for larger values of $Y$, which affects the shape of hailstones near the freezing level.

Another issue discussed in Sec. 9.1.2 is related to the filling-in model. Assuming that the diameter of the frozen droplet is preserved, if the diameter is larger than the ice particle's maximum dimension, we propose replacing Eq. (50) by Eq. (108) and Eq. (54) by Eq. (109).

We validate these two corrections in Sec. 9.1.5. More discussions to refine our riming model will be presented in Sec. 9.3.7.

## 4.1.11  Aggregation between two ice particles

Finally, we consider the aggregation of ice particles. Following Connolly et al. (2012), we use the projected area of particles to evaluate the geometric cross-sectional area. The collision-aggregation kernel is then given by

$$K_{\text{agg}} = E_{\text{agg}} \left( A_j^{\frac{1}{2}} + A_k^{\frac{1}{2}} \right)^2 |v_j^\infty - v_k^\infty|, \tag{59}$$

where $E_{\text{agg}}$ is the collision-aggregation collection efficiency. Following Morrison and Grabowski (2010), we assume that the efficiency is given by a constant, $E_{\text{agg}} = 0.1$, in this study. Field et al. (2006) confirmed that $E_{\text{agg}} = 0.09$ produces a good agreement with aircraft observations.

If aggregation takes place, ice particles $j$ and $k$ merge into a single ice particle. Thus, we keep $j$ and remove $k$ from the system. However, no reliable model exists for calculating the next porous spheroid. Chen and Lamb (1994b) proposed a model, but it tends to create snow aggregates with impossibly low apparent densities (lighter than vapor). In this study, we propose another intuitive model by incorporating the compaction of fluffy snowflakes to cope with the problem.

Snow aggregates have complicated fractal structures. However, if we circumscribe them using a spheroid, the growth by aggregation is in three dimensions, rather than one (columnar) or two (planar). Therefore, as in the case of riming, we assume that only the minor dimension grows by aggregation.

If the volume weighted average density $\bar{\rho}_{jk}^{\text{i}} = (m_j + m_k)/(V_j + V_k)$ is closer to the true density of ice $\rho_{\text{true}}^{\text{i}}$, the two particles aggregate without changing their shapes. Hence, when we approximate the resultant aggregate with a spheroid, there are more empty spaces inside, thus reducing the apparent density. Let us denote the minimum possible apparent density as $\rho_{jk}^{\text{i,min}}$, which can be evaluated using Eq. (61), which we will derive shortly.

In contrast, if $\bar{\rho}_{jk}^{\text{i}}$ is small, compaction of the fluffy snowflakes occurs, and the empty space of the larger ice particle could be filled with the smaller ice particle or the particles might deform because of the collision-aggregation impact. Because of this compaction mechanism, we assume there is a limiting value of the apparent density, and let it be $\rho_{\text{crt}}^{\text{i}} = 10\,\text{kg m}^{-3}$. This choice of value is roughly consistent with observations by Magono and Nakamura (1965). If $\bar{\rho}_{jk}^{\text{i}}$ is closer to $\rho_{\text{crt}}^{\text{i}}$, we consider that the apparent density of the resultant aggregate is closer to the maximum possible density $\rho_{jk}^{\text{i,max}}$. Let us assume $\rho_{jk}^{\text{i,max}} = \bar{\rho}_{jk}^{\text{i}}$.

In the following, we derive equations describing how to update the attributes.

Without loss of generality, assume that $D_j \geq D_k$. For $\phi_j \leq 1$ (planar),

$$a_j' = a_j, \tag{60}$$

because we assumed the maximum dimension is preserved. The longest possible minor axis length is $c_j + \min(a_k, c_k)$, hence the largest possible volume becomes $V_{\text{max}} = (4\pi/3)a_j^2\{c_j + \min(a_k, c_k)\}$. The minimum possible apparent density $\rho_{jk}^{\text{i,min}}$ then

becomes

$$\rho_{jk}^{i,\min} = \frac{m_j + m_k}{V_{\max}}. \tag{61}$$

The resultant particle's apparent density is given by a weighted average of $\rho_{jk}^{i,\max} = \bar{\rho}_{jk}^{i}$ and $\rho_{jk}^{i,\min}$:

$$\rho_j^{i\prime} = \frac{(\rho_{\text{true}}^i - \bar{\rho}_{jk}^i)\rho_{jk}^{i,\max} + (\bar{\rho}_{jk}^i - \rho_{\text{crt}}^i)\rho_{jk}^{i,\min}}{\rho_{\text{true}}^i - \rho_{\text{crt}}^i}, \tag{62}$$

where primed values indicate the resultant ice particle. All other attributes are updated as follows:

$$c_j' = \frac{m_j + m_k}{\rho_j^{i\prime}(4\pi/3)a_j'^2}, \tag{63}$$

$$m_j^{\text{rime}\prime} = m_j^{\text{rime}} + m_k^{\text{rime}}, \tag{64}$$

$$n_j^{\text{mono}\prime} = n_j^{\text{mono}} + n_k^{\text{mono}}, \tag{65}$$

$$m_{\alpha j}^{\text{sol}\prime} = m_{\alpha j}^{\text{sol}} + m_{\alpha k}^{\text{sol}}, \quad \alpha = 1, 2, \ldots, N^{\text{sol}}, \tag{66}$$

$$m_{\beta j}^{\text{insol}\prime} = m_{\beta j}^{\text{insol}} + m_{\beta k}^{\text{insol}}, \quad \beta = 1, 2, \ldots, N^{\text{insol}}, \tag{67}$$

$$T_j^{\text{fz}\prime} = \max(T_j^{\text{fz}}, T_k^{\text{fz}}). \tag{68}$$

For $\phi_j > 1$ (columnar), the polar axis length is preserved

$$c_j' = c_j. \tag{69}$$

If approximating the largest possible particle using an ellipsoid, the largest possible volume becomes $V_{\max} = (4\pi/3)c_j\{a_j + \min(a_k, c_k)\}\max(a_j, a_k, c_k)$. Then, the resultant ice particle's apparent density $\rho_j^{i\prime}$ can be calculated using Eqs. (61) and (62). Then, the minor axis is updated by

$$a_j' = \left\{ \frac{m_j + m_k}{\rho_j^{i\prime}(4\pi/3)c_j'} \right\}^{\frac{1}{2}}, \tag{70}$$

and other attributes are updated by Eqs. (64)-(68).

Note that our aggregation outcome model does not produce particles lighter than $\rho_{\text{crt}}^i = 10\,\text{kg}\,\text{m}^{-3}$.

## 4.1.12 Limitations of our cloud microphysics model

Eqs. (2)–(70) provide time evolution equations for mixed-phase cloud microphysics. Our model is based on a detailed kinetic description, and all aerosol, cloud, and precipitation particles in the system are followed. The respective activation and deactivation of cloud droplets from and to CCNs, and their growth by diffusion and collision are also explicitly predicted. Additionally, the formation of ice particles by condensation/immersion and homogeneous freezing, and gradual morphology changes in ice particles during their growth by diffusion and collision are also predicted explicitly without relying on artificial ice categories or predefined mass-dimension relationships. However, because our basic understanding of mixed-phase cloud microphysics is

still insufficient, the introduced models have room for improvement. Further, several processes critical for mixed-phase clouds are ignored for simplicity. For example, collisional breakup of ice particles and rime-splintering are not considered, although they are thought to be responsible for secondary ice production (e.g., Field et al., 2017). In Sec. 9.3, we will discuss more on the limitations and possible future refinements of our model.

## 4.2 Fluid dynamics of moist air

Moist air fluid dynamics can be described by the compressible Navier–Stokes equation for moist air:

$$\frac{\partial \rho}{\partial t} + \nabla \cdot (\rho \boldsymbol{U}) = \left. \frac{\partial \rho}{\partial t} \right|_{\mathrm{cm}}, \tag{71}$$

$$\frac{\partial \rho q_{\mathrm{v}}}{\partial t} + \nabla \cdot (\rho q_{\mathrm{v}} \boldsymbol{U}) = \left. \frac{\partial \rho q_{\mathrm{v}}}{\partial t} \right|_{\mathrm{cm}} + D_{\mathrm{v}} \nabla^2 (\rho q_{\mathrm{v}}), \tag{72}$$

$$\frac{\partial \rho \boldsymbol{U}}{\partial t} + \nabla \cdot (\rho \boldsymbol{U} \otimes \boldsymbol{U}) = -\nabla P - \rho g \hat{\boldsymbol{z}} + \left. \frac{\partial \rho \boldsymbol{U}}{\partial t} \right|_{\mathrm{cm}} + \mu \nabla^2 \boldsymbol{U}, \tag{73}$$

$$\frac{\partial \rho \theta}{\partial t} + \nabla \cdot (\rho \theta \boldsymbol{U}) = \left. \frac{\partial \rho \theta}{\partial t} \right|_{\mathrm{cm}} + \frac{k}{c_{\mathrm{p}}} \nabla^2 \theta, \tag{74}$$

$$P = \rho R T = P_0 \left( \frac{\rho \theta R}{P_0} \right)^{c_{\mathrm{p}}/(c_{\mathrm{p}} - R)}, \tag{75}$$

where the four terms with the form $\partial \cdot / \partial t|_{\mathrm{cm}}$ represent cloud microphysics coupling terms.

$\partial \rho / \partial t|_{\mathrm{cm}} = \partial \rho q_{\mathrm{v}} / \partial t|_{\mathrm{cm}}$ is the source of vapor:

$$\left. \frac{\partial \rho}{\partial t} \right|_{\mathrm{cm}} = \left. \frac{\partial \rho q_{\mathrm{v}}}{\partial t} \right|_{\mathrm{cm}} = s_{\mathrm{v}} + s_{\mathrm{s}}. \tag{76}$$

Here, $s_{\mathrm{v}}$ and $s_{\mathrm{s}}$ are sources of vapor through condensation/evaporation and deposition/sublimation, respectively:

$$s_{\mathrm{v}}(\boldsymbol{x}, t) = - \sum_{i \in I_{\mathrm{r}}(t)} \delta^3(\boldsymbol{x} - \boldsymbol{x}_i(t)) \left. \frac{dm_i}{dt} \right|_{\mathrm{cnd/evp}}, \tag{77}$$

$$s_{\mathrm{s}}(\boldsymbol{x}, t) = - \sum_{i \in I_{\mathrm{r}}(t)} \delta^3(\boldsymbol{x} - \boldsymbol{x}_i(t)) \left. \frac{dm_i}{dt} \right|_{\mathrm{dep/sbl}}, \tag{78}$$

where $\delta^3(\boldsymbol{x})$ is the three-dimensional Dirac's delta function, and the time derivatives for condensation/evaporation and deposition/sublimation are given by Eqs. (7) and (11), respectively.

$\partial \rho \theta / \partial t|_{\mathrm{cm}}$ represents heating due to the phase transition of water:

$$\left. \frac{\partial \rho \theta}{\partial t} \right|_{\mathrm{cm}} = - \frac{L_{\mathrm{v}} s_{\mathrm{v}} + L_{\mathrm{s}} s_{\mathrm{s}} + L_{\mathrm{f}} s_{\mathrm{f}}}{c_p \Pi}, \tag{79}$$

where $L_{\mathrm{f}}$ is the latent heat of fusion, and $s_{\mathrm{f}}$ is the production rate of liquid water through freezing, melting, or riming. Let $t_n^{\mathrm{fz}}$ be the time of the $n$-th freezing event and $i_n^{\mathrm{fz}}$ be the index of the frozen droplet. Similarly, let $t_n^{\mathrm{mlt}}$ and $i_n^{\mathrm{mlt}}$ be the time and melted ice particle of the $n$-th melting event, respectively. Let $t_n^{\mathrm{rime}}$ and $i_n^{\mathrm{rime}}$ be the time and rimed droplet of the $n$-th riming

event, respectively. Then, $s_f$ is given by

$$s_f(\boldsymbol{x}, t) = -\sum_{\substack{\text{freezing event } n}} \delta^3(\boldsymbol{x} - \boldsymbol{x}_{i_n^{fz}}(t))\delta(t - t_n^{fz})m_{i_n^{fz}}(t)$$

$$+ \sum_{\substack{\text{melting event } n}} \delta^3(\boldsymbol{x} - \boldsymbol{x}_{i_n^{mlt}}(t))\delta(t - t_n^{mlt})m_{i_n^{mlt}}(t)$$

$$- \sum_{\substack{\text{riming event } n}} \delta^3(\boldsymbol{x} - \boldsymbol{x}_{i_n^{rime}}(t))\delta(t - t_n^{rime})m_{i_n^{rime}}(t). \tag{80}$$

$\partial \rho \boldsymbol{U}/\partial t|_{cm}$ is the drag force from the particles. From Eq. (2), we can derive $\boldsymbol{F}_i^{drg} = m_i g\hat{\boldsymbol{z}} + d(m_i \boldsymbol{v}_i)/dt$. The terminal velocity assumption does not mean that the second term vanishes because $m_i$ and $\boldsymbol{v}_i$ are still time dependent. However, even if a droplet accelerated from $0 \text{ m s}^{-1}$ to $10 \text{ m s}^{-1}$ in $100$ s through rapid precipitation development, the contribution of the second term is much smaller than that of the first term: $10 \text{ m s}^{-1}/100 \text{ s} \ll g$. Thus, we finally obtain

$$\left.\frac{\partial \rho \boldsymbol{U}}{\partial t}\right|_{cm} = -\sum_{i \in I_r(t)} \delta^3(\boldsymbol{x} - \boldsymbol{x}_i(t))\boldsymbol{F}_i^{drg}$$

$$\approx -\left[\sum_{i \in I_r(t)} \delta^3(\boldsymbol{x} - \boldsymbol{x}_i(t))m_i(t)\right] g\hat{\boldsymbol{z}}. \tag{81}$$

### 4.3   Summary of the section

Now, we have the complete set of the system's time evolution equations: Eqs. (2)–(70) for cloud microphysics (i.e., aerosol, cloud, and precipitation particles) and Eqs. (71)–(81) for cloud dynamics (i.e., moist air). With suitable initial and boundary conditions, our mathematical model can predict mixed-phase cloud behavior. In the next section, we explain how SCALE-SDM solves those time evolution equations numerically.

### 5   Numerical schemes and implementation

We develop a numerical model known as SCALE-SDM to solve the mathematical model of mixed-phase clouds presented in the preceding sections.

SCALE is a library of weather and climate models of the Earth and other planets (Nishizawa et al., 2015; Sato et al., 2015, http://r-ccs-climate.riken.jp/scale/). We implemented SDM into SCALE version 0.2.5, thus constructing a mixed-phase cloud model called SCALE-SDM 0.2.5-2.2.0.

In our model, we use SDM to solve cloud microphysics as defined by Eqs. (2)–(70). SDM is a particle-based scheme using an efficient Monte Carlo algorithm for coalescence, riming, and aggregation, which enables the accurate simulation of aerosol, cloud, and precipitation particles with lower computational demand (Shima et al., 2009).

Moist air fluid dynamics are solved using SCALE's dynamical core. We solve the compressible Navier–Stokes equation for moist air (71)–(81) using a forward temporal integration scheme using a finite volume method with an Arakawa-C staggered grid. In this study, we resolve only large eddies and do not use a sub-grid scale (SGS) turbulence model. To stabilize the

calculation, we add an artificial fourth-order hyperdiffusion term. Numerical schemes and implementation are described in further detail.

## 5.1 Spatial discretization of moist air

We consider the density of moist air $\rho$, density of water vapor $\rho q_\mathrm{v}$, momentum of moist air $\rho \boldsymbol{U}$, and mass-weighted potential temperature $\rho \theta$ as prognostic variables for moist air. We employ the Arakawa-C staggered grid for discretization: $\rho$, $\rho q_\mathrm{v}$, and $\rho \theta$ are defined at the center of each grid cell, and the three components of $\rho \boldsymbol{U}$ are defined on the faces of each grid cell. To simplify the notation, we use $\boldsymbol{G}_{lmn}$ to denote the status of moist air at each point on the center grid and the face grid. Let $\Delta x$, $\Delta y$, and $\Delta z$ represent the grid sizes.

## 5.2 Super-particles and real particles

There are many particles in the atmosphere, thus it is practically impossible to follow all of them in a numerical model. However, it is reasonable to assume that only the collective properties of the particle population are relevant to predict the behavior of clouds, because clouds are insensitive to each individual particle. Therefore, let us approximate the population of real particles $\{\{\boldsymbol{x}_i(t), \boldsymbol{a}_i(t)\}, i = 1, 2, \ldots, N_\mathrm{r}^\mathrm{wp}\}$ by a population of super-particles: $\{\{\xi_i(t), \boldsymbol{x}_i(t), \boldsymbol{a}_i(t)\}, i = 1, 2, \ldots, N_\mathrm{s}^\mathrm{wp}\}$ (see, e.g., Fig. 4 of Grabowski et al., 2019). A super-particle is characterized by multiplicity $\xi_i$, position $\boldsymbol{x}_i$, and attributes $\boldsymbol{a}_i$. We consider that the $i$-th super-particle represents $\xi_i$ real particles $\{\boldsymbol{x}_i, \boldsymbol{a}_i\}$. Note that multiplicity $\xi_i$ is an integer and is time dependent. $N_\mathrm{s}^\mathrm{wp}$ is the total number of super-particles accumulated over the whole period.

The relationship between super-particles and real particles can be expressed more precisely as follows. Let $n(\boldsymbol{a}, \boldsymbol{x}, t)$ be the particle distribution function, i.e., the mean number density of particles with attributes $\boldsymbol{a}$ at position $\boldsymbol{x}$ and time $t$. The following relation then holds:

$$n(\boldsymbol{a}, \boldsymbol{x}, t) = \left\langle \sum_{i \in I_\mathrm{r}(t)} \delta^d(\boldsymbol{a} - \boldsymbol{a}_i(t)) \delta^3(\boldsymbol{x} - \boldsymbol{x}_i(t)) \right\rangle, \tag{82}$$

where $\langle \cdots \rangle$ denotes the mean, and $\delta^d(\boldsymbol{a})$ is the $d$-dimensional Dirac's delta function. Super-particles reproduce the behavior of particles in expectation:

$$\begin{aligned} n(\boldsymbol{a}, \boldsymbol{x}, t) &= \left\langle \sum_{i \in I_\mathrm{s}(t)} \xi_i(t) \delta^d(\boldsymbol{a} - \boldsymbol{a}_i(t)) \delta^3(\boldsymbol{x} - \boldsymbol{x}_i(t)) \right\rangle \\ &= N_\mathrm{s}(t) \sum_{\xi=1}^\infty \xi p(\xi, \boldsymbol{a}, \boldsymbol{x}, t), \end{aligned} \tag{83}$$

where $p(\xi, \boldsymbol{a}, \boldsymbol{x}, t)$ is the probability density that a super-particle has multiplicity $\xi$, attributes $\boldsymbol{a}$, and position $\boldsymbol{x}$ at time $t$; $I_\mathrm{s}(t)$ is the set of super-particle indices existing in the domain at time $t$; and $N_\mathrm{s}(t) := \#I_\mathrm{s}(t)$ is the number of super-particles existing at time $t$.

## 5.3 Initialization of super-particles

There is an arbitrariness in how to initialize super-particles. In this study, we use the uniform sampling method.

Any probability density function $p(\xi, \boldsymbol{a}, \boldsymbol{x}, t = 0)$ that satisfies Eq. (83) can be used to initialize super-particles; however, Unterstrasser et al. (2017) showed that SDM's performance is sensitive to the choice of the probability density function.

Let us consider a specific type of procedure wherein we assign $\boldsymbol{a}$ and $\boldsymbol{x}$ based on the probability density function $p(\boldsymbol{a}, \boldsymbol{x})$, and determine the super-particle's multiplicity $\xi$ by using a deterministic function of $\boldsymbol{a}$ and $\boldsymbol{x}$, i.e., $\xi = \xi(\boldsymbol{a}, \boldsymbol{x})$. Then, Eq. (83) at $t = 0$ reduces to

$$n(\boldsymbol{a}, \boldsymbol{x}, 0) = N_{\mathrm{s}}(0)\xi(\boldsymbol{a}, \boldsymbol{x})p(\boldsymbol{a}, \boldsymbol{x}). \tag{84}$$

If we set $\xi(\boldsymbol{a}, \boldsymbol{x})$ as a constant, the probability density function must be proportional to the initial distribution function of real particles: $p(\boldsymbol{a}, \boldsymbol{x}) \propto n(\boldsymbol{a}, \boldsymbol{x}, 0)$. This so-called constant multiplicity method was adopted in Shima et al. (2009). However, Unterstrasser et al. (2017) found that the numerical convergence of this method regarding the super-particle number is slow. Note that constant multiplicity method is referred to as $\nu_{\mathrm{const}}$-init in Unterstrasser et al. (2017).

Instead, we can set $p(\boldsymbol{a}, \boldsymbol{x})$ as a constant (i.e., uniform sampling). Multiplicity then becomes proportional to the initial distribution function of real particles:

$$\xi(\boldsymbol{a}, \boldsymbol{x}) = \frac{n(\boldsymbol{a}, \boldsymbol{x}, 0)}{N_{\mathrm{s}}(0)p}, \quad p(\boldsymbol{a}, \boldsymbol{x}) = p = \mathrm{const}. \tag{85}$$

Using the uniform sampling method, we can more frequently sample rare but important particles in the tail of the distribution, thus improving the numerical convergence. This uniform sampling method was used in various studies (e.g., Arabas and Shima, 2013; Shima et al., 2014; Sato et al., 2017, 2018).

Unterstrasser et al. (2017) proposed several other procedures using a grid, known as SingleSIP-init, multiSIP-init, and $\nu_{\mathrm{random}}$-init to more uniformly distribute super-particles along the particle size axis. They confirmed that their methods had much better performance than the constant multiplicity method but did not try the uniform sampling method. Dziekan and Pawlowska (2017) also proposed a similar procedure. However, both works focused on coalescence and their initialization procedures are tested only in a zero-dimensional simulation (box model) with one particle attribute (size). It is questionable whether their procedures would work efficiently for three-dimensional (3D) simulations with several particle attributes. The "discrepancy" of axis-aligned grid decreases slowly in higher dimensions (e.g., Niederreiter, 1978). Therefore, an axis-aligned grid is generally unsuitable for sampling high dimensional spaces. A uniform sampling method should be more efficient for such a purpose and using quasi-random numbers would further improve performance. Meanwhile, as indicated in Grabowski et al. (2018), we should also note that the unbalanced mass of super-particles could cause larger statistical fluctuations when super-particles are advected from one grid cell of moist air to another.

Overall, further investigation is required to determine an optimal method for initializing super-particles. In this study, we use the uniform sampling method given by Eq. (85). More details of our procedure will be specified in Sec. 6.1.7. As shown in Fig. 9, our model's numerical convergence regarding super-particle numbers is good for at least the 2D cumulonimbus simulation that we will conduct to evaluate our model.

| | t ————→ t+Δt | | | | | |
|---|---|---|---|---|---|---|
| fluid dynamics | 1 | 2 | 3 | 4 | 5 | 6 |
| advection | 7 | | | | | |
| freezing/melting | 8 | | | | | |
| condensation/evaporation | 9 | 12 | 13 | 15 | 17 | 19 |
| deposition/sublimation | 10 | | 14 | | 18 | |
| collision-coalescence/ riming/aggregation | 11 | | | 16 | | |

**Table 1.** An example of the calculation order when updating the system state from $t$ to $t + \Delta t$. We first calculate the fluid dynamics and then calculate cloud microphysics. Each process is integrated one time step forward at a time. Processes lagging in time are calculated preferentially.

## 5.4 Operator splitting of the time integration

We separately evaluate each process using the first-order operator splitting scheme. Let $\Delta t$ be the common time step. Here, we explain how $\{\{\xi_i, \boldsymbol{x}_i, \boldsymbol{a}_i\}\}$ and $\boldsymbol{G}_{lmn}$ are updated from time $t$ to $t + \Delta t$.

Let $\Delta t_{\mathrm{adv}}$, $\Delta t_{\mathrm{fz/mlt}}$, $\Delta t_{\mathrm{cnd/evp}}$, $\Delta t_{\mathrm{dep/sbl}}$, and $\Delta t_{\mathrm{collis}}$ be the time steps for the advection and sedimentation of particles,

freezing and melting, condensation and evaporation, deposition and sublimation, and collision-coalescence, -riming, and -aggregation, respectively.

Let $\Delta t_{\mathrm{dyn}}$ be the time step for moist air fluid dynamics.

These process time steps are all divisors of the common time step $\Delta t$.

We first calculate fluid dynamics without the coupling terms from particles to moist air (76)–(81), and update moist air

from $\boldsymbol{G}_{lmn}(t)$ to $\boldsymbol{G}'_{lmn}(t)$. Then, we update super-particles $\{\{\xi_i, \boldsymbol{x}_i, \boldsymbol{a}_i\}\}$ from $t$ to $t + \Delta t$. We select one elementary cloud microphysics process, integrate it forward by one time step, and then move on to the next process. Here, processes lagging in time are calculated preferentially. Simultaneously, we evaluate feedback from the particles to moist air through the coupling terms (76)–(81), and update the moist air from $\boldsymbol{G}'_{lmn}(t)$ to $\boldsymbol{G}_{lmn}(t + \Delta t)$. Table 1 shows an example of the calculation order.

## 5.5 Time integration of cloud microphysics

We use SDM to solve cloud microphysics. We provide details of the numerical schemes used to calculate cloud microphysics in this section. The state of ambient air $\boldsymbol{G}_i := \boldsymbol{G}(\boldsymbol{x}_i)$ around a super-particle $i$ is often needed. For scalar variables, we use the value at the center point of the grid cell in which the super-particle is located, whereas we interpolate wind velocities from face grids, as detailed in the next section.

### 5.5.1 Advection and sedimentation

For each super-particle, the motion equation (3) is solved using a time step $\Delta t_{\mathrm{adv}}$. We normally select a short enough $\Delta t_{\mathrm{adv}}$ to satisfy the Courant-Friedrichs-Lewy (CFL) condition for wind velocity. So that we can predict the particle number concentra-

tion accurately, we use the predictor-corrector scheme with the "simple linear interpolation" of wind velocities from the face grid following Grabowski et al. (2018). The momentum $\rho U$ is defined on the face grid and density $\rho$ is defined on the center grid. Therefore, we average the $\rho_{lmn}$ on both sides of the face grid to calculate wind velocity $U_{lmn}$ on the face grid. We then interpolate $U_{lmn}$ to the super-particle position using the simple linear scheme of Grabowski et al. (2018), which ensures that the wind velocity divergence over any subgrid volume becomes exactly the same as that over the grid cell volume.

The reaction force acting on moist air is calculated using Eq. (81). Feedback from each super-particle is imposed only on the $(\rho W)_{lmn}$ nearest to the super-particle.

### 5.5.2 Freezing and melting

Every $\Delta t_{\text{fz/mlt}}$ interval, for each super-particle, freezing and melting is examined following the model detailed in Secs. 4.1.4 and 4.1.5. The exchange of latent heat of fusion is calculated using Eqs. (74), (79), and (80). Feedback from each super-particle is imposed only on the grid cell where the super-particle is located.

### 5.5.3 Condensation and evaporation

For each super-droplet, we solve the condensation and evaporation equation (8) with a time step of $\Delta t_{\text{cnd/evp}}$. The activation/deactivation time scale is much shorter than that of other processes. To eliminate stiffness, we convert the equation to the time evolution equation of $r^2$ following Shima et al. (2009) and adopt the backward Euler scheme.

The exchange of vapor and latent heat with moist air is calculated using Eqs. (71), (72), (74), (76), (77), and (79). Feedback from each super-droplet is imposed only on the grid cell where the super-droplet is located.

The growth of droplets is calculated implicitly; however, the evolution of supersaturation through feedback is calculated explicitly. Therefore, the length of $\Delta t_{\text{cnd/evp}}$ is restricted mostly by supersaturation's phase relaxation time regarding condensation and evaporation, which is the timescale in which a supersaturation fluctuation decays through condensation or evaporation.

### 5.5.4 Deposition and sublimation

For each ice super-particle, we solve the deposition and sublimation time evolution equations detailed in Sec. 4.1.7 using the time step $\Delta t_{\text{dep/sbl}}$. Contrary to the condensation and evaporation equation (8), the time evolution equation of mass (11) is not stiff because the curvature term is ignored and the solute effect does not exist. Let us convert the equation to the time evolution equation of $m^{2/3}$. Then, in a situation when the ice particle is spherical and, at the same time, so small that the ventilation effect can be ignored, then the equation reduces to $dm^{2/3}/dt = \text{const.}$, i.e., the r.h.s. does not depend on $m$. Inspired by this fact, we adopt the forward Euler scheme to solve the time evolution equation of $m^{2/3}$ even when the ice particle is not spherical or small.

The exchange of vapor and latent heat with moist air is calculated using Eqs. (71), (72), (74), (76), (78), and (79). Feedback from each ice super-particle is imposed only on the grid cell where the ice super-particle is located.

$\Delta t_{\mathrm{dep/sbl}}$ is restricted by the timescale of individual ice particle growth through deposition and sublimation, and the phase relaxation time of supersaturation regarding deposition and sublimation.

### 5.5.5 Coalescence, riming, and aggregation

The stochastic process of coalescence, riming, and aggregation detailed in Secs. 4.1.8–4.1.11 is solved using the Monte Carlo algorithm of SDM (Shima et al., 2009). The computational cost of this algorithm is proportional to the number of super-particles $O(N_{\mathrm{s}})$, which is achieved by an efficient collision candidate pair number reduction technique. An additional advantage of this technique is the parallelizability of computation; each super-particle belongs to only one candidate pair, and hence, dependencies are eliminated.

$\Delta t_{\mathrm{collis}}$ can be determined using the argument presented in the last paragraph of Sec. 5.1.3 in Shima et al. (2009); however, here we repeat it in a slightly different way to provide a precise physical interpretation. In short, the time step $\Delta t_{\mathrm{collis}}$ is restricted by the mean free time of a particle, i.e., the average waiting time for a particle between two successive coalescence/riming/aggregation events. Let $\overline{P}$ be the typical probability that a particle coalescence/rime/aggregate with another particle within a small time interval $\Delta t_{\mathrm{collis}}$. From Eq. (31), $\overline{P}$ can be evaluated as

$$\overline{P} \approx N_{\mathrm{r}}' \overline{K} \frac{\Delta t_{\mathrm{collis}}}{\Delta V} \approx n_{\mathrm{r}} \overline{K} \Delta t_{\mathrm{collis}}, \tag{86}$$

where $N_{\mathrm{r}}'$ is the number of real particles in a volume $\Delta V$, $\overline{K}$ is the typical value of the coalescence/riming/aggregation kernel $K$, and $n_{\mathrm{r}}$ is the number concentration of real particles. Requiring that $\overline{P} < 1$ has to be satisfied, we obtain

$$\Delta t_{\mathrm{collis}} < 1/(n_{\mathrm{r}} \overline{K}). \tag{87}$$

Here, we relate the above argument to that of Shima et al. (2009). Let $\overline{P_{\mathrm{s}}}$ be the typical probability that a collision candidate super-particle pair coalescence/rime/aggregate after the pair number reduction technique is applied. Note that $\overline{P_{\mathrm{s}}}$ is what Shima et al. (2009) evaluated in the last paragraph of Sec. 5.1.3. We can derive $\overline{P_{\mathrm{s}}} \approx \overline{P}$ as follows:

$$\begin{aligned}
\overline{P_{\mathrm{s}}} &\approx \left\{ \frac{N_{\mathrm{s}}'(N_{\mathrm{s}}'-1)}{2} \middle/ \left[ \frac{N_{\mathrm{s}}'}{2} \right] \right\} \overline{\xi}\, \overline{K} \frac{\Delta t_{\mathrm{collis}}}{\Delta V} \\
&\approx N_{\mathrm{s}}' \frac{N_{\mathrm{r}}'}{N_{\mathrm{s}}'} \overline{K} \frac{\Delta t_{\mathrm{collis}}}{\Delta V} \\
&\approx \overline{P},
\end{aligned} \tag{88}$$

where $N_{\mathrm{s}}'$ is the number of super-particles in the volume $\Delta V$, the first term $\{\dots\}$ represents the scale-up factor due to the candidate pair number reduction, and $\overline{\xi} \approx N_{\mathrm{r}}'/N_{\mathrm{s}}'$ is the typical multiplicity.

In SDM, the multiple coalescence technique is used to make the algorithm robust to larger $\Delta t_{\mathrm{collis}}$. Here, we clarify how we adapt it to riming and aggregation. If it is a coalescence between a droplet $j$ and $\widetilde{\gamma}$ number of droplets $k$ (see Sec. 5.1.3 of Shima et al. (2009) for the definition of $\widetilde{\gamma}$), we modify Eqs. (33)–(35) by applying

$$(r_k^3, m_{\alpha k}^{\mathrm{sol}}, m_{\beta k}^{\mathrm{insol}}) \to \widetilde{\gamma}(r_k^3, m_{\alpha k}^{\mathrm{sol}}, m_{\beta k}^{\mathrm{insol}}). \tag{89}$$

If it is a coalescence between $\widetilde{\gamma}$ number of droplets $j$ and a droplet $k$, we apply

$$(r_j^3, m_{\alpha j}^{\text{sol}}, m_{\beta j}^{\text{insol}}) \rightarrow \widetilde{\gamma}(r_j^3, m_{\alpha j}^{\text{sol}}, m_{\beta j}^{\text{insol}}). \tag{90}$$

Similarly, if it is a riming/aggregation between a particle $j$ and $\widetilde{\gamma}$ number of particles $k$, we apply the following replacement to Eqs. (42)–(54) and (60)–(70):

$$(m_k, V_k, m_k^{\text{rime}}, n_k^{\text{mono}}, m_{\alpha k}^{\text{sol}}, m_{\beta k}^{\text{insol}})$$
$$\rightarrow \widetilde{\gamma}(m_k, V_k, m_k^{\text{rime}}, n_k^{\text{mono}}, m_{\alpha k}^{\text{sol}}, m_{\beta k}^{\text{insol}}). \tag{91}$$

If it is a riming/aggregation between $\widetilde{\gamma}$ number of particles $j$ and a particle $k$,

$$(m_j, V_j, m_j^{\text{rime}}, n_j^{\text{mono}}, m_{\alpha j}^{\text{sol}}, m_{\beta j}^{\text{insol}})$$
$$\rightarrow \widetilde{\gamma}(m_j, V_j, m_j^{\text{rime}}, n_j^{\text{mono}}, m_{\alpha j}^{\text{sol}}, m_{\beta j}^{\text{insol}}). \tag{92}$$

What is not straightforward is the calculation of $V_{\max}$ used in the aggregation outcome formula. For planar collector $j$, we consider that $V_{\max}$ is given by

$$V_{\max} = \begin{cases} (4\pi/3)a_j^2\{c_j + \widetilde{\gamma}\min(a_k, c_k)\}, \\ (4\pi/3)a_j^2\{\widetilde{\gamma}c_j + \min(a_k, c_k)\}. \end{cases} \tag{93}$$

For columnar collector $j$,

$$V_{\max} = \begin{cases} (4\pi/3)c_j\{a_j + \widetilde{\gamma}\min(a_k, c_k)\}\max(a_j, a_k, c_k), \\ (4\pi/3)c_j\{\widetilde{\gamma}a_j + \min(a_k, c_k)\}\max(a_j, a_k, c_k). \end{cases} \tag{94}$$

The exchange of the latent heat of fusion due to riming is calculated using Eqs. (74), (79), and (80). Feedback from each super-particle is imposed only on the grid cell where the super-particle is located.

## 5.6 Time integration of moist air fluid dynamics

Moist air fluid dynamics is governed by the compressible Navier–Stokes equation (71)–(81). In this study, as explained in the previous section, the four coupling terms from cloud microphysics denoted by $\partial \cdot /\partial t|_{\text{cm}}$ are evaluated when calculating cloud microphysics.

We solve the compressible Navier–Stokes equation without the coupling terms using a finite volume method with an Arakawa-C staggered grid. For spatial discretization, the fourth-order central difference scheme is used for advection terms and the second-order central difference scheme is used for other spatial derivatives. To preserve the monotonicity, we apply the flux-corrected transport scheme of Zalesak (1979) to water vapor advection. For time integration, we use the three-step Runge–Kutta scheme of Wicker and Skamarock (2002). An artificial, fourth-order hyper-diffusion term is added to stabilize the calculation. For this study, we set the non-dimensional diffusion coefficient defined in Eq. (A132) of Nishizawa et al. (2015)

as $\gamma = 10^{-3}$. For more details of the numerical schemes used for fluid dynamics, see Nishizawa et al. (2015) and Sato et al. (2015).

The time step $\Delta t_{\mathrm{dyn}}$ must satisfy the CFL condition of acoustic waves.

# 6 Design of numerical experiments for model evaluation: 2D simulation of an isolated cumulonimbus

The preceding sections described the basic equations and numerical implementation of SCALE-SDM. To evaluate our numerical model's performance, we conduct a 2D simulation of an isolated cumulonimbus following the setup of Khain et al. (2004). In this section, we first describe the atmospheric conditions and numerical parameters used for the control case denoted by CTRL. To evaluate fluctuation, we conduct a 10-member ensemble of simulations by changing the pseudo-random number sequence. To investigate the simulation's numerical convergence, we will change the super-particle number concentration, grid sizes, and time steps of CTRL. Those ensembles are denoted by NSP, DX, and DT, respectively. Our choice of parameters is specified in the subsequent sections. Table 2 summarizes the model setup for all cases.

## 6.1 Control ensemble (CTRL)

In this section, we specify the atmospheric conditions and numerical parameters used for the CTRL ensemble.

### 6.1.1 Initial moist air conditions

The domain is 2D ($x$-$z$), $60\,\mathrm{km}$ in the horizontal direction and $16\,\mathrm{km}$ in the vertical direction.

The initial atmospheric profile is horizontally uniform, and the vertical moist air profile is given by sounding data from Midland, Texas, on 13 August 1999, as shown in Fig. 4 of Khain et al. (2004). The cloud base and freezing level are at about $2.2\,\mathrm{km}$ ($14\,^\circ\mathrm{C}$) and $4.1\,\mathrm{km}$, respectively. We consider that the wind is initially horizontal and wind velocity increases from $4\,\mathrm{m\,s^{-1}}$ near the surface to $7\,\mathrm{m\,s^{-1}}$ at $400\,\mathrm{hPa}$, and remains unchanged at higher levels.

### 6.1.2 Moist air boundary conditions

For the lateral boundaries, we impose periodic boundary conditions. For the upper and lower boundaries, we set the vertical wind velocity $W$ to zero, i.e., a zero-fixed boundary condition for vertical momentum $\rho W$, and no flux boundary conditions for other prognostic variables.

### 6.1.3 Initial conditions of particles

Initially, the particles are distributed uniformly in space at random, and consist of pure ammonium bisulfate aerosol particles and mineral dust internally mixed with ammonium bisulfate.

**Table 2.** Summary of numerical experiments for model evaluation. The domain is two-dimensional ($x$-$z$), $60\,$km in the horizontal direction and $16\,$km in the vertical direction. The initial profile of moist air is given by sounding data from Midland, Texas, on 13 August 1999, as shown in Fig. 4 of Khain et al. (2004). The particles are initially distributed uniformly in space at random and consist of pure ammonium bisulfate aerosol particles and mineral dust internally mixed with ammonium bisulfate. The numerical parameters used in each case are listed in the table and values changed from the CTRL case are in bold. We conducted a 10-member ensemble of simulations for each case by changing the pseudo random number sequence to evaluate fluctuation.

| Case | Super-particle number concentration $c^{SP}$ [/cell] | Grid size $\Delta x = \Delta y = \Delta z$ [m] | $\Delta t$ [s] | $\Delta t_{adv}$ [s] | $\Delta t_{fz/mlt}$ [s] | $\Delta t_{collis}$ [s] | $\Delta t_{cnd/evp}$ [s] | $\Delta t_{dep/sbl}$ [s] | $\Delta t_{dyn}$ [s] |
|---|---|---|---|---|---|---|---|---|---|
| CTRL | 128 | 62.5 | 0.4 | 0.4 | 0.4 | 0.2 | 0.1 | 0.1 | 0.05 |
| NSP002 | **2** | 62.5 | 0.4 | 0.4 | 0.4 | 0.2 | 0.1 | 0.1 | 0.05 |
| NSP004 | **4** | 62.5 | 0.4 | 0.4 | 0.4 | 0.2 | 0.1 | 0.1 | 0.05 |
| $\vdots$ | $\vdots$ | $\vdots$ | $\vdots$ | $\vdots$ | $\vdots$ | $\vdots$ | $\vdots$ | $\vdots$ | $\vdots$ |
| NSP128 (CTRL) | 128 | 62.5 | 0.4 | 0.4 | 0.4 | 0.2 | 0.1 | 0.1 | 0.05 |
| NSP256 | **256** | 62.5 | 0.4 | 0.4 | 0.4 | 0.2 | 0.1 | 0.1 | 0.05 |
| NSP512 | **512** | 62.5 | 0.4 | 0.4 | 0.4 | 0.2 | 0.1 | 0.1 | 0.05 |
| DXx4 | 128 | **250.0** | **1.6** | **1.6** | **1.6** | 0.2 | 0.1 | 0.1 | **0.2** |
| DXx2 | 128 | **125.0** | **0.8** | **0.8** | **0.8** | 0.2 | 0.1 | 0.1 | **0.1** |
| DXx1 (CTRL) | 128 | 62.5 | 0.4 | 0.4 | 0.4 | 0.2 | 0.1 | 0.1 | 0.05 |
| DX/2 | 128 | **31.25** | **0.2** | **0.2** | **0.2** | 0.2 | 0.1 | 0.1 | **0.025** |
| DTx10 | 128 | 62.5 | **4.0** | **4.0** | **4.0** | **2.0** | **1.0** | **1.0** | 0.05 |
| DTx5 | 128 | 62.5 | **2.0** | **2.0** | **2.0** | **1.0** | **0.5** | **0.5** | 0.05 |
| DTx2 | 128 | 62.5 | **0.8** | **0.8** | **0.8** | **0.4** | **0.2** | **0.2** | 0.05 |
| DTx1 (CTRL) | 128 | 62.5 | 0.4 | 0.4 | 0.4 | 0.2 | 0.1 | 0.1 | 0.05 |
| DT/2 | 128 | 62.5 | **0.2** | **0.2** | **0.2** | **0.1** | **0.05** | **0.05** | 0.05 |
| DT/4 | 128 | 62.5 | **0.1** | **0.1** | **0.1** | **0.05** | **0.025** | **0.025** | 0.05 |

The initial number-size distribution of the population of pure ammonium bisulfate particles is given by a bimodal log-normal distribution,

$$\frac{dN^{\text{sulf}}}{d\log r^{\text{sulf}}_{\text{dry}}} = \sum_{a=1}^{2} \frac{1}{\sqrt{2\pi}} \frac{c^{\text{sulf}}_{a}}{\log \sigma_a} \exp\left[ \frac{-\left(\log r^{\text{sulf}}_{\text{dry}} - \log r_a\right)^2}{2\log^2 \sigma_a} \right], \tag{95}$$

where $r_{\text{dry}}^{\text{sulf}}$ is the dry radius of the ammonium bisulfate component and $N^{\text{sulf}}$ is the accumulated number of particles smaller than $r_{\text{dry}}^{\text{sulf}}$ per unit volume of air. The particle number concentrations are $c_1^{\text{sulf}} = 270\,\text{cm}^{-3}$ and $c_2^{\text{sulf}} = 45\,\text{cm}^{-3}$, thus, the total particle number concentration is $c^{\text{sulf}} = c_1^{\text{sulf}} + c_2^{\text{sulf}} = 315\,\text{cm}^{-3}$. The geometric mean radii are $r_1 = 0.03\,\mu\text{m}$ and $r_2 = 0.14\,\mu\text{m}$, with geometric standard deviations of $\sigma_1 = 1.28$ and $\sigma_2 = 1.75$, respectively. This distribution is based on *in situ* maritime aerosol data as detailed in Sec. 2.2.3 of VanZanten et al. (2011), but the number concentration is multiplied by three. As discussed in Sec. 2.4, we consider that a droplet containing only soluble substances freezes only through a homogeneous freezing mechanism; therefore, the freezing temperature of these particles is $T^{\text{fz}} = -38\,^{\circ}\text{C}$. Therefore, pure ammonium bisulfate's initial distribution function can be calculated as

$$n^{\text{sulf}}(\log r_{\text{dry}}^{\text{sulf}}, T^{\text{fz}}) = \frac{dN^{\text{sulf}}}{d\log r_{\text{dry}}^{\text{sulf}}} \delta(T^{\text{fz}} - (-38\,^{\circ}\text{C})). \tag{96}$$

The other aerosol population consists of mineral dust internally mixed with ammonium bisulfate. We set the number concentration to $c^{\text{dust}} = 1\,\text{cm}^{-3}$, and for simplicity, set the mineral dust particle diameter to $d^{\text{dust}} = 1\,\mu\text{m}$ initially (see, e.g., Fig. 3 of Hoose et al., 2010). We assume that the size distribution of internally mixed ammonium bisulfate is the same as that of the pure ammonium bisulfate given by Eq. (95). The probability density function of the freezing temperature $p(T^{\text{fz}})$ is given by eq. (1). Here, we use the INAS density formula from Niemand et al. (2012), but based on the discussion in Niedermeier et al. (2015), we do not extrapolate the formula to lower or higher temperatures:

$$n_{\text{S}}(T) = \begin{cases} 0, & \text{for } T > T_{\text{max}}^{\text{fz}}; \\ n_{\text{S}}^{\text{Niemand}}(T), & \text{for } T_{\text{max}}^{\text{fz}} \geq T > T_{\text{min}}^{\text{fz}}; \\ n_{\text{S}}^{\text{Niemand}}(T_{\text{min}}^{\text{fz}}), & \text{for } T_{\text{min}}^{\text{fz}} \geq T; \end{cases} \tag{97}$$

where $T_{\text{max}}^{\text{fz}} = -12\,^{\circ}\text{C}$ and $T_{\text{min}}^{\text{fz}} = -36\,^{\circ}\text{C}$. The mineral dust surface area is given by $A^{\text{insol}} = \pi(d^{\text{dust}})^2$. As discussed in Sec. 2.4, we set $T^{\text{fz}} = -38\,^{\circ}\text{C}$ if the mineral dust is IN inactive and no INAS appears until $T^{\text{fz}} = -38\,^{\circ}\text{C}$. Altogether, the mineral dust distribution function is given by

$$n^{\text{dust}}(d^{\text{dust}}, \log r_{\text{dry}}^{\text{sulf}}, T^{\text{fz}})$$

$$= \delta(d^{\text{dust}} - 1\,\mu\text{m}) \frac{c^{\text{dust}}}{c^{\text{sulf}}} \frac{dN^{\text{sulf}}}{d\log r_{\text{dry}}^{\text{sulf}}}$$

$$[p(T^{\text{fz}})H(T^{\text{fz}} + 38\,^{\circ}\text{C}) + P_{\text{INia}}\delta(T^{\text{fz}} + 38\,^{\circ}\text{C})], \tag{98}$$

where $H(T)$ is the Heaviside step function and $P_{\text{INia}} := P(T^{\text{fz}} \leq -38\,^{\circ}\text{C})$ is the probability that a single INAS does not appear until $T^{\text{fz}} = -38\,^{\circ}\text{C}$. For $d^{\text{dust}} = 1\,\mu\text{m}$, $P_{\text{INia}} \approx 0.056$.

### 6.1.4 Boundary conditions for particles

We also impose periodic boundary conditions on particles for the lateral boundaries. If a particle crosses the upper or lower boundary, we remove that particle from the system.

### 6.1.5 Near-surface heating

Convective cloud development is triggered by a $20\,\mathrm{min}$ heating started from the beginning within a $10\,\mathrm{km}$ wide region centered at $x = 5\,\mathrm{km}$, and is expressed as

$$\left.\frac{\partial \rho\theta}{\partial t}\right|_{\mathrm{sfc}} = \rho H \max(W, 0), \tag{99}$$

$$W = \left(-\frac{4}{w^2}\right)\left[(x - x_0)^2 - \left(\frac{w}{2}\right)^2\right]\exp\left[-\frac{z - z_0}{z_0}\right], \tag{100}$$

where $H = 10\,\mathrm{K/h}$, $x_0 = 5\,\mathrm{km}$, $w = 10\,\mathrm{km}$, and $z_0 = 0.5\,\mathrm{km}$. The heating has a parabolic shape in the horizontal direction and decays exponentially in the vertical direction.

### 6.1.6 Grid size and time steps

We use a uniform grid throughout this study, with a grid size of $\Delta x = \Delta y = \Delta z = 62.5\,\mathrm{m}$ in the CTRL case. The time steps in the CTRL case are $\Delta t = 0.4\,\mathrm{s}$, $\Delta t_{\mathrm{adv}} = \Delta t_{\mathrm{fz/mlt}} = 0.4\,\mathrm{s}$, $\Delta t_{\mathrm{collis}} = 0.2\,\mathrm{s}$, $\Delta t_{\mathrm{cnd/evp}} = \Delta t_{\mathrm{dep/sbl}} = 0.1\,\mathrm{s}$, and $\Delta t_{\mathrm{dyn}} = 0.05\,\mathrm{s}$.

### 6.1.7 Initialization of super-particles

Initially, the super-particles are distributed uniformly throughout the domain at random with a number concentration of $c^{\mathrm{SP}} = 128\,/\mathrm{cell}$. We consider half of them as pure ammonium bisulfate aerosol particles, a few of them as IN inactive mineral dust particles internally mixed with ammonium bisulfate, and the remainder to be IN active mineral dust particles internally mixed with ammonium bisulfate.

The multiplicity, ammonium bisulfate mass, and freezing temperature of each pure ammonium bisulfate super-particle is assigned as follows. For each pure ammonium bisulfate super-particle we draw a random number uniformly in log-space from the interval $[r_{\mathrm{dry,min}}^{\mathrm{sulf}}, r_{\mathrm{dry,max}}^{\mathrm{sulf}}]$ and determine the dry radius $r_{\mathrm{dry},i}^{\mathrm{sulf}}$. To accurately represent the size distribution given in eq. (95), we set $r_{\mathrm{dry,min}}^{\mathrm{sulf}} = 10.0\,\mathrm{nm}$ and $r_{\mathrm{dry,max}}^{\mathrm{sulf}} = 5.0\,\mathrm{\mu m}$. From Eqs. (85) and (96), the super-particle's multiplicity is then given by

$$
\begin{aligned}
\xi_i &= \frac{n^{\mathrm{sulf}}(\log r_{\mathrm{dry},i}^{\mathrm{sulf}}, T_i^{\mathrm{fz}})}{N_{\mathrm{s}}(0)/2}\frac{V_{\mathrm{domain}}\log(r_{\mathrm{dry,max}}^{\mathrm{sulf}}/r_{\mathrm{dry,min}}^{\mathrm{sulf}})}{\delta(T^{\mathrm{fz}} - (-38\,^{\circ}\mathrm{C}))} \\
&= \frac{dN^{\mathrm{sulf}}}{d\log r_{\mathrm{dry}}^{\mathrm{sulf}}}\left(\log r_{\mathrm{dry},i}^{\mathrm{sulf}}\right)\frac{\log(r_{\mathrm{dry,max}}^{\mathrm{sulf}}/r_{\mathrm{dry,min}}^{\mathrm{sulf}})}{c^{\mathrm{SP}}/2},
\end{aligned} \tag{101}
$$

where $V_{\mathrm{domain}}$ is the total volume of the domain, $n$ and $p$ in Eq. (85) in this case are given by

$$n = n^{\mathrm{sulf}}(\log r_{\mathrm{dry},i}^{\mathrm{sulf}}, T_i^{\mathrm{fz}}), \tag{102}$$

$$p = \frac{\delta(T^{\mathrm{fz}} - (-38\,^{\circ}\mathrm{C}))}{V_{\mathrm{domain}}\log(r_{\mathrm{dry,max}}^{\mathrm{sulf}}/r_{\mathrm{dry,min}}^{\mathrm{sulf}})}, \tag{103}$$

and $N_{\mathrm{s}}(0)$ in Eq. (85) is replaced by $N_{\mathrm{s}}(0)/2$ because we use half of the super-particles for pure ammonium bisulfate aerosol particles. The ammonium bisulfate mass is calculated from the dry radius $r_{\mathrm{dry},i}^{\mathrm{sulf}}$ as $m_{1i}^{\mathrm{sol}} = (4\pi/3)\rho_{\mathrm{(NH)_4HSO_4}}(r_{\mathrm{dry},i}^{\mathrm{sulf}})^3$, where $\rho_{\mathrm{(NH)_4HSO_4}} = 1.78\,\mathrm{g\,cm^{-3}}$. The soluble aerosol particle freezing temperature is $T_i^{\mathrm{fz}} = -38\,^{\circ}\mathrm{C}$.

For IN inactive mineral dust super-particles, we use $P_{\text{INia}}^{\text{SP}} = 0.05$. The mineral dust initially has the same size $d^{\text{dust}} = 1\,\mu\text{m}$. The dry radius $r_{\text{dry},i}^{\text{sulf}}$ is calculated using the same procedure as the pure ammonium bisulfate aerosol particles, i.e., for each super-particle we draw a random number uniformly in log-space from the interval $[r_{\text{dry,min}}^{\text{sulf}}, r_{\text{dry,max}}^{\text{sulf}}]$. The IN inactive mineral dust freezing temperature is $T_i^{\text{fz}} = -38\,^{\circ}\text{C}$. From Eqs. (85) and (98), an IN inactive mineral dust super-particle's multiplicity is then given by

$$\xi_i = \frac{c^{\text{dust}}}{c^{\text{sulf}}} \frac{dN^{\text{sulf}}}{d\log r_{\text{dry}}^{\text{sulf}}} \left(\log r_{\text{dry},i}^{\text{sulf}}\right) \frac{\log(r_{\text{dry,max}}^{\text{sulf}}/r_{\text{dry,min}}^{\text{sulf}})}{c^{\text{SP}}/2} \frac{P_{\text{INia}}}{P_{\text{INia}}^{\text{SP}}}. \tag{104}$$

Finally, we consider IN active mineral dust internally mixed with ammonium bisulfate. The remaining super-particles, i.e., $(1 - P_{\text{INia}}^{\text{SP}})/2$, are used for this population. The initial diameter of the mineral dust initial is $d^{\text{dust}} = 1\,\mu\text{m}$, and the dry radius $r_{\text{dry},i}^{\text{sulf}}$ is determined as in the other populations. We draw another random number uniformly from the interval $[T_{\text{min}}^{\text{fz}}, T_{\text{max}}^{\text{fz}}]$ and determine the freezing temperature $T_i^{\text{fz}}$. From Eqs. (85) and (98), an IN active mineral dust super-particle's multiplicity is then given by

$$\xi_i = \frac{c^{\text{dust}}}{c^{\text{sulf}}} \frac{dN^{\text{sulf}}}{d\log r_{\text{dry}}^{\text{sulf}}} \left(\log r_{\text{dry},i}^{\text{sulf}}\right) p(T_i^{\text{fz}})$$
$$\frac{\log(r_{\text{dry,max}}^{\text{sulf}}/r_{\text{dry,min}}^{\text{sulf}})(T_{\text{max}}^{\text{fz}} - T_{\text{min}}^{\text{fz}})}{(c^{\text{SP}}/2)(1 - P_{\text{INia}}^{\text{SP}})}. \tag{105}$$

Note that multiplicity $\xi_i$ is an integer variable. We round the r.h.s. of Eqs. (101)–(105) to the nearest integer, and if the r.h.s. is $< 1$, we draw a random number to decide whether to choose $\xi_i = 1$ or $\xi_i = 0$ to avoid sampling error. If $\xi_i = 0$, the super-particle will be removed from the system.

Assuming that all the particles are deliquescent, we consider that the initial droplet radius $r_i$ is equal to the equilibrium radius of condensation/evaporation growth equation (8). As the vapor profile is initially sub-saturated relative to liquid water and all particles contain soluble substances, the growth equation (8) has a unique, stable equilibrium solution.

### 6.1.8 Pseudo-random numbers

To evaluate the fluctuation, we conduct a 10-member ensemble of simulations by changing the pseudo-random number sequence.

Now, the atmospheric conditions and numerical parameters used for the CTRL ensemble have all been specified.

### 6.2 Other ensembles for investigating numerical convergence

We also try various other test cases by changing the CTRL ensemble's numerical parameters, and assess the sensitivity of results to numerical parameters. Our parameter selections are specified in the following sections and a summary is provided in Table 2.

### 6.2.1 NSP ensembles for super-particle number convergence

To investigate numerical convergence with respect to initial the super-particle number concentration $c^{SP}$, we vary $c^{SP}$ as follows: 2, 4, …, $512$/cell. Grid size and time steps are not changed. These cases are respectively denoted by NSP002, NSP008, ..., NSP512. Note that NSP128 and CTRL are the same.

### 6.2.2 DX ensembles for grid convergence

To investigate numerical convergence with respect to the grid size, we run ensembles using different grid sizes.

The grid size of the DXx4 ensemble is four times that of CTRL: $\Delta x = \Delta y = \Delta z = 250\,\mathrm{m}$, $\Delta t_{\mathrm{dyn}} = 0.2\,\mathrm{s}$, and $\Delta t = \Delta t_{\mathrm{adv}} = \Delta t_{\mathrm{fz/mlt}} = 1.6\,\mathrm{s}$, but other time steps, i.e., $\{\Delta t_{\mathrm{collis}}, \Delta t_{\mathrm{cnd/evp}}, \Delta t_{\mathrm{dep/sbl}}\}$, are not changed.

The DXx2 ensemble's grid size is twice that of CTRL: $\Delta x = \Delta y = \Delta z = 125\,\mathrm{m}$, $\Delta t_{\mathrm{dyn}} = 0.1\,\mathrm{s}$, and $\Delta t = \Delta t_{\mathrm{adv}} = \Delta t_{\mathrm{fz/mlt}} = 0.8\,\mathrm{s}$.

Note that DXx1 and CTRL are the same.

The DX/2 ensemble has a grid size that is half that of CTRL: $\Delta x = \Delta y = \Delta z = 31.25\,\mathrm{m}$. $\Delta t_{\mathrm{dyn}} = 0.025\,\mathrm{s}$, and $\Delta t = \Delta t_{\mathrm{adv}} = \Delta t_{\mathrm{fz/mlt}} = 0.2\,\mathrm{s}$.

### 6.2.3 DT ensembles for time step convergence

To investigate numerical convergence with respect to the cloud microphysics time steps, we change the cloud microphysics time steps for CTRL without changing the time step for fluid dynamics.

The time steps for the DTx10 ensemble's cloud microphysics are ten times that of CTRL: $\Delta t = \Delta t_{\mathrm{adv}} = \Delta t_{\mathrm{fz/mlt}} = 4.0\,\mathrm{s}$, $\Delta t_{\mathrm{collis}} = 2.0\,\mathrm{s}$, and $\Delta t_{\mathrm{cnd/evp}} = \Delta t_{\mathrm{dep/sbl}} = 1.0\,\mathrm{s}$. $\Delta t_{\mathrm{dyn}}$ is not changed.

The time steps of the DTx5 ensemble are five times that of CTRL: $\Delta t = \Delta t_{\mathrm{adv}} = \Delta t_{\mathrm{fz/mlt}} = 2.0\,\mathrm{s}$, $\Delta t_{\mathrm{collis}} = 1.0\,\mathrm{s}$, and $\Delta t_{\mathrm{cnd/evp}} = \Delta t_{\mathrm{dep/sbl}} = 0.5\,\mathrm{s}$.

The time steps of the DTx2 ensemble are twice that of CTRL: $\Delta t = \Delta t_{\mathrm{adv}} = \Delta t_{\mathrm{fz/mlt}} = 0.8\,\mathrm{s}$, $\Delta t_{\mathrm{collis}} = 0.4\,\mathrm{s}$, and $\Delta t_{\mathrm{cnd/evp}} = \Delta t_{\mathrm{dep/sbl}} = 0.2\,\mathrm{s}$.

Note that DTx1 and CTRL are the same.

The time steps of the DT/2 ensemble are half that of CTRL: $\Delta t = \Delta t_{\mathrm{adv}} = \Delta t_{\mathrm{fz/mlt}} = 0.2\,\mathrm{s}$, $\Delta t_{\mathrm{collis}} = 0.1\,\mathrm{s}$, and $\Delta t_{\mathrm{cnd/evp}} = \Delta t_{\mathrm{dep/sbl}} = 0.05\,\mathrm{s}$.

The time steps of the DT/4 ensemble are one quarter that of CTRL: $\Delta t = \Delta t_{\mathrm{adv}} = \Delta t_{\mathrm{fz/mlt}} = 0.1\,\mathrm{s}$, $\Delta t_{\mathrm{collis}} = 0.05\,\mathrm{s}$, and $\Delta t_{\mathrm{cnd/evp}} = \Delta t_{\mathrm{dep/sbl}} = 0.025\,\mathrm{s}$.

## 7 Typical behavior of CTRL ensemble

From the 10 CTRL ensemble members, we selected the one that produced accumulated precipitation amounts closest to the mean value as the representative, hereafter referred to as the typical realization of CTRL. In this section, we analyze these results in detail.

### 7.1 Hydrometeor categorization

We do not categorize hydrometeors during the simulation, which is one of the salient features of our model because the artificial partitioning of hydrometeors could cause various artifacts. In contrast, when analyzing results, dividing hydrometeors into categories is useful to precisely understand the results.

In this study, we assume that hydrometeors completely freeze or melt instantaneously (see Secs. 4.1.4 and 4.1.5). Further, we assume that all particles contain soluble components and are hygroscopic. If not frozen, we assume that the particles are deliquescent even when humidity is low (see Sec. 4.1.6). We also introduced a limiter (14) to prevent complete sublimation. Hence, all particles can be categorized as either droplets or ice particles with no ambiguity.

If a particle is a droplet and its radius $r$ is $< 40\,\mu\mathrm{m}$, we consider it a cloud droplet. Otherwise, the particle is considered a rain droplet.

If a particle is an ice particle with a rimed mass fraction satisfying $m^{\mathrm{rime}}/m > 0.3$, we consider it a graupel particle. This criterion is based on the riming categories in Fig. 5 of Mosimann et al. (1994), in which $0.3$ corresponds to a densely rimed ice crystal. If the maximum dimension of a graupel particle is $> 5\,\mathrm{mm}$, we consider it a hailstone. However, for the sake of simplicity, we consider hailstones as a subset of graupel and they will not be distinguished in the figures. If the ice particle is not a graupel particle, but rather is composed of $> 10$ monomers, i.e., $n^{\mathrm{mono}} > 10$, we consider the ice particle a snow aggregate. Otherwise, we categorize the ice particle as a cloud ice particle.

### 7.2 Spatial structure of the cloud, water path, and precipitation amount

We first analyze the cloud's overall properties, and then, in the next section, we analyze the properties of individual ice particles.

Figure 1 shows how the cloud's spatial structure in the typical realization of CTRL evolved over time. The mixing ratio of cloud water, rainwater, cloud ice, graupel, and snow aggregates are plotted in fading white, yellow, blue, red, and green, respectively. See also Movie 1 in the Supplement.

Figure 2 shows how the amounts of hydrometeors in the atmosphere evolved over time. The domain-averaged cloud water, rainwater, cloud ice water, graupel water, and snow aggregate water paths are plotted in gray, yellow, blue, red, and green, respectively. Figure 3 shows the time evolution of domain-averaged accumulated precipitation amounts. The solid lines represent the typical realization of CTRL in both figures. The dark shades indicate the mean $\pm$ standard deviation that were calculated using the 10 CTRL ensemble members. The unbiased estimator was used to calculate the standard deviation. Pale shades indicate the maximum and minimum values of the 10 ensemble members.

The cloud started to form at approximately $t = 1200\,\text{s}$, and at approximately $t = 1900\,\text{s}$, rain droplets started to be created through warm rain microphysics processes. Soon after that, supercooled droplets near the cloud top started to freeze and the number of supercooled cloud droplets quickly decreased because of the Wegener–Bergeron–Findeisen process and riming. At the same time, we also observed that convective cores near the homogeneous freezing level ($z \approx 9.3\,\text{km}$) containing high

liquid water content were sustained until around $t = 5000\,\text{s}$. For example, at $t = 3300\,\text{s}$, we observed a liquid water content of $2.1\,\text{g}\,\text{m}^{-3}$ at $(x, z) = (21.8\,\text{km}, 9.8\,\text{km})$, where $T = -37.5\,°\text{C}$. The existence of such a high supercooled liquid water content down to the homogeneous freezing limit $-38\,°\text{C}$ are frequently observed in deep convective clouds (Rosenfeld and Woodley, 2000). The ice particles quickly evolved into graupel particles through riming, and then fell toward the surface. When crossing the freezing level at approximately $z = 4.1\,\text{km}$, the graupel instantaneously melted into rain droplets, based on our model. The

peak of the rainwater path at $t = 2800\,\text{s}$ was created by graupel melting. The first rain droplet reached the surface at about $t = 2800\,\text{s}$, and heavy precipitation was sustained for $1200\,\text{s}$, followed by weak precipitation. At the end of the simulation ($t = 5400\,\text{s}$), the domain-averaged accumulated precipitation amount was $1.2\,\text{mm}$. An anvil cloud was created between $z = 10\,\text{km}$ and $z = 12\,\text{km}$. The anvil cloud was mostly composed of cloud ice particles, with a small amount of snow aggregates that increased slowly over time through the aggregation of cloud ice particles. The maximum updraft and downdraft speeds

were $39.0\,\text{ms}^{-1}$ and $21.9\,\text{ms}^{-1}$, which were observed at $(t, x, z) = (2340\,\text{s}, 12.8\,\text{km}, 11.1\,\text{km})$ and $(1620\,\text{s}, 9.5\,\text{km}, 4.1\,\text{km})$, respectively.

Our model successfully simulated the life cycle of a cumulonimbus typically observed in nature (see, e.g., Chap. 8 of Cotton et al., 2010). At the same time, our results are limited because the simulation was conducted in 2D; the turbulence characteristics are different in 2D and 3D. Furthermore, the convection was initiated from a stratified, non-turbulent atmosphere; however,

this is unrealistic. Following Lasher-Trapp et al. (2005), imposing a spin-up period to develop turbulence in the boundary layer before initiating the deep convection would be desirable.

### 7.3  Ice particle morphology and fall speeds

Now, we analyze the properties of individual ice particles in the typical realization of CTRL.

Figure 4 shows the mass-dimension relationship of ice particles at $t = 2040\,\text{s}$ (towering stage), $3000\,\text{s}$ (mature stage), and

$5400\,\text{s}$ (dissipating stage). The 2D mass densities of cloud ice particles, graupel particles, and snow aggregates are plotted in fading blue, red, and green, respectively. The horizontal axis represents the maximum ice particle dimension $D$. The vertical axis represents the normalized ice particle mass $m^*$, which is defined by the ratio of ice particle mass $m$ to the mass of a spherical ice particle with the same maximum dimension $D$ and the true density of ice $\rho^{\text{i}}_{\text{true}}$:

$$m^* := \frac{\rho^{\text{i}} a^2 c}{\rho^{\text{i}}_{\text{true}} (D/2)^3}, \quad D = 2\max(a, c). \tag{106}$$

Note that $m^* \leq 1$ always holds. To calculate 2D mass densities, we divided the 2D $D$-$m^*$ space into $100 \times 100$ bins, accumulated the masses of ice particles in each bin, and divided the total masses by the area of each bin measured in $\log_{10}(D)$ and $\log_{10}(m^*)$. The colored slopes in Fig. 4 represent mass-dimension relationship formulas from various studies, and M96, HK87, K89, M90, and LH74 indicate Mitchell (1996), Heymsfield and Kajikawa (1987), Kajikawa (1989), Mitchell et al. (1990), and

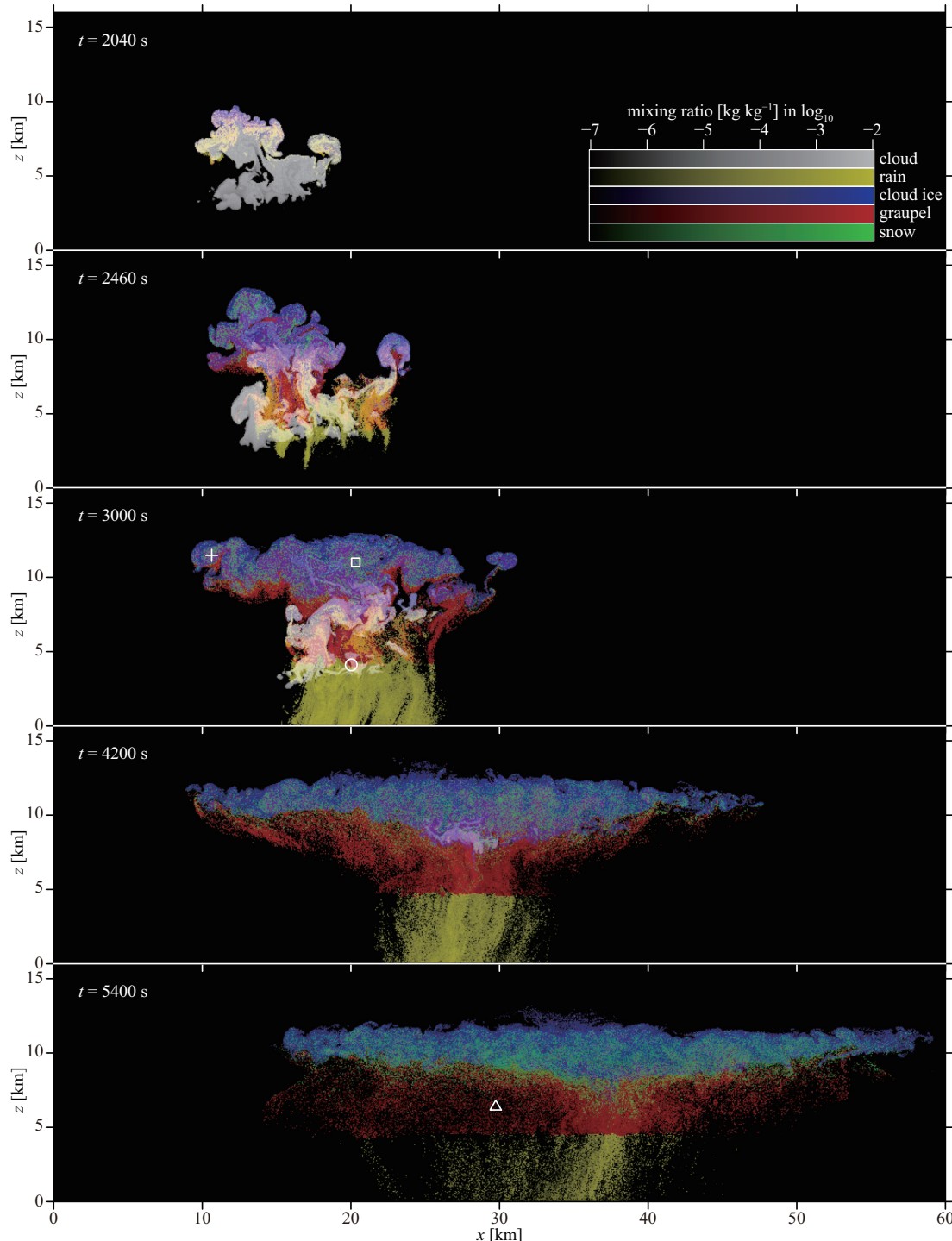

**Figure 1.** Typical realization of CTRL cloud spatial structures at $t = 2040\,\text{s}$, $2460\,\text{s}$, $3000\,\text{s}$, $4200\,\text{s}$, and $5400\,\text{s}$. The mixing ratio of cloud water, rainwater, cloud ice, graupel, and snow aggregates are plotted in fading white, yellow, blue, red, and green, respectively. The symbols indicate examples of unrealistic predicted ice particles (Sec. 7.3 and Sec. 9.1). See also Movie 1 in the Supplement.

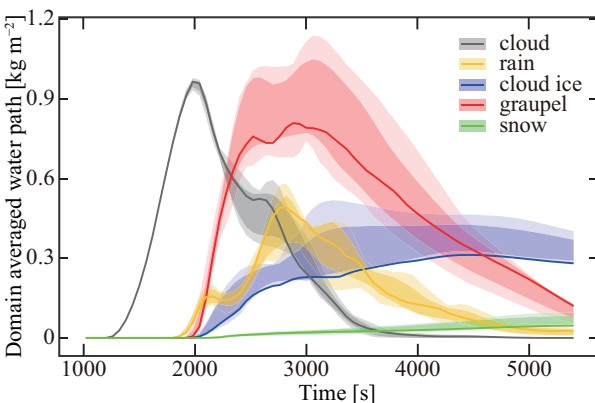

**Figure 2.** Time evolution of the domain-averaged water path in the CTRL ensemble. The cloud water, rainwater, cloud ice water, graupel water, and snow aggregate water paths are plotted in gray, yellow, blue, red, and green, respectively. The solid line represents the typical realization of CTRL. Dark shades indicate the mean $\pm$ standard deviation, and pale shades indicate the maximum and minimum values of the 10 ensemble members.

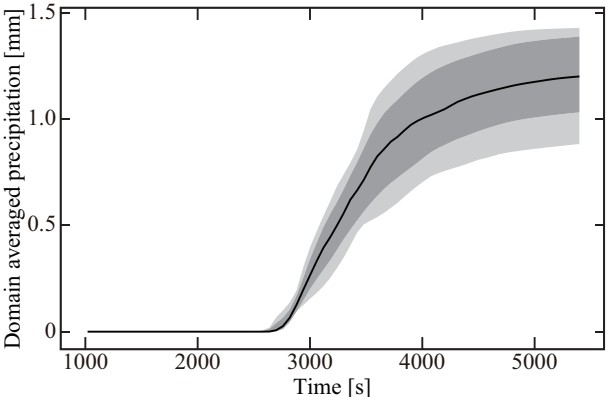

**Figure 3.** Time evolution of domain-averaged accumulated precipitation amounts in the CTRL ensemble. The solid line represents the typical realization of CTRL. The dark shade indicates the mean $\pm$ standard deviation, and the pale shade indicates the maximum and minimum values of the 10 ensemble members.

Locatelli and Hobbs (1974), respectively. Note that "crystals with sector like branches (M96)" and "stellar crystals with broad arms (M96)" consists of two slopes, respectively, but both are not continuous. See also Movie 2 in the Supplement.

Figure 5 shows the relationship between ice particle aspect ratios and dimensions at $t = 2040\,\text{s}$, $3000\,\text{s}$, and $5400\,\text{s}$. The horizontal axis represents the maximum ice particle dimension $D$, and the vertical axis represents the ice particle aspect ratio $\phi$. The 2D mass densities of cloud ice particles, graupel particles, and snow aggregates are plotted in the same manner as Fig. 4, except for differences in the vertical axis. Note that if $\phi > 1$ ($\phi < 1$), ice particles are columnar (planar). See also Movie 3 in the Supplement.

Figure 6 shows the relationship between ice particle apparent densities and dimensions at $t = 2040\,\text{s}$, $3000\,\text{s}$, and $5400\,\text{s}$. The horizontal axis represents the maximum ice particle dimension $D$, and the vertical axis represents ice particle apparent density $\rho^{\text{i}}$. The 2D mass densities of cloud ice particles, graupel particles, and snow aggregates are plotted in the same manner as Fig. 4, except for differences in the vertical axis. See also Movie 4 in the Supplement.

Figure 7 shows the relationship between ice particle velocities and dimensions at $t = 2040\,\text{s}$, $3000\,\text{s}$, and $5400\,\text{s}$. The horizontal axis represents the maximum ice particle dimension $D$, and the vertical axis represents ice particle terminal velocity $v^{\infty}$. The 2D mass densities of cloud ice particles, graupel particles, and snow aggregates are plotted in the same manner as Fig. 4, except for differences in the vertical axis. The colored slopes in Fig. 7 represent the velocity-dimension relationship formulas from various studies, and SC85, W08, H72, KH83, A72, and H02 indicate Starr and Cox (1985), Westbrook et al. (2008), Heymsfield (1972), Knight and Heymsfield (1983), Auer (1972), and Heymsfield et al. (2002), respectively. "Stokes' law for ice spheres" is based on the Stokes' terminal velocity for spherical ice particles with the true ice density. We use the dynamic viscosity at a temperature of $-20\,°\text{C}$, i.e., $\mu = 1.630 \times 10^{-5}\,\text{kg}\,\text{m}^{-1}\text{s}^{-1}$. The two slopes of W08 are based on the analytical formula of Westbrook et al. (2008) for $< 100\,\mu\text{m}$ ice particles. For "hexagonal plates", $L/2a = 0.05$ is assumed, with $L$ being the height of the hexagonal prism and $a = D/2$ being the hexagon's maximal radius. The effective radius is calculated using the horizontal orientation model from Roscoe (1949). For "hexagonal columns", $L/2a = 20$ is assumed, and the effective radius is calculated using the random orientation model of Hubbard and Douglas (1993). In both cases, we use the dynamic viscosity at a temperature of $-20\,°\text{C}$. See also Movie 5 in the Supplement.

At $t = 2040\,\text{s}$ (towering stage), cloud glaciation had just started, and a small amount of planar and columnar cloud ice particles and graupel particles can be observed. The two horizontal red bands at $\phi = 0.8$ and $\phi = 1/0.8$ in Fig. 5 were created because of our assumption that riming growth eventually makes ice particles quasi-spherical.

At $t = 3000\,\text{s}$ (mature stage), many hailstones (graupel particles $> 5\,\text{mm}$) can be observed in the cloud's middle layer. We also have many columnar cloud ice particles and a small number of snow aggregates in the upper part of the cloud. Those cloud ice particles were columnar because our model's inherent growth ratio $\Gamma(T)$ is $> 1$ at this height. Many of the snow aggregates were spherical because our model assumed that the aspect ratio $\phi$ approaches 1 as aggregation occurs.

At $t = 5400\,\text{s}$ (dissipating stage), most of the graupel particles had fallen away and only a small amount remained. More columnar cloud ice particles and snow aggregates can be observed in the anvil.

The mass-dimension relationship shown in Fig. 4, and the velocity-dimension relationship shown in Fig. 7 show a reasonable agreement between our model's predicted results and existing formulas based on laboratory measurements and observations.

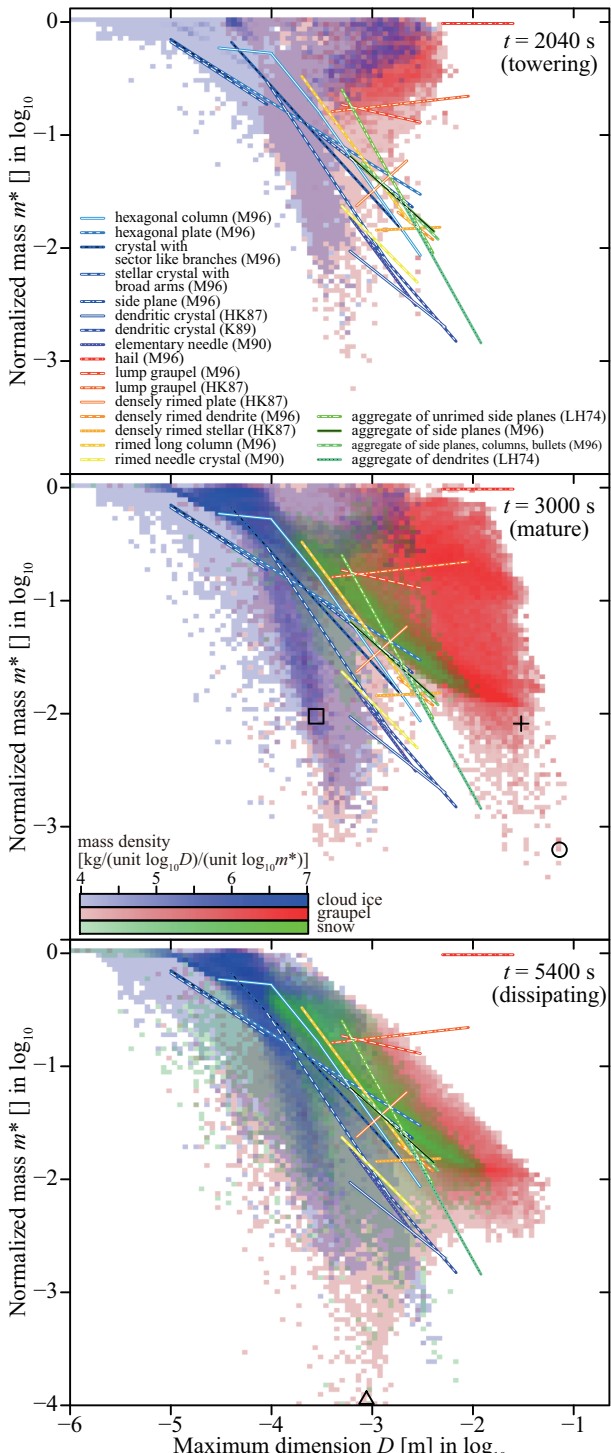

**Figure 4.** Mass-dimension relationship of ice particles in the typical realization of CTRL at $t = 2040\,\mathrm{s}$, $3000\,\mathrm{s}$, and $5400\,\mathrm{s}$. The 2D mass densities of cloud ice particles, graupel particles, and snow aggregates are plotted in fading blue, red, and green, respectively. The horizontal and vertical axes represent the maximum ice particle dimension $D$, and the normalized ice particle mass $m^*$, respectively. The colored slopes represent various mass-dimension relationship formulas. The symbols indicate examples of unrealistically predicted ice particles (Sec. 7.3 and Sec. 9.1). See also Movie 2 in the Supplement.

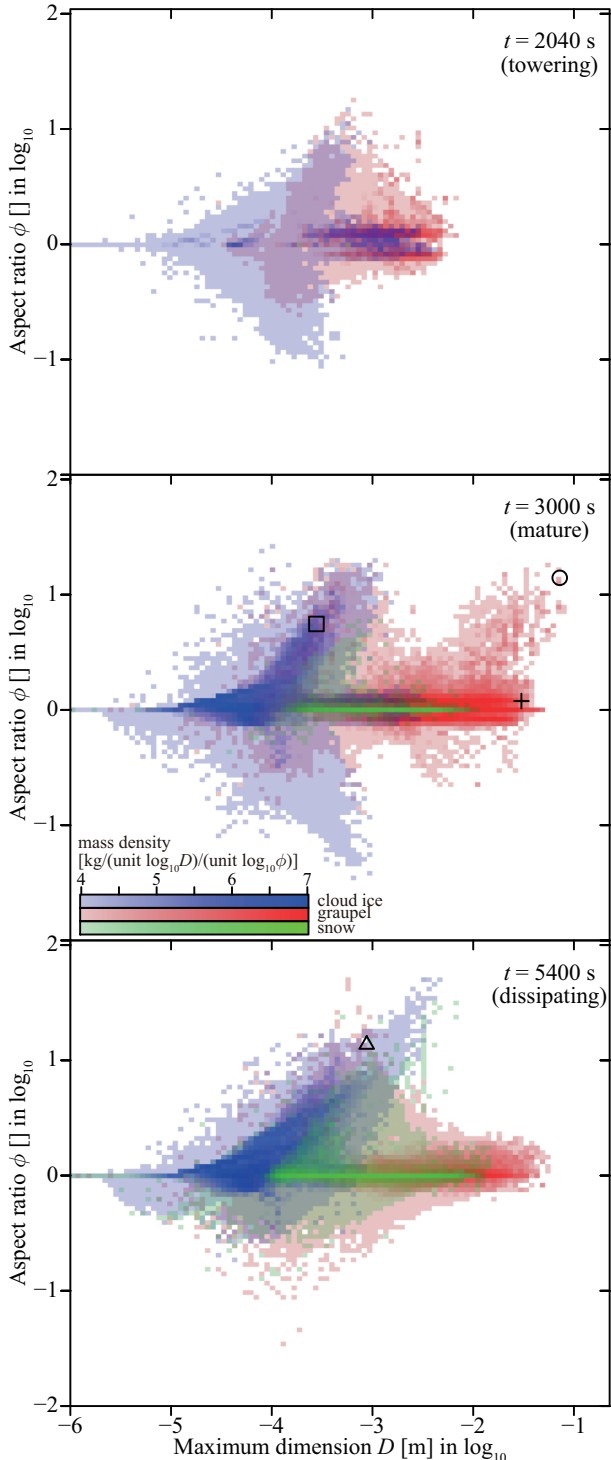

**Figure 5.** Aspect ratio-dimension relationship of ice particles in the typical realization of CTRL at $t = 2040\,\mathrm{s}$, $3000\,\mathrm{s}$, and $5400\,\mathrm{s}$. The vertical axis represents the ice particle aspect ratio $\phi$. This figure is the same as Fig. 4, except for the vertical axis. The symbols indicate examples of unrealistically predicted ice particles (Sec. 7.3 and Sec. 9.1). See also Movie 3 in the Supplement.

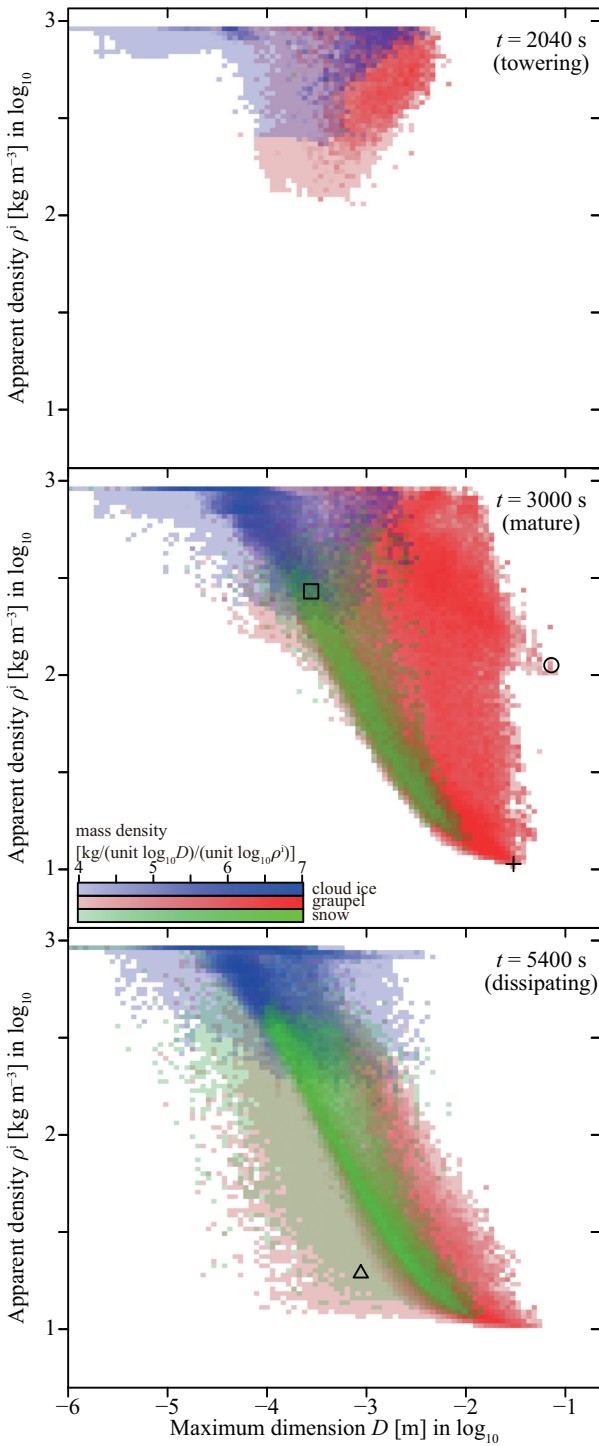

**Figure 6.** Apparent density-dimension relationship of ice particles in the typical realization of CTRL at $t = 2040$ s, $3000$ s, and $5400$ s. The vertical axis represents the ice particle apparent density $\rho^i$. This figure is the same as Fig. 4, except for the vertical axis. The symbols indicate examples of unrealistically predicted ice particles (Sec. 7.3 and Sec. 9.1). See also Movie 4 in the Supplement.

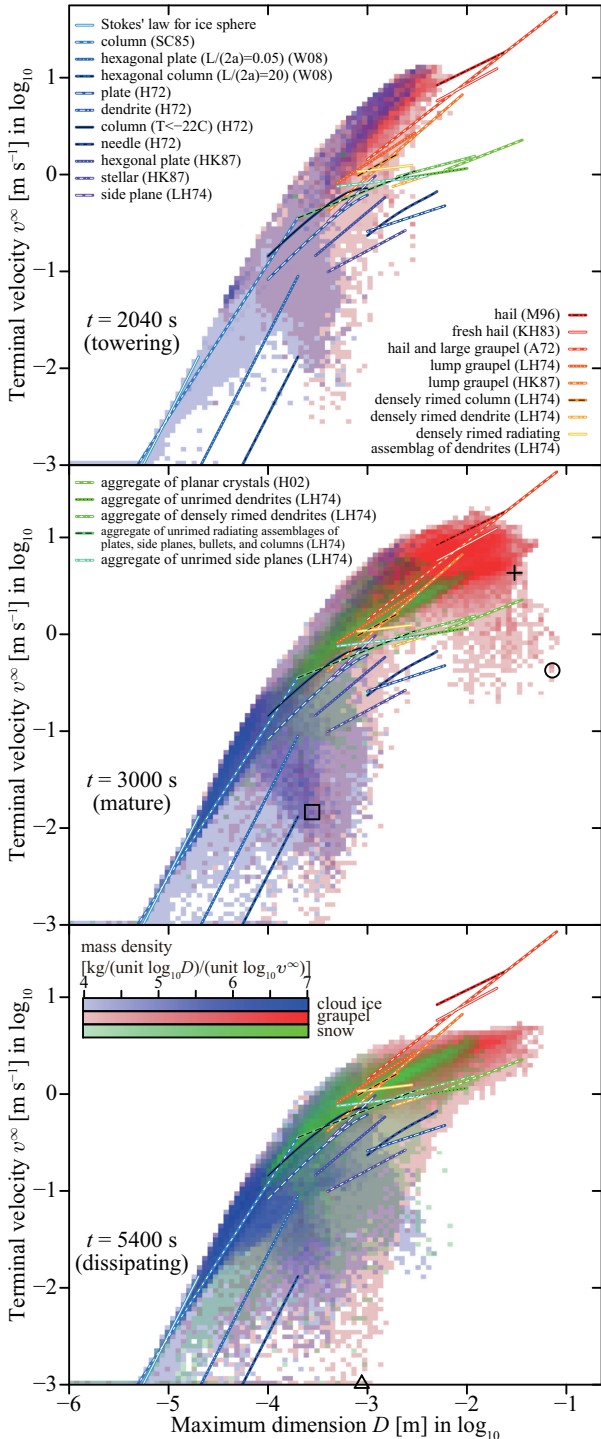

**Figure 7.** Velocity-dimension relationship of ice particles in the typical realization of CTRL at $t = 2040\,\text{s}$, $3000\,\text{s}$, and $5400\,\text{s}$. The vertical axis represents the ice particle terminal velocity $v^{\infty}$. This figure is the same as Fig. 4, except for the vertical axis. The colored slopes represent various velocity-dimension relationship formulas. The symbols indicate examples of unrealistically predicted ice particles (Sec. 7.3 and Sec. 9.1). See also Movie 5 in the Supplement.

In both figures, cloud ice particles, graupel particles, and snow aggregates are distributed near the blue, red, and green slopes, respectively. In Fig. 7, one might note that snow aggregates in our model fall faster than those in the formulas; however, this bias can be explained by the air density dependence of the fall speed. The green slopes in Fig. 7 represent the formulas of LH74 and H02. LH74's formulas are constructed from data measured between altitudes of $750$ and $1500\,\mathrm{m}$ above sea level; hence, the density is approximately $1.1\,\mathrm{kg\,m^{-3}}$. H02's formula is for temperature and pressure of $-10\,^\circ\mathrm{C}$ and $500\,\mathrm{hPa}$; hence, the density is approximately $0.66\,\mathrm{kg\,m^{-3}}$. In our simulation, most of the snow aggregates exist in the anvil cloud, wherein the density is approximately $0.38\,\mathrm{kg\,m^{-3}}$. Khvorostyanov and Curry (2002) estimated that the terminal velocities of large ice particles scale with the ambient density to the power of $-1/2$. Therefore, we can incorporate the density dependence by multiplying the LH74's formulas for aggregates by a factor of $(0.38\,\mathrm{kg\,m^{-3}}/1.1\,\mathrm{kg\,m^{-3}})^{1/2} \approx 1.70$ and that of H02 for aggregates by a factor of $(0.38\,\mathrm{kg\,m^{-3}}/0.66\,\mathrm{kg\,m^{-3}})^{1/2} \approx 1.32$. We confirmed that these corrections improve the agreement between our model results and the formulas (see Fig. R2-1 in the authors' response to anonymous referee #2).

However, at the same time, we also see several types of seemingly unrealistic ice particles, representative examples of which are indicated by symbols in Figs. 1, 4–7: The ice particle denoted by the circle at $t = 3000\,\mathrm{s}$ is a long, slowly falling hailstone. The square at $t = 3000\,\mathrm{s}$ is a columnar cloud ice particle that is inconsistent with known mass-dimension relationships. The cross at $t = 3000\,\mathrm{s}$ is a hailstone with an extremely low apparent density. The triangle at $t = 5400\,\mathrm{s}$ is an extremely long graupel particle with a low apparent density. In Sec. 9.1, we will investigate the causes of these odd behaviors in more detail, but those issues could be attributed to uncertainties in ice microphysics process formulations.

## 8  Numerical convergence characteristics

Our numerical model uses three types of numerical parameters, namely the super-particle number concentration, grid size, and time steps. These parameters correspond to the resolution of aerosol/cloud/precipitation particle distribution in real space and attribute space, the spatial resolution of moist air, and temporal resolution. The numerical solution from our model approaches the true solution of time evolution equations (2)–(81) as the super-particle number approaches the number of real particles, and the grid size and time steps approach zero.

To confirm the numerical convergence of the cumulonimbus case, we conducted a series of simulations changing the numerical parameters of CTRL. These ensembles are referred to as NSP, DX, and DT (see Table 2). Our results suggest that the numerical parameters used for the CTRL case could produce accurate numerical results. In what follows, the detail of this analysis is presented. Then, we conduct a general discussion of our model's numerical convergence characteristics and computational cost.

### 8.1  NSP ensembles and super-particle number convergence

Numerical convergence regarding the initial super-particle number concentration $c^{\mathrm{SP}}$ was investigated by varying the $c^{\mathrm{SP}}$ value of CTRL as follows: 2, 4, …, $512\,/\mathrm{cell}$ (see Table 2). These cases are referred to as NSP002, NSP004, ..., and NSP512, respectively. Note that NSP128 and CTRL are the same.

Figure 8 shows the accumulated precipitation amount statistics at the end of the simulation ($t = 5400\,\mathrm{s}$) versus the initial super-particle number concentration $c^{\mathrm{SP}}$. The error bars indicate the mean and standard deviation calculated from the 10 members of each NSP ensemble. The unbiased estimator was used to calculate standard deviations. The crosses denote the maximum and minimum values of the 10 ensemble members.

Figure 9 shows the statistics of the maximum water path of each hydrometeor type during the simulation (i.e., the maximum of each line in Fig. 2) versus the initial super-particle number concentration $c^{\mathrm{SP}}$. The error bars indicate the mean and standard deviation from the 10 members of each NSP ensemble. The unbiased estimator was used for calculating the standard deviations. The symbols indicate the maximum and minimum values of each hydrometeor type in the 10 ensemble members.

Our model has two sources of fluctuation, namely atmospheric turbulence and SDM randomness. Pseudo-random numbers
are used for the Monte Carlo calculation of coalescence, riming, and aggregation, and to initialize the super-particles. The standard deviation (i.e., fluctuation) caused by SDM randomness decreases proportionally to the inverse of the square root of the super-particle number. However, Figs. 8 and 9 show that the fluctuation is not sensitive to the initial super-particle number concentration $c^{\mathrm{SP}}$. This indicates that fluctuations in all simulations are mostly dominated by atmospheric turbulence. One might note that the fluctuations are slightly increasing as $c^{\mathrm{SP}}$ increases. This suggests that the super-particle number affects the
turbulence characteristics; however, we leave that for further investigation in future work.

Figure 8 indicates that the accumulated precipitation amount is less sensitive to the super-particle number. However, Fig. 9 reveals that the initial super-particle number concentration $c^{\mathrm{SP}}$ affects the maximum water path statistics. The numerical convergence of the maximum cloud water path is noticeably slow. This is closely related to the onset of warm rain through coalescence. From Fig. 2, we determine that the maximum of the cloud water path coincides with the emergence of rainwater.
Therefore, a small shift of the warm rain onset time changes the maximum cloud water path; however, it does not have a considerable impact on the overall properties of the simulated cloud. The maximum water paths of all the other hydrometeor types do not show a significant difference if $c^{\mathrm{SP}}$ is larger than 64 or $128\,/\mathrm{cell}$ (see also Table R2-1 of authors' response to anonymous referee #2). When the number of super-particles was too low, more rain droplets were produced because of an erroneous enhancement of coalescence that reduced the amount of cloud droplets, cloud ice particles, and graupel particles.
To summarize, we conclude that the numerical convergence regarding the super-particle number is fairly well achieved at NSP128 (CTRL), i.e., $c^{\mathrm{SP}} = 128\,/\mathrm{cell}$.

## 8.2  DX ensembles and grid convergence

We investigated the numerical convergence with respect to the grid size by varying $\Delta x = \Delta y = \Delta z$ of CTRL as follows: 31.25, 62.5, 125, and $250\,\mathrm{m}$ (see Table 2). These cases are referred to as DX/2, DXx1, DXx2, and DXx4, respectively. Note that DXx1
and CTRL are the same.

Figure 10 shows the accumulated precipitation amount statistics at the end of the simulation versus the grid size, plotted in the same way as Fig. 8, except for the difference in the horizontal axis.

Figure 11 shows the maximum water path statistics for each hydrometeor type during the simulation versus the grid size, plotted in the same way as Fig. 9, except for the difference in the horizontal axis.

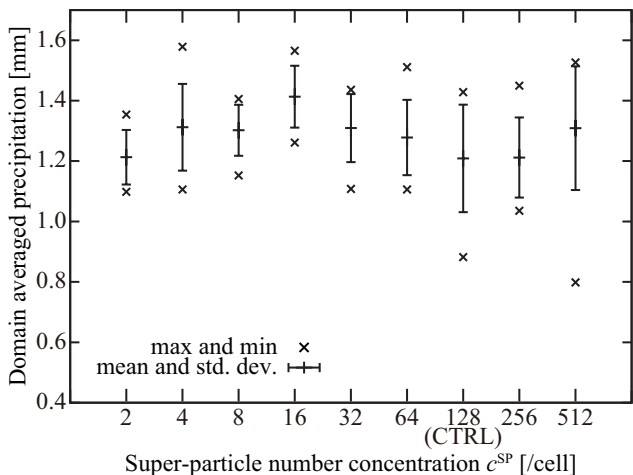

**Figure 8.** Statistics of NSP ensemble accumulated precipitation amounts. The vertical axis represents the accumulated precipitation at the end of the simulation ($t = 5400\,$s), and the horizontal axis represents the initial super-particle number concentration $c^{\mathrm{SP}}$. The error bars indicate the mean and standard deviation calculated from the 10 members of each NSP ensemble. The crosses denote maximum and minimum values of the 10 ensemble members.

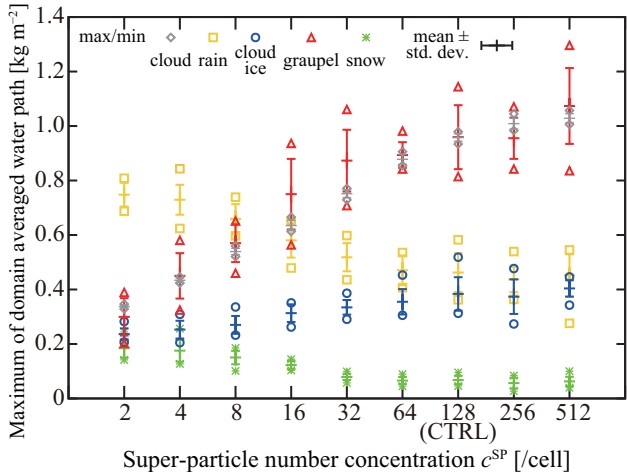

**Figure 9.** Statistics of NSP ensemble maximum water paths for each hydrometeor type . The vertical axis represents the maximum water path of each hydrometeor type during the simulation (i.e., the maximum of each line in Fig. 2), and the horizontal axis represents the initial super-particle number concentration $c^{\mathrm{SP}}$. The error bars indicate the mean and standard deviation calculated from the 10 members of each NSP ensemble. The symbols denote the maximum and minimum values of the 10 ensemble members.

The DX/2 ensemble is the highest grid resolution tested in this study, and a snapshot of the cloud from the DX/2 ensemble is shown in Fig. 12. The mixing ratios are plotted in the same manner as Fig. 1. See also Movie 6 in the Supplement.

Figure 10 shows that the accumulated precipitation amount increased from a grid size of $125\,\mathrm{m}$ to a grid size of $62.5\,\mathrm{m}$, but no significant difference exists between the $62.5\,\mathrm{m}$ and $31.25\,\mathrm{m}$ grids. Similar behavior can be observed in the maximum rainwater path in Fig. 11; however, no significant difference exists for other hydrometeor types. Comparing Fig. 12 ($31.25\,\mathrm{m}$) and Fig. 1 ($62.5\,\mathrm{m}$), the spatial structures of the clouds also look similar.

Therefore, we conclude that the numerical convergence with respect to the grid size is achieved at DXx1 (CTRL), i.e., $\Delta x = \Delta y = \Delta z = 62.5\,\mathrm{m}$.

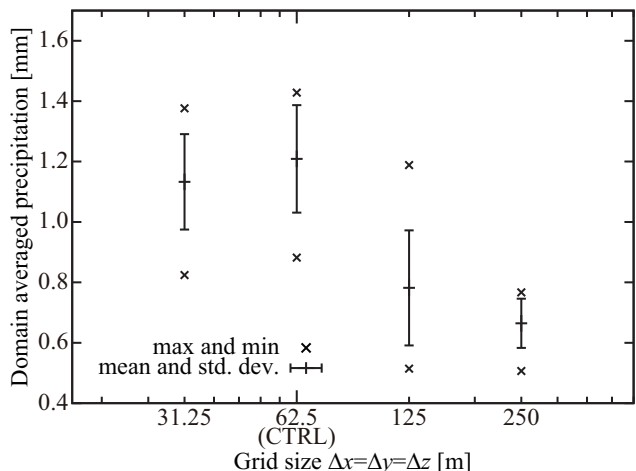

**Figure 10.** Statistics of DX ensemble accumulated precipitation amounts. The horizontal axis represents the grid size $\Delta x = \Delta y = \Delta z$. This figure has the same form as Fig. 8, except for the horizontal axis.

## 8.3    DT ensembles and time step convergence

We investigated the numerical convergence with respect to the cloud microphysics time steps by varying CTRL's cloud microphysics time steps by factors of $1/4$, $1/2$, $1$, $2$, $5$, and $10$ (see Table 2). These cases are referred to as DT/4, DT/2, DTx1, DTx2, DTx5, and DTx10, respectively. Note that DTx1 and CTRL are the same.

We found that DTx10 diverges at around $t = 1200\,\mathrm{s}$ because of a numerical instability. Let us compare the results of the other five ensembles.

Figure 13 shows the statistics of the accumulated precipitation amounts at the end of the simulation versus the ratio of cloud microphysics time steps to CTRL, plotted in the same manner as Fig. 8, except for the difference in the horizontal axis.

Figure 14 shows the statistics of the maximum water path of each hydrometeor type during the simulation versus the ratio of cloud microphysics time steps to CTRL, plotted in the same manner as Fig. 9, except for the difference in the horizontal axis.

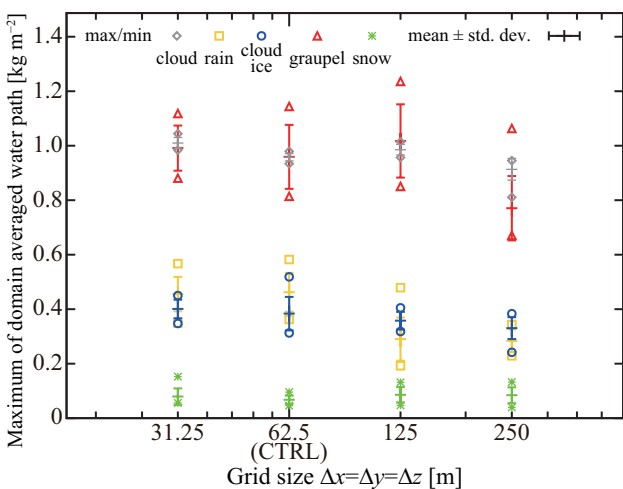

**Figure 11.** Statistics of the DX ensemble maximum water paths for each hydrometeor type. The horizontal axis represents the grid size $\Delta x = \Delta y = \Delta z$. This figure has the same form as Fig. 9, except for the horizontal axis.

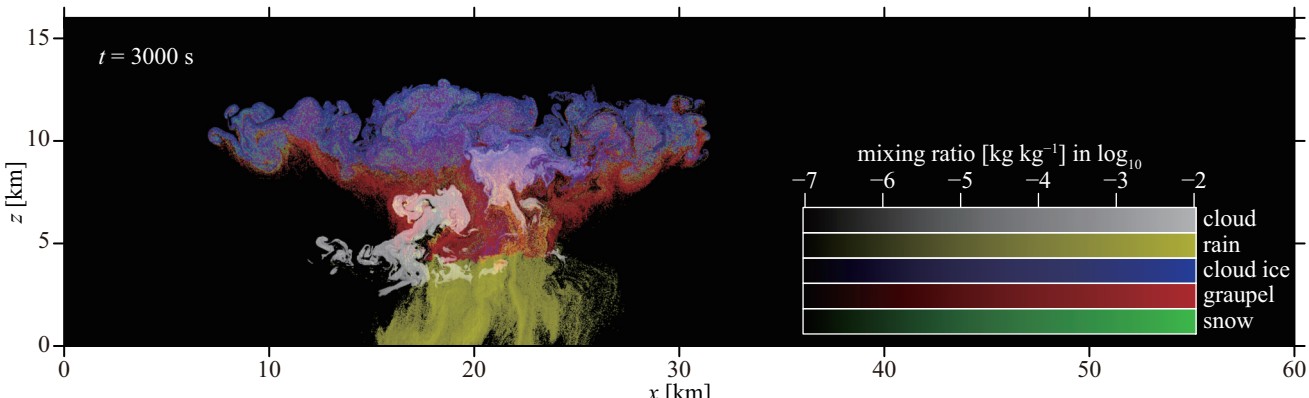

**Figure 12.** Spatial structure of the cloud at $t = 3000\,\mathrm{s}$ from a DX/2 ensemble member. This figure is formatted the same as Fig. 1, except for the grid resolution. See also Movie 6 in the Supplement.

Both figures show no significant difference among the five ensembles; therefore, we conclude that the numerical convergence with respect to the time steps is already attained at DTx1 (CTRL), i.e., $(\Delta t, \Delta t_{\mathrm{adv}}, \Delta t_{\mathrm{fz/mlt}}, \Delta t_{\mathrm{collis}}, \Delta t_{\mathrm{cnd/evp}}, \Delta t_{\mathrm{dep/sbl}})$ $= (0.4\,\mathrm{s}, 0.4\,\mathrm{s}, 0.4\,\mathrm{s}, 0.2\,\mathrm{s}, 0.1\,\mathrm{s}, 0.1\,\mathrm{s})$. Because DTx5 does not show any difference, time steps of five to ten times as large could suffice.

5    Further discussion of numerical convergence characteristics is provided in Sec. 8.4.

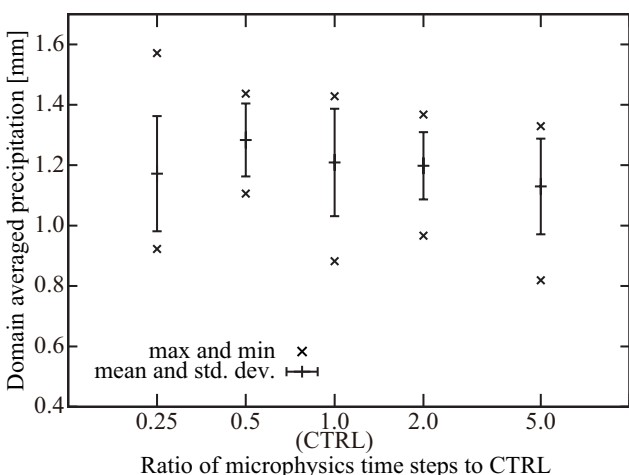

**Figure 13.** Statistics of DT ensemble accumulated precipitation amounts. The horizontal axis represents the ratio of cloud microphysics time steps to CTRL. This figure has the same form as Fig. 8, except for the horizontal axis.

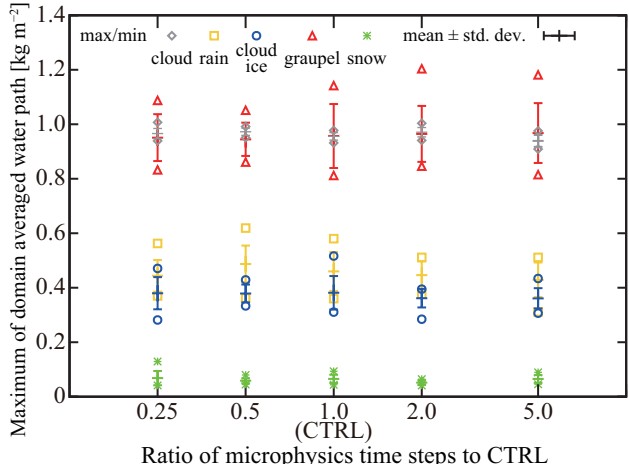

**Figure 14.** Statistics of the DT ensemble maximum water path for each hydrometeor type. The horizontal axis represents the ratio of cloud microphysics time steps to CTRL. This figure has the same form as Fig. 9, except for the horizontal axis.

## 8.4 Interpretation and computational cost

As confirmed in the preceding sections, the numerical parameters used for the CTRL ensemble (see Table 2) could produce an accurate numerical solution of the cumulonimbus case.

The CTRL ensemble's super-particle number concentration is $c^{\mathrm{SP}} = 128 \,/\mathrm{cell}$, which is comparable to various previous studies (e.g., Andrejczuk et al., 2010; Sölch and Kärcher, 2010; Riechelmann et al., 2012; Arabas and Shima, 2013; Unterstrasser and Sölch, 2014; Unterstrasser et al., 2017; Grabowski et al., 2018; Jaruga and Pawlowska, 2018; Dziekan et al., 2019; Hoffmann et al., 2019). Those studies reported that approximately $50$–$200 \,/\mathrm{cell}$ of super-particles are needed to accurately simulate clouds in two or three dimensions. If the number of attributes is increased, we generally need more super-particles to cover the higher dimensional attribute space. In this study, we used $5$ attributes to represent ice particles, which is relatively large compared to previous studies. Therefore, achieving numerical convergence with a super-particle number concentration as low as $128 \,/\mathrm{cell}$ is a remarkable result, revealing the efficacy of a particle-based cloud modeling approach. Another example of studies using many attributes is Jaruga and Pawlowska (2018), which included $8$ attributes to study aqueous-phase oxidation of sulfur to sulfate and confirmed that the results do not change significantly if the number concentration of super-droplets is larger than $64 \,/\mathrm{cell}$. However, the readers must be warned that the performance is sensitive to how super-particles are initialized (Unterstrasser et al., 2017), as we discussed in Sec. 5.3.

The CTRL ensemble's grid size is $\Delta x = \Delta y = \Delta z = 62.5 \,\mathrm{m}$, which is highly dependent on the simulated cloud's energy injection scale. As we will discuss in Sec. 9.3.10, introducing SGS turbulence models should improve the grid convergence characteristics.

The time steps for cloud microphysical processes used in the CTRL ensemble are $(\Delta t_{\mathrm{adv}}, \Delta t_{\mathrm{fz/mlt}}, \Delta t_{\mathrm{collis}}, \Delta t_{\mathrm{cnd/evp}}, \Delta t_{\mathrm{dep/sbl}}) = (0.4\,\mathrm{s}, 0.4\,\mathrm{s}, 0.2\,\mathrm{s}, 0.1\,\mathrm{s}, 0.1\,\mathrm{s})$. As shown in Sec. 8.3, time steps as large as five to ten times could suffice. In the following section, we discuss how those time steps are determined and whether the constraints could be relaxed.

To accurately trace the flow of moist air, $\Delta t_{\mathrm{adv}}$ should be limited by the CFL condition of wind velocity.

To avoid a sudden release of latent heat, $\Delta t_{\mathrm{fz/mlt}}$ must also be restricted through the CFL condition.

$\Delta t_{\mathrm{collis}}$ is the time step of coalescence, riming, and aggregation. As shown in Shima et al. (2009) and clarified in Sec. 5.5.5, $\Delta t_{\mathrm{collis}}$ can be estimated from the number concentration and size of real particles, and that $\Delta t_{\mathrm{collis}}$ does not depend on the numerical parameters such as super-particle number concentration or grid size. To make the calculation robust to larger time steps, a technique to allow multiple coalescence, riming, and aggregation occurrences is implemented in the SDM (see Sec. 5.5.5); however, this does not work properly if the collected super-particle's multiplicity is not sufficiently large. Multiple coalescence/riming/aggregation of collector particles would not be an accurate approximation either. These issues could be improved by introducing a recursive algorithm (Okawa, 2015), which could allow us to use larger $\Delta t_{\mathrm{collis}}$ values.

$\Delta t_{\mathrm{cnd/evp}}$ and $\Delta t_{\mathrm{dep/sbl}}$ are determined by the phase relaxation time of supersaturation, $\tau_{\mathrm{phase}} \propto 1/\sum \xi_i r_i$ (e.g., Squires, 1952). The time scale of CCN activation/deactivation is normally much smaller than the phase relaxation time; however, our model is not constrained by the activation/deactivation time scale because the condensation and evaporation equation (8) is solved implicitly (see Sec. 5.5.3). However, we explicitly calculate the exchange of vapor and latent heat with moist air (see

Sec. 5.5.3), thus, $\Delta t_{\mathrm{cnd/evp}}$ and $\Delta t_{\mathrm{dep/sbl}}$ must be smaller than the phase relaxation time. Otherwise, numerical instability occurs (Árnason and Brown, 1971). This restriction could be relaxed if we fully implicitly solve this coupled process of droplets and moist air. Perhaps the approach described in Sec. 2.6 of Grabowski et al. (2018) for mitigating spurious cloud-edge supersaturations could also be used for this purpose.

We used the first-order operator splitting scheme to separate the calculation (Table. 1). Employing higher-order operator splitting and/or tendencies would also improve numerical convergence characteristics.

    Lastly, we discuss SCALE-SDM's actual computational cost. Calculating one realization of the CTRL case required approximately 10 hours using 160 Intel Xeon E5-2650v3 CPU cores. To compare computational cost, we also tried the two-moment bulk scheme of Seiki and Nakajima (2014) implemented on SCALE. This took approximately 20 min, which is about 30 times

10 faster than SDM. As SDM's computational cost scales linearly with the number of super-particles and the number concentration of super-particles for the CTRL case was $128/\mathrm{cell}$, it is a plausible result. We can solve the same mathematical model using a multi-dimensional bin scheme. Let us also estimate the bin model's computational cost. The effective number of attributes we used for ice particles is 5, implying that the bin space is 5-dimensional. If we assume that 10–100 bins are needed for each axis, the total number of bins becomes $10^5$–$100^5$. For the binary collision calculation, most bin models assess all the

15 combinations of the bins. In this case, the computational cost scales with the square of the number of bins, i.e., $10^{10}$–$100^{10}$. However, we can reduce the cost of bin models by introducing a collision pair number reduction technique similar to that of SDM (Sato et al., 2009). Therefore, if we enhance the efficiency by using this algorithm, the computational cost of bin models scales linearly with the number of bins, i.e., $10^5$–$100^5$. However, this is still much larger than 100, i.e., the computational cost of SDM.

In SCALE-SDM, super-particles are distributed all over the simulated domain. If we use super-particles only inside the clouds by employing, e.g., the Twomey super-droplet methodology (Grabowski et al., 2018), computational costs could be considerably reduced.

## 9   Improvement of the model

Results of the typical realization of CTRL presented in Sec. 7 show that the life cycle of a cumulonimbus was successfully

simulated and the predicted mass- and velocity-dimension relationships agree fairly well with the existing formulas based on laboratory measurements and observations. At the same time, as indicated by the symbols in Figs. 1 and 4–7, our model produces several types of seemingly unrealistic ice particles. Another issue is the underestimation of columnar ice particle terminal velocities. As stated in Sec. 4.1.3, we did not properly implement the formula of Böhm (1989, 1992c, 1999) in our model. Further, not all of the elementary cloud microphysics processes critical for mixed-phase clouds are incorporated in our

model yet. In this section, we explore the possible improvements and further sophistication of the model.

## 9.1 Origins of odd particles

Let us determine the origins of the four types of odd particles denoted by the symbols in Figs. 1 and 4–7. Once determined, we then modify the time evolution equations to resolve three of the four issues in effect.

### 9.1.1 Long, slowly falling hailstones

The ice particle denoted by the circle at $t = 3000\,\mathrm{s}$ is an example of hailstones that are too long and slow-falling. The attributes related to this ice particle's morphology are $\{a, c, \rho^{\mathrm{i}}, m^{\mathrm{rime}}/m, n^{\mathrm{mono}}\} = \{2.58\,\mathrm{mm}, 36.1\,\mathrm{mm}, 1.12 \times 10^2\,\mathrm{kg\,m^{-3}}, 0.98, 213\}$. Therefore, this ice particle is categorized as a hailstone. However, the aspect ratio is $> 10$, which is unrealistically long for a hailstone. Because of the odd shape, its terminal velocity $v^{\infty} = 4.25 \times 10^{-1}\,\mathrm{m\,s^{-1}}$ is also much smaller than that of typical hailstones (Fig. 7). It is located at $(x, z) = (20.0\,\mathrm{km}, 4.11\,\mathrm{km})$, which is near the freezing level (see Fig. 1).

This odd particle was caused by a problem with the riming density formula (55)–(57). By analyzing this particle's history, we found that it was created by only a single riming event between a graupel particle and a similarly sized rain droplet. We can explain the mechanism as follows: Consider the riming of a quasi-spherical columnar graupel particle with a radius of $1\,\mathrm{mm}$ and a rain droplet with a radius slightly smaller than $1\,\mathrm{mm}$. Assume also that the ambient temperature is slightly lower than $0\,^{\circ}\mathrm{C}$. Then, from Eqs. (55)–(57), $\rho_{\mathrm{rime}} = 0.1\,\mathrm{g\,cm^{-3}}$. In other words, the apparent volume of the rimed rain droplet expands

10-fold. Because of the filling-in model we employed for riming outcome (see Sec. 4.1.10), the resultant ice particle became a long columnar hailstone: $(a, c) = (1\,\mathrm{mm}, 11\,\mathrm{mm})$.

    However, $\rho_{\mathrm{rime}} = 0.1\,\mathrm{g\,cm^{-3}}$ must be reconsidered. Equation (56) has a global maximum of approximately $0.95\,\mathrm{g\,cm^{-3}}$ at around $Y = 3.7$ and then quickly decreases, becoming $< 0.1\,\mathrm{g\,cm^{-3}}$ at around $Y = 5.5$. Then, from Eq. (55), $\rho_{\mathrm{rime}} = 0.1\,\mathrm{g\,cm^{-3}}$ for $Y > 5.5$. Consequently, considering the definition $Y := (-r_k v_{\mathrm{imp}}/T_j^{\mathrm{sfc}})/(\mathrm{\mu m\,ms^{-1}/^{\circ}C})$, $\rho_{\mathrm{rime}} = 0.1\,\mathrm{g\,cm^{-3}}$

frequently happens near the freezing level. For example, $Y = 1000$ for $r_k = 1\,\mathrm{mm}$, $v_{\mathrm{imp}} = 1\,\mathrm{ms^{-1}}$, and $T_j^{\mathrm{sfc}} = -1\,^{\circ}\mathrm{C}$. However, $\rho_{\mathrm{rime}}$ would be much larger and even closer to $\rho_{\mathrm{true}}^{\mathrm{i}}$ in such a situation in reality because the rimed droplet freezes slowly. Therefore, we argue that Eq. (56) is valid only up to $Y = 3.5$ and levels off after that. This correction can be made by replacing the $Y$ value in Eq. (56) (but not in Eq. (57)) with

$$Y^{\downarrow} = \min(Y, 3.5). \tag{107}$$

In Sec. 9.1.5, we will confirm that this correction eliminates those long hailstones (Figs. 15–18).

    Additionally, the same problem occurs if a quasi-spherical planar graupel particle and a slightly smaller rain droplet collide and rime near the freezing level. However, it is less evident than with the previous case because the equatorial radius grows as the square root of the volume (Eq. (53)). Regardless, this problem can also be addressed using the above correction.

### 9.1.2 Columns with steep mass-dimension relationship

The square at $t = 3000\,\mathrm{s}$ indicates another odd particle. If we look around the square in Figs. 4 and 5, we see that this particle belongs to a population of columnar cloud ice particles that have a steeper mass-dimension relationship than observed. The

attributes related to this cloud ice particle's morphology are $\{a, c, \rho^{\mathrm{i}}, m^{\mathrm{rime}}/m, n^{\mathrm{mono}}\} = \{24.9\,\mu\mathrm{m}, 138.8\,\mu\mathrm{m}, 269.7\,\mathrm{kg\,m^{-3}},$ $0.29, 1\}$. Its terminal velocity is $v^{\infty} = 1.46 \times 10^{-2}\,\mathrm{m\,s^{-1}}$ and it is located at $(x, z) = (20.3\,\mathrm{km}, 11.0\,\mathrm{km})$ (see Fig. 1).

As in the previous case, we found that a single riming event between a cloud ice particle and a cloud droplet followed by depositional growth created this columnar ice particle type. We can explain the mechanism as follows: Consider a quasi-spherical columnar ice particle with a radius of $10\,\mu\mathrm{m}$ and a supercooled droplet with a radius slightly smaller than $10\,\mu\mathrm{m}$. Assuming an impact velocity of $10^{-2}\,\mathrm{m\,s^{-1}}$ and ambient temperature of $-10\,^{\circ}\mathrm{C}$, then, from Eqs. (55)–(57), $Y = 10^{-2}$ and $\rho_{\mathrm{rime}} = 0.1\,\mathrm{g\,cm^{-3}}$. In other words, the apparent volume of the rimed droplet expands 10-fold and creates a columnar graupel particle: $(a, c) = (10\,\mu\mathrm{m}, 110\,\mu\mathrm{m})$ because of our riming outcome model's filling-in assumption. Then, through subsequent depositional growth, this columnar graupel particle turns back into a columnar cloud ice particle.

Contrary to the previous case, the low riming density is reasonable. Instead, we must reconsider the filling-in model. We assumed that the ice particle's maximum dimension is preserved. However, this is not realistic for riming between an ice particle and a similarly sized droplet, as our thought experiment revealed. Generalizing the idea, we consider that the frozen droplet's diameter is preserved if the diameter is larger than the ice particle's maximum dimension. That is, we propose to replace Eq. (50) with

$$a'_j = \max(a_j, r_k(\rho^{\mathrm{w}}/\rho_{\mathrm{rime}})^{1/3}), \tag{108}$$

and Eq. (54) with

$$c'_j = \max(c_j, r_k(\rho^{\mathrm{w}}/\rho_{\mathrm{rime}})^{1/3}). \tag{109}$$

In Sec. 9.1.5, we will confirm that those columns that follow an extremely steep mass-dimension relationship can be eliminated using this correction (Figs. 15–18).

### 9.1.3 Low-density hailstones

The cross at $t = 3000\,\mathrm{s}$ represents a hailstone with an extremely low apparent density. The attributes related to this hailstone's morphology are $\{a, c, \rho^{\mathrm{i}}, m^{\mathrm{rime}}/m, n^{\mathrm{mono}}\} = \{12.6\,\mathrm{mm}, 15.0\,\mathrm{mm}, 10.7\,\mathrm{kg\,m^{-3}}, 0.85, 1585116\}$. Its terminal velocity is $v^{\infty} = 4.31\,\mathrm{m\,s^{-1}}$ and it is located at $(x, z) = (10.6\,\mathrm{km}, 11.5\,\mathrm{km})$ (see Fig. 1). What is unusual here is the very low apparent density $\rho^{\mathrm{i}} = 10.7\,\mathrm{kg\,m^{-3}}$. This particle is composed of many monomers $n^{\mathrm{mono}} = 1585116$, and we set the limiting value of aggregate density in Eq. (62) to $\rho^{\mathrm{i}}_{\mathrm{crt}} = 10\,\mathrm{kg\,m^{-3}}$. Thus, we can conclude that this hailstone is created by the repeated aggregation between graupel particles.

Lump graupel particles with apparent densities as low as $50\,\mathrm{kg\,m^{-3}}$ were reported in Locatelli and Hobbs (1974). Therefore, a hailstone with an apparent density of $10\,\mathrm{kg\,m^{-3}}$ is not extremely unrealistic. However, our aggregation model is crude. Following Morrison and Grabowski (2010), we assumed that collision-aggregation collection efficiency is a fixed constant of $E_{\mathrm{agg}} = 0.1$ regardless of morphology or temperature. Therefore, this could cause the accumulation of graupel particles near the limiting value $\rho^{\mathrm{i}}_{\mathrm{crt}}$ in Fig. 6. There should be further detailed investigation to assess our aggregation model's applicability to graupel particles. See also Sec. 9.3.8, which provides a discussion to refine our aggregation model.

### 9.1.4 Long graupel particles

The triangle at $t = 5400\,\mathrm{s}$ is an extremely long graupel particle with a low apparent density. The attributes related to this cloud ice particle's morphology are $\{a, c, \rho^{\mathrm{i}}, m^{\mathrm{rime}}/m, n^{\mathrm{mono}}\} = \{31.9\,\mu\mathrm{m}, 438.2\,\mu\mathrm{m}, 19.4\,\mathrm{kg\,m}^{-3}, 0.53, 26679\}$. Its terminal velocity is $v^{\infty} = 1.02 \times 10^{-3}\,\mathrm{m\,s}^{-1}$, it is located at $(x, z) = (29.7\,\mathrm{km}, 6.4\,\mathrm{km})$ (see Fig. 1), and the ambient temperature is

$T = -14.4\,^{\circ}\mathrm{C}$.

     The particle is created by a sublimation of a graupel particle. The inherent growth ratio $\Gamma(T)$ proposed by Chen and Lamb (1994a) was used to calculate the deposition and sublimation process as described in Sec. 4.1.7. $\Gamma(T) < 1$ if $T$ is in the range of approximately $[-20\,^{\circ}\mathrm{C}, -10\,^{\circ}\mathrm{C}]$ and $[-5\,^{\circ}\mathrm{C}, 0\,^{\circ}\mathrm{C}]$; therefore, in this temperature range, ice particles grow to become planar through deposition and shrink to become columnar by sublimation.

However, $\Gamma(T)$ was derived from measurements of depositional growth; hence, it is questionable whether it is applicable for sublimation. According to Harrington et al. (2019) and references therein, $\Gamma(T)$ should be considered as unity for sublimation,

$$\Gamma(T_i) = 1, \quad \text{for } dm_i < 0 \text{ (sublimation);} \tag{110}$$

thus, the aspect ratios of ice particles are preserved during sublimation.

In Sec. 9.1.5, we will confirm that those long graupel particles can be eliminated using this correction (Figs. 15–18).

### 9.1.5 Results after corrections

In the preceding sections, we proposed three corrections to the time evolution equations (Eq. (107)–(110)) to avoid the creation of ice particles with unrealistic morphologies.

     We incorporated the proposed corrections into our model to create a new revision, SCALE-SDM 0.2.5-2.2.1. To assess the

validity of these corrections, we conducted the same simulations as the typical realization of CTRL using the new model. By comparing these results (Figs. 15–18) to the original results (Figs. 4–7), we confirm that the three types of odd ice particles no longer exist, as we intended. See also Movies 7–11 in the Supplement. Note that we left the issue of low-density hailstones for future studies. These corrections have little effect on the overall cloud properties, i.e., spatial structure (Movie 7 in the Supplement), the time evolution of the water path (Fig. 19), and the accumulated precipitation amount (Fig. 20).

## 9.2 Fix of underestimated terminal velocities of columnar ice particles

As explained in Sec. 4.1.3, our model did not properly implement the ice particle terminal velocity formula of Böhm (1989, 1992c, 1999). In this section, we fix the problem and assess its impact on this study.

     Noting that the area ratio $q_i \leq 1$ always holds in our model, Böhm's formula $v_{\mathrm{Böhm}}^{\infty}(m_i, \phi_i, d_i, q_i; \rho_i, T_i)$ can be summarized as follows:

$$X = \frac{8 m_i g \rho}{\pi \mu^2 \max{(\phi_i, 1)} q_i^{1/4}}, \tag{111}$$

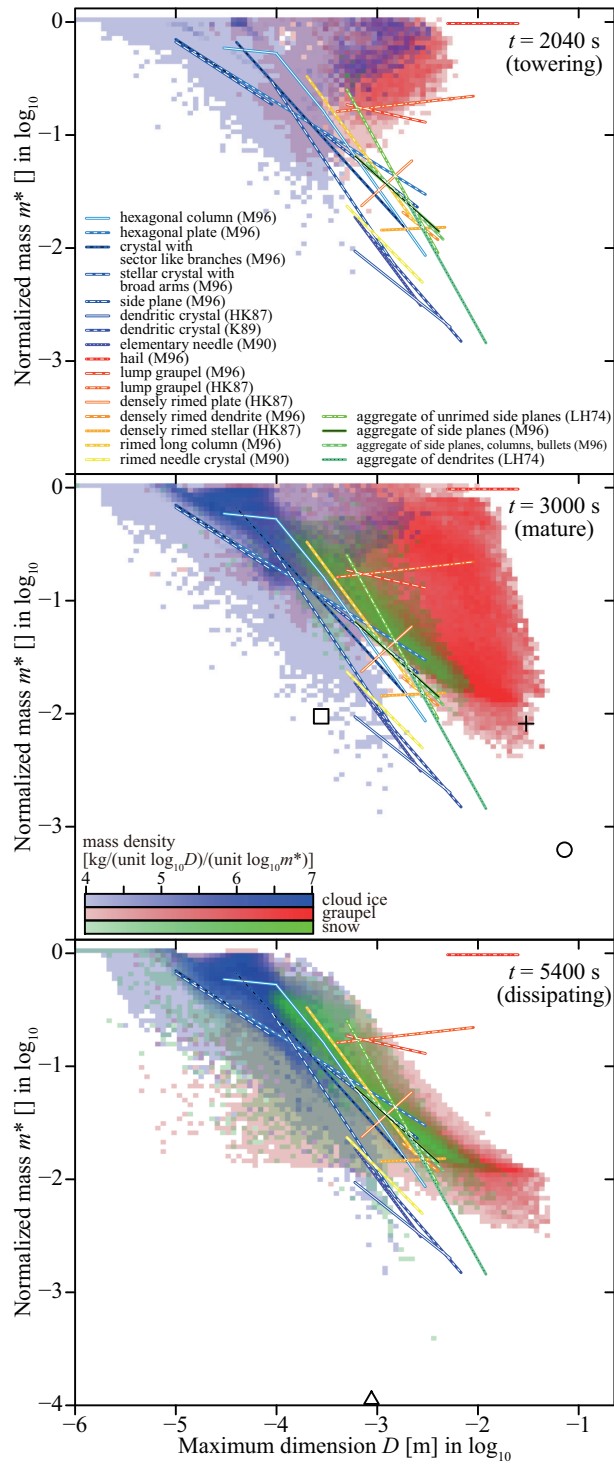

**Figure 15.** This figure is the same as Fig. 4 but shows results from SCALE-SDM 0.2.5-2.2.1, which incorporates the three corrections (107)–(110) proposed to avoid the creation of ice particles with unrealistic morphologies. See also Movie 8 in the Supplement.

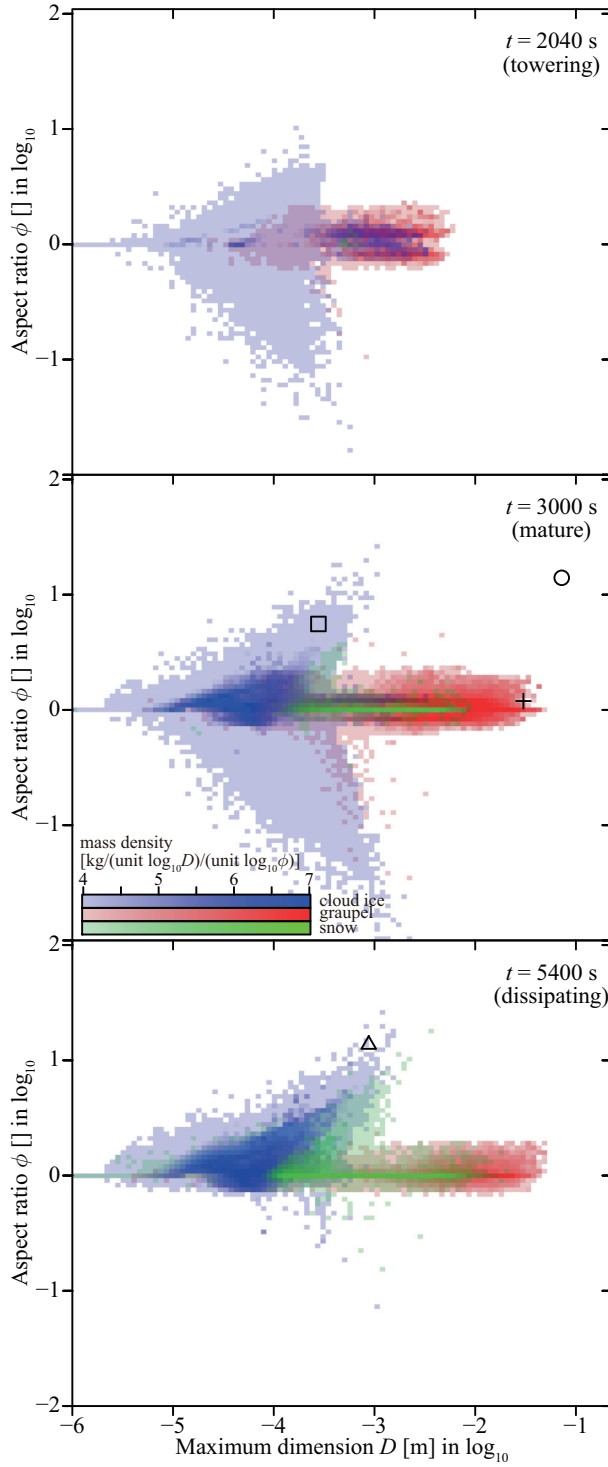

**Figure 16.** This figure is the same as Fig. 5 but shows results from SCALE-SDM 0.2.5-2.2.1, which incorporates the three corrections (107)–(110) proposed to avoid the creation of ice particles with unrealistic morphologies. See also Movie 9 in the Supplement.

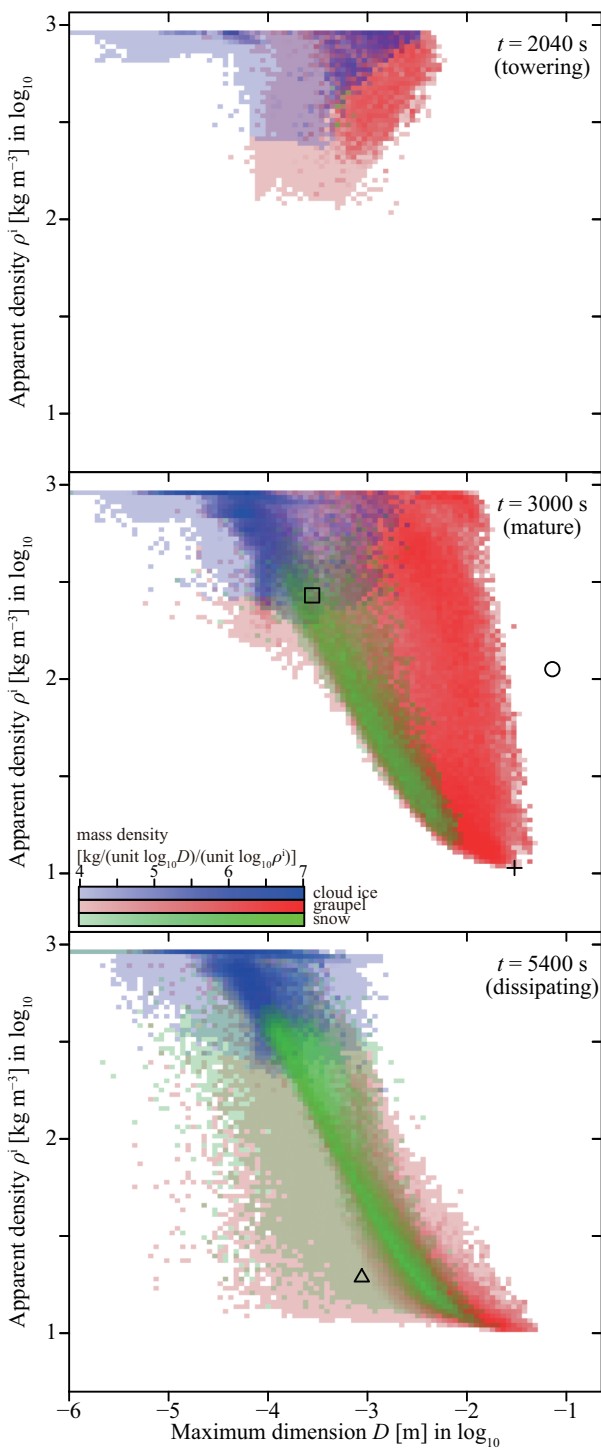

**Figure 17.** This figure is the same as Fig. 6 but shows results from SCALE-SDM 0.2.5-2.2.1, which incorporates the three corrections (107)–(110) proposed to avoid the creation of ice particles with unrealistic morphologies. See also Movie 10 in the Supplement.

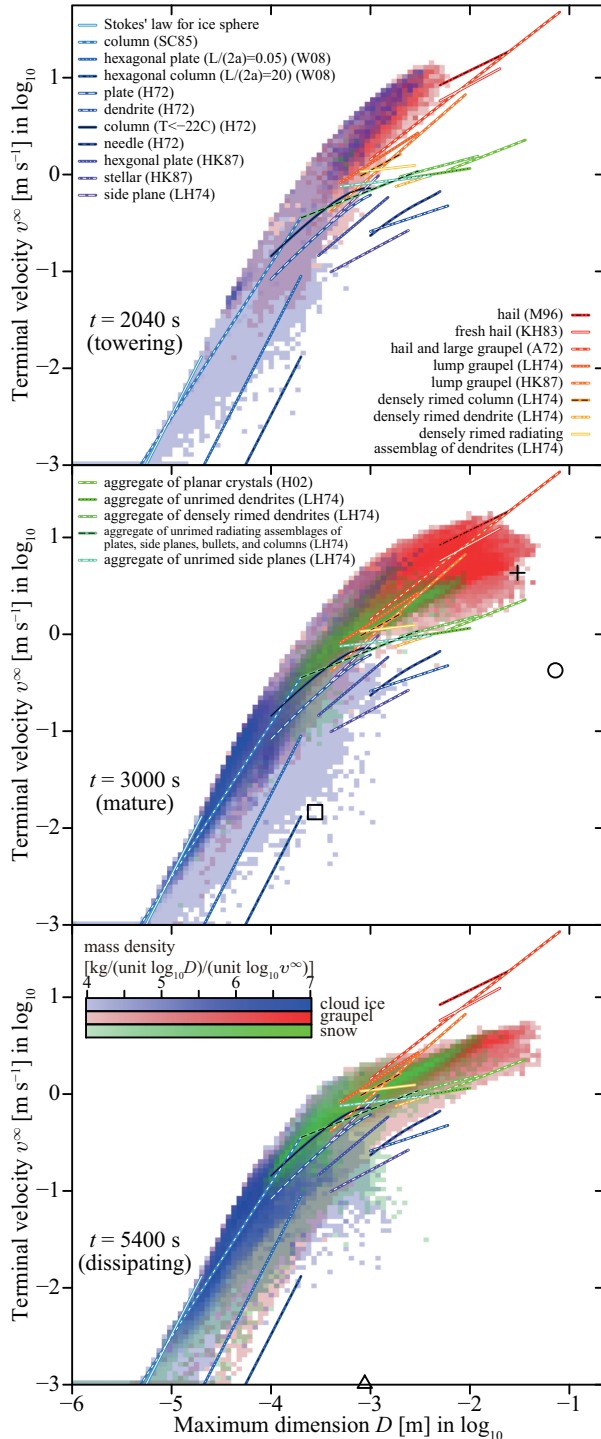

**Figure 18.** This figure is the same as Fig. 7 but shows results from SCALE-SDM 0.2.5-2.2.1, which incorporates the three corrections (107)–(110) proposed to avoid the creation of ice particles with unrealistic morphologies. See also Movie 11 in the Supplement.

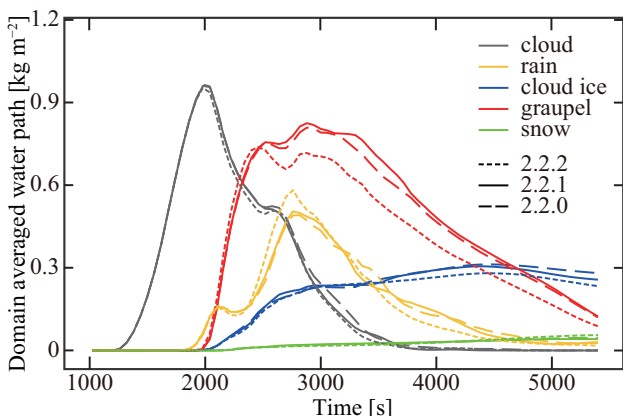

**Figure 19.** Changes in the domain-averaged water path before and after corrections. The long dashed, solid, and short dashed lines represent the SCALE-SDM 0.2.5-2.2.0, -2.2.1, and -2.2.2, respectively.

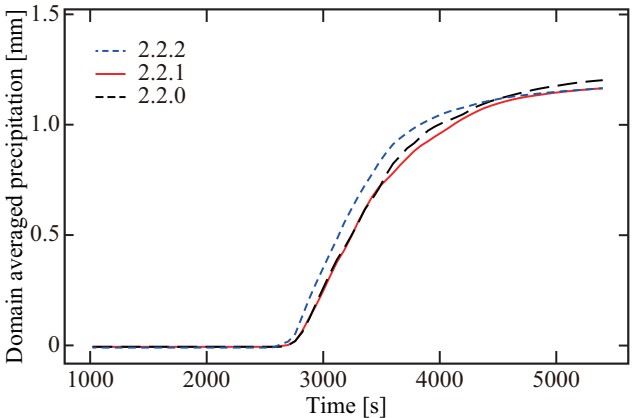

**Figure 20.** Changes in accumulated precipitation amounts before and after corrections. The long dashed, solid, and short dashed lines represent the SCALE-SDM 0.2.5-2.2.0, -2.2.1, and -2.2.2, respectively.

$$X' = X \frac{1 + (X/X_0)^2}{1 + 1.6(X/X_0)^2}, \tag{112}$$

$$X_0 = 2.8 \times 10^6, \quad \text{for ice particles}, \tag{113}$$

$$k = \min \left\{ \max\left(0.82 + 0.18\phi_i, 0.85\right), \left(0.37 + \frac{0.63}{\sqrt{\phi_i}}\right), \right.$$
$$\left. \frac{1.33}{\max\left(\log \phi_i, 0\right) + 1.19} \right\}, \tag{114}$$

$$\Gamma = \max\left\{1, \min\left(1.98, 3.76 - 8.41\phi_i + 9.18\phi_i^2 - 3.53\phi_i^3\right)\right\}, \tag{115}$$

$$C_{\mathrm{DP}} = \max\left(0.292k\Gamma, 0.492 - 0.200/\sqrt{\phi_i}\right), \tag{116}$$

$$C_{\mathrm{DO}} = 4.5k^2 \max\left(\phi_i, 1\right), \tag{117}$$

$$\beta = \left[1 + \frac{C_{\mathrm{DP}}}{6k}\left(\frac{X'}{C_{\mathrm{DP}}}\right)^{1/2}\right]^{1/2} - 1, \tag{118}$$

$$\gamma = \frac{C_{\mathrm{DO}} - C_{\mathrm{DP}}}{4C_{\mathrm{DP}}}, \tag{119}$$

$$N_{\mathrm{Re}} = \frac{6k}{C_{\mathrm{DP}}} \beta^2 \left[1 + \frac{2\beta e^{-\beta\gamma}}{(2+\beta)(1+\beta)}\right], \tag{120}$$

$$v_{\mathrm{B\ddot{o}hm}}^{\infty} = \frac{\mu N_{\mathrm{Re}}}{\rho d_i}. \tag{121}$$

In our model (SCALE-SDM 0.2.5-2.2.0/2.2.1), we assumed that the characteristic length $d_i$ is given by the maximum dimension $D_i = 2\max\left(a_i, c_i\right)$, and the area ratio $q_i$ is given by the area ratio regarding the circumcircle $q_i^{\mathrm{cc}} = A_i/A_i^{\mathrm{cc}}$ (Eq. (4)). However, in Böhm's theory, they are defined by

$$d_i = 2a_i, \quad q_i = q_i^{\mathrm{ce}} = A_i/A_i^{\mathrm{ce}}, \tag{122}$$

i.e., for columnar particles, minor axis is used for $d_i$, and the area ratio regarding circumscribed ellipse is used for $q_i$. Figure 1 in Böhm (1989) suggests $q_i = q_i^{\mathrm{ce}}$. It is not clearly specified, but from the second equality of Eq. 17 in Böhm (1992c), we can confirm that $d_i = 2a_i$.

For planar ice particles ($\phi_i < 1$), $v_{\mathrm{B\ddot{o}hm}}^{\infty}(d_i = 2a_i, q_i = q_i^{\mathrm{ce}})$ and $v_{\mathrm{B\ddot{o}hm}}^{\infty}(d_i = D_i, q_i = q_i^{\mathrm{cc}})$ yield the same results, because $2a_i = D_i$ and $q_i^{\mathrm{ce}} = q_i^{\mathrm{cc}}$ hold. However, for columnar ice particles ($\phi_i > 1$), $v_{\mathrm{B\ddot{o}hm}}^{\infty}(D_i, q_i^{\mathrm{cc}})$ always underestimates the fall velocity. From Eqs. (111)–(121), we can derive $v_{\mathrm{B\ddot{o}hm}}^{\infty}(2a_i, q_i^{\mathrm{ce}})/v_{\mathrm{B\ddot{o}hm}}^{\infty}(D_i, q_i^{\mathrm{cc}}) = \phi_i^{3/4}$ for $X \ll 1$, and $v_{\mathrm{B\ddot{o}hm}}^{\infty}(2a_i, q_i^{\mathrm{ce}})/v_{\mathrm{B\ddot{o}hm}}^{\infty}(D_i, q_i^{\mathrm{cc}}) = \phi_i^{7/8}$ for $X \gg 1$. Therefore, if $\phi_i = 2$, the ratio $v_{\mathrm{B\ddot{o}hm}}^{\infty}(2a_i, q_i^{\mathrm{ce}})/v_{\mathrm{B\ddot{o}hm}}^{\infty}(D_i, q_i^{\mathrm{cc}})$ is in the rage of 1.68–1.83. If $\phi_i = 10$, the range is 5.62–7.50, and if $\phi_i = 20$, it is 9.46–13.75. We also confirmed that Böhm's original definition $v_{\mathrm{B\ddot{o}hm}}^{\infty}(2a_i, q_i^{\mathrm{ce}})$ agrees well with the formulas of Westbrook et al. (2008), and Heymsfield and Westbrook (2010) (see also Fig. R2-2 in the authors' response to anonymous referee #2).

Therefore, the correction (122) generally increases the fall speed of columnar ice particles, and the increase factor is larger for longer particles. Then, through the ventilation effects (11) and (15), the diffusional growth of columnar ice particles is enhanced. Owing to this mechanism, we observed the creation of extremely long ice particles with aspect ratio $\phi_i > 100$ if we

incorporate the correction (122) to SCALE-SDM 0.2.5-2.2.1. However, this is unrealistic. The maximum aspect ratio reported is approximately 30 in Auer and Veal (1970) (Fig. 12 therein), and 15.77 in Um et al. (2015). In nature, such an extreme-shaped ice particle would be shattered spontaneously or by collision. However, for the moment, we fix this issue in an ad-hoc way. We do not allow an ice particle to grow by diffusion slenderer than $\phi_i = 40$ by imposing a limiter to the effective inherent growth ratio $\Gamma^*$ as follows.

$$\Gamma^* = 1 \quad \text{for } dm_i \geq 0 \ \wedge \ \phi_i > 40. \tag{123}$$

We incorporated the corrections (122) and (123) into SCALE-SDM 0.2.5-2.2.1 to create a revision, SCALE-SDM 0.2.5-2.2.2. To assess the impact of these corrections, we conducted the same simulation as the typical realization of CTRL using the new model. We observed that the precipitation was developed a few minutes faster, but the total precipitation amount was almost the same as the previous versions (Fig. 20). Figure 19 compares the time evolution of water paths. Here, a noticeable decrease in the graupel water path can be observed, which is attributed to the increased fall speed of columnar graupel particles (i.e., densely rimed columns). This, in turn, increased the rainwater path. The time evolution of other hydrometeor water paths (cloud, cloud ice, and snow) was almost unchanged. Ice particle morphology distributions resemble closely to the previous results, except for the vanishing of cloud ice particles with relatively slow terminal velocities (Figs. R2-5 – R2-8 in the authors' response to anonymous referee #2. See also Movies 13–16 in the Supplement). The corrections also do not alter the spatial structure of the cloud (Movie 12 in the Supplement).

## 9.3 Further sophistication of the model

Our model is based on a kinetic description, i.e., individual dynamics of particles and their stochastic collisions. However, a quantitative understanding of mixed-phase cloud microphysics is a long-standing meteorological issue, and a kinetic description of mixed-phase cloud microphysics has not been established. Further, our model does not incorporate several elementary processes that are critical for mixed-phase clouds. In this section, we explore the possibilities of further refining and sophisticating our model. Readers can also refer to Chen and Lamb (1994a, b), Misumi et al. (2010), Hashino and Tripoli (2007, 2008, 2011a, b), Jensen and Harrington (2015), Sölch and Kärcher (2010), Brdar and Seifert (2018), and Seifert et al. (2019), as these are modeling studies closely relevant to our study.

### 9.3.1 Ice nucleation pathways

There are various ice nucleation pathways (e.g., Kanji et al., 2017); however, in this study, we only considered condensation/immersion freezing and homogeneous freezing, as these are the dominant mechanisms in mixed-phase clouds.

Based on the singular hypothesis (Levine, 1950), we considered that each insoluble particle has its own freezing temperature $T^{\mathrm{fz}}$ that can be determined by INAS formulas. In the model evaluation experiments, we assumed that ice nuclei consist of mineral dust and used the INAS formula of Niemand et al. (2012). Formulas from Wex et al. (2015) and Ullrich et al. (2017) can be used for biogenic substances and soot, respectively.

The singular hypothesis ignores the time dependence of ice nucleation; thus, we assumed that particles initiate freezing immediately after the temperature drops below $T^{\mathrm{fz}}$ and the ambient air becomes saturated over liquid water. However, the time dependence of ice nucleation could be critical for clouds with long lifetimes, known as the "stochastic hypothesis". The Soccer Ball Model of Niedermeier et al. (2011, 2014, 2015), which is based on classical nucleation theory, could be used to

5 incorporate time dependence. Then, instead of the freezing temperature $T^{\mathrm{fz}}$, the contact angle of the surface site $\theta$ must be treated as an attribute.

Note that our requirement that the ambient water vapor must be supersaturated over liquid water would be too restrictive for immersion freezing. Even under an unsaturated condition, it is reasonable to allow immersion freezing if the droplet is sufficiently large, for instance, larger than $1\,\mu\mathrm{m}$ in radius.

To express homogeneous freezing, we assigned a fixed freezing temperature of $T^{\mathrm{fz}} = -38\,^{\circ}\mathrm{C}$ to all the IN inactive particles and ignored the time dependence of ice nucleation. However, this is not appropriate for the homogeneous freezing of deliquescent aerosol particles because homogeneous ice nucleation is suppressed when solute concentration increases. Additionally, the time dependence of ice nucleation could be also critical because the probability that a droplet freezing homogeneously is proportional to the liquid water volume. These effects can all be incorporated using the model of Koop et al. (2000).

Condensation/immersion freezing of deliquescent IN particles can also be incorporated by considering the depression of the freezing temperature $T^{\mathrm{fz}}$ by the solute (see Wex et al., 2014, and references therein). Alternatively, a model based on classical nucleation theory proposed by Khvorostyanov and Curry (2004, 2005) can be used to incorporate time dependence.

The formation of ice directly from the vapor phase onto an IN particle is known as deposition freezing. This can be observed at $< -25\,^{\circ}\mathrm{C}$ and in air that is below water saturation. Marcolli (2014) suggested that the phenomena conventionally known

as deposition freezing could be reinterpreted as pore condensation and freezing. We can use the temperature-dependent and saturation-ratio-dependent INAS formula proposed by Steinke et al. (2015) to incorporate this process. Here, INAS density $n_{\mathrm{S}}$ is a function of $x_{\mathrm{therm}}$, and $x_{\mathrm{therm}}$ is a function of temperature $T$ and saturation ratio over ice $S^{\mathrm{i}}$. We can assign $x_{\mathrm{therm},i}$ to each particle as an attribute. We consider that freezing occurs when $x_{\mathrm{therm}}(T, S^{\mathrm{i}}) > x_{\mathrm{therm},i}$.

A crude model of pre-activation is incorporated in our model by inhibiting complete sublimation (see Eq. (14) and the expla-

25 nation that follows). Pre-activation denotes "the capability of particles or materials to nucleate ice at lower relative humidities or higher temperatures compared to their intrinsic ice nucleation efficiency after having experienced an ice nucleation event or low temperature before" (Marcolli, 2017). Intensive sophistication based on laboratory studies is required; however, particle-based models are suitable for exploring the atmospheric relevance of pre-activation. Conversely, one might want to switch off pre-activation in our model, which is possible by resetting the particles as deliquescent aerosol particles when complete

sublimation occurs.

Contact freezing is another ice nucleation mechanism in which solid particles can initiate freezing upon contacting the surface of a supercooled droplet. Contact freezing occurs at temperatures greater than that of the same particle immersed in a droplet (e.g., Shaw et al., 2005); therefore, it might also be relevant to mixed-phase clouds. To explain the scavenging of aerosol particles by droplets, we must consider Brownian diffusion and phoretic forces. This process can be incorporated into

35 our model by introducing the collision-coalescence kernels detailed in Sec. 17.4.2 of Pruppacher and Klett (1997). Then, based

on the results of Shaw et al. (2005), as suggested by Will H. Cantrell (2017, private comm.), contact freezing could be expressed by increasing the particle's $T^{\text{fz}}$ by $4.5\,^{\circ}\text{C}$ in each single particle-droplet collision event. Another possibility is using laboratory data from Niehaus et al. (2014), who measured the freezing efficiency of various insoluble particles. This can be interpreted as the probability that each particle-droplet collision results in a freezing event.

It is also known that the evaporation of a droplet could lead to inside-out contact freezing (e.g., Durant and Shaw, 2005); however, there are still substantial uncertainties.

### 9.3.2 Onset of melting

We assumed that ice particles start melting immediately after the ambient temperature reaches $> 0\,^{\circ}\text{C}$. However, evaporative cooling delays the melting onset. For example, at a relative humidity of $50\,\%$, melting starts at $+4\,^{\circ}\text{C}$. We can incorporate this

effect by considering ice particle surface temperatures, as discussed in Rasmussen and Pruppacher (1982).

### 9.3.3 Partially frozen/melted particles

After the onset of freezing or melting, we assumed that complete freezing/melting occurs instantaneously.

  However, as shown in Murray and List (1972), the freezing time of millimeter-sized droplets could be of the order of $100\,\text{s}$. We can explicitly incorporate this process using the time evolution equation summarized in Sec. 16.1.4 of Pruppacher and Klett

(1997), which is derived from a quasi-steady assumption of vapor and thermal diffusion around a partially frozen droplet.

  We also assumed that rimed supercooled droplets freeze instantaneously; however, wet growth of graupel particles is critical to accurately predict hailstone formation. We can use the model from Rasmussen and Heymsfield (1987) to incorporate the wet growth process.

  Depending on the relative humidity and warming rate, the melting time of spherical ice particles with radii of approximately

20 $300\text{–}400\,\mu\text{m}$ ranges between $20\text{–}70\,\text{s}$ (Rasmussen and Pruppacher, 1982). A large hailstone could escape complete melting and reach the ground. The shedding of droplets could also occur if a partially melted hailstone contains excess meltwater, which could affect the raindrop size distribution below the cloud. Partially melted snow aggregates could create a layer of stronger radar reflectivity below the melting level, known as the "bright band". We can explicitly incorporate these processes using the model summarized in Phillips et al. (2007).

Additionally, to complete the model, all other time evolution equations must be extended to make them compatible with partially frozen/melted particles, which would require some effort.

### 9.3.4 Condensation and evaporation

In SCALE-SDM, we assumed that water vapor's diffusivity in air and moist air's thermal conductivity in Eq. (9) are fixed constants, $D_{\text{v}} = 2.52 \times 10^{-5}\,\text{m}^2\text{s}^{-1}$ and $k = 2.55 \times 10^{-2}\,\text{Jm}^{-1}\text{s}^{-1}\text{K}^{-1}$, which are the values for $T = 20\,^{\circ}\text{C}$ and $p = 1000\,\text{hPa}$.

However, this approximation is erroneous, particularly because diffusivity $D_{\text{v}}$ is inversely proportional to pressure. In the case of the initial profile we used for model evaluation, $T = -44\,^{\circ}\text{C}$ and $p = 250\,\text{hPa}$ at $z = 10\,\text{km}$. Thus, $D_{\text{v}} = 6.08 \times 10^{-5}\,\text{m}^2\text{s}^{-1}$,

which is about 2.4 times larger than we assumed. The temperature and pressure dependence of water vapor's diffusivity in air $D_v$, and the temperature dependence of moist air's thermal conductivity $k$ must be considered. The formulas summarized in Sec. 13.1 of Pruppacher and Klett (1997) can be used.

We considered the ventilation effect for deposition and sublimation but not for condensation and evaporation, even though it also enhances the growth and evaporation of larger droplets. We can include this effect by using the model described in Sec. 13.2.3 of Pruppacher and Klett (1997).

For cloud droplets, we must also consider kinetic correction to $D_v$ and $k$. See, e.g., Sec. 13.1 of Pruppacher and Klett (1997) and Kogan (1991).

In our model, aerosol particle hygroscopicity is expressed by Raoult's law with the van't Hoff factor $i$ (Low, 1969); however, using the kappa parameterization of Petters and Kreidenweis (2007) would be more convenient.

### 9.3.5 Deposition and sublimation

There are many issues around $\Gamma(T)$, which represents the primary growth habit of ice crystals. Considering the amount of data used for the fitting, the proposed shape of $\Gamma(T)$ is subject to large uncertainties (see Fig. 3 of Chen and Lamb, 1994a). The applicable range is also unclear. We set $\Gamma(T) = 1$ for small ice crystals $D < 10\,\mu\text{m}$. As discussed in Sec. 9.1.4, $\Gamma(T) = 1$ should be used for sublimation (Harrington et al., 2019, and references therein). We might need to use some other form of $\Gamma(T)$ for graupel particles and snow aggregates. Connolly et al. (2012) had to adjust $\Gamma(T)$ somewhat arbitrarily to obtain a better agreement.

Further, as shown by Kumai (1982) and Bailey and Hallett (2004), at $T < -20\,^\circ\text{C}$ both plates and columns can be created at the same temperature depending on the saturation ratio over ice $S^i$, and polycrystals can also be created. Therefore, for $T < -20\,^\circ\text{C}$, $\Gamma$ might better be considered a function of both $T$ and $S^i$, and formation of polycrystals must be somehow incorporated into our model. We can employ the mathematical model from Hashino and Tripoli (2008), which extends Chen and Lamb (1994a)'s model to describe these behaviors.

Harrington et al. (2019) reformulated the model from Chen and Lamb (1994a), and their model does not rely on $\Gamma(T)$, predicting the aspect-ratio evolution using the "facet-based hypothesis". The model is as good as Chen and Lamb's original model at liquid saturation, and further, it can be applied to a wider range of environmental conditions, such as low supersaturation and low pressure. However, it is still unclear how well the model would work for polycrystals or irregular ice particles.

We used Chen and Lamb (1994a)'s deposition density formula; however, as discussed in Jensen and Harrington (2015), their formula does not capture the wind tunnel data of Takahashi et al. (1991) very well. Instead, Jensen and Harrington (2015) proposed a simple formula: $\rho_{\text{dep}} = \rho^i_{\text{true}}\Gamma(T)$ for $\Gamma < 1$; $\rho_{\text{dep}} = \rho^i_{\text{true}}/\Gamma(T)$ for $\Gamma > 1$. Their idea to relate deposition density $\rho_{\text{dep}}$ to axis growth ratio is plausible, but its dependence on $S^i$ is lost. Because $\rho_{\text{dep}}$ accounts for the secondary growth habit, dependence on $S^i$ must be reconsidered.

In our model, each ice particle is approximated by a porous spheroid $(a, c, \rho^i)$. We used spheroid capacitance $C(a, c)$ to evaluate $C$ in Eq. (11). However, the spheroid $(a, c)$ represents the ice particle's spatial extent, and it might have a more detailed internal structure, which is represented by the apparent density $\rho^i$. The actual ice particle capacitance also depends

on the internal structure. Westbrook et al. (2008) accurately calculated the capacitance of realistic ice particles by directly simulating the trajectories of diffusing water molecules. Thus, we can use their formulas to refine our model's accuracy. For example, they showed that the capacitance of snow aggregates can be approximated by $C = D/4$, which is half that of a sphere.

As with condensation and evaporation, we assumed that water vapor's diffusivity in air $D_{\mathrm{v}}$ and moist air's thermal conductivity $k$ in Eq. (12) are fixed constants, but this must be revised.

Demange et al. (2017) constructed a sophisticated phase field model for ice crystal growth that successfully reproduced the formation of diverse ice crystal shapes. This model could help us construct a more accurate kinetic description of the deposition and sublimation processes.

### 9.3.6 Coalescence

For the collision efficiency of collision-coalescence $E_{\mathrm{coal}}^{\mathrm{collis}}$, we used a modified table of Hall (1980) proposed in Seeßelberg et al. (1996) and Bott (1998). However, the table of Pinsky et al. (2001) is more comprehensive and reliable. It is based on numerical results, but supported by the laboratory experiments of Vohl et al. (2007). Another option is to use the formula of Böhm (1992b, 1999, 2004). It is interesting to note that Böhm's formula (1992b; 1999) predicts that the collision-coalescence kernel $K_{\mathrm{coal}}$ does not vanish for equal size droplets owing to wake capture effect, but caution must be taken because his theory has an error (Böhm, 2004).

We assumed that the coalescence efficiency is unity, $E_{\mathrm{coal}}^{\mathrm{coal}} = 1$, for simplicity; however, it can be much smaller than 1 for large droplets. Straub et al. (2010) proposed a simple formula based on their numerical results. We can also use the formula of Seifert et al. (2005), which compiles the formulas of Low and List (1982) and Beard and Ochs (1995).

### 9.3.7 Riming

For the collection efficiency of collision-riming $E_{\mathrm{rime}}$, when a large spherical ice particle collects a supercooled droplet, we used the formula from Beard and Grover (1974) with a mixed Froude number (Eqs.(40) and (41)). von Blohn et al. (2009) demonstrated that the formula underestimates the efficiency if the spherical ice particle is large, but Eq. (11) in their paper seems to be incorrect and thus, we did not consider this.

When a large droplet collects ice particles, we used the original formula from Beard and Grover (1974), approximating the ice particle as spherical. To consider the ice particle shape, we can use the formulas from Lew and Pruppacher (1983) for a large droplet collecting small columns, and Lew et al. (1985) for a large droplet collecting small planar crystals.

Beard and Grover (1974)'s formula is valid only for $p < 0.1$, where $p$ is the size ratio of the collector ice/droplet and collected droplet/ice. We forcibly applied the formula beyond this range, which increases the collection efficiency of riming between small similar size droplets and ice particles, as $E_{\mathrm{BG74}}(p, N_{\mathrm{Re}}, N_{\mathrm{St}}) \approx p^2/(1+p^2)$ for $N_{\mathrm{St}} \ll 1$. This must be corrected.

When an ice particle collects a droplet, we employed the filling-in model and preserved the ice particle's maximum dimension. However, if the collector is a snow aggregate, we should use the similarity model proposed by Seifert et al. (2019). Unrimed/rimed snow aggregates have fractal structures, and Seifert et al. (2019) found a universal self-similar relation in snow

aggregate growth through riming. The similarity model considers the maximum dimension's increase during the early stages of riming, which could lead to a more rapid ice particle growth due to riming.

### 9.3.8 Aggregation

We assumed that collision-aggregation's collection efficiency is given by a constant $E_{\mathrm{agg}} = 0.1$, following Morrison and Grabowski (2010), but this is a simplification. $E_{\mathrm{agg}}$ should be larger for large particles because of the interlocking mechanism, and near water-saturated conditions. $E_{\mathrm{agg}}$ can be decomposed into $E_{\mathrm{agg}} = E_{\mathrm{agg}}^{\mathrm{collis}} E_{\mathrm{agg}}^{\mathrm{stick}}$, where $E_{\mathrm{agg}}^{\mathrm{collis}}$ is collision efficiency and $E_{\mathrm{agg}}^{\mathrm{stick}}$ is sticking efficiency. For $E_{\mathrm{agg}}^{\mathrm{collis}}$, we can use the formula of Böhm (1989, 1992a, b, c, 1994, 1999, 2004). For $E_{\mathrm{agg}}^{\mathrm{stick}}$, Pruppacher and Klett (1997, Sec. 16.2) provides a simple formula that depends solely on temperature. The $E_{\mathrm{agg}}^{\mathrm{stick}}$ formula provided by Phillips et al. (2015) is physically based and should thus be more reliable.

Calculating the attributes of the resultant ice particles is also not easy. Let $(a', c', \rho^{\mathrm{i}\prime}, m^{\mathrm{rime}\prime}, n^{\mathrm{mono}\prime})$ be the ice particle created by the aggregation of $(a_1, c_1, \rho_1^{\mathrm{i}}, m_1^{\mathrm{rime}}, n_1^{\mathrm{mono}})$ and $(a_2, c_2, \rho_2^{\mathrm{i}}, m_2^{\mathrm{rime}}, n_2^{\mathrm{mono}})$. For rime mass and number of monomers, $m^{\mathrm{rime}\prime} = m_1^{\mathrm{rime}} + m_2^{\mathrm{rime}}$ and $n^{\mathrm{mono}\prime} = n_1^{\mathrm{mono}} + n_2^{\mathrm{mono}}$ hold. To determine the remainder, $(a', c', \rho^{\mathrm{i}\prime})$, specifying two out of the three attributes is sufficient because of the conservation of the total mass. In this study, as in the case of riming, we assumed that the filling-in model can be applied to aggregation, i.e., the maximum dimension is conserved and only the minor axis grows. Therefore, $D' = \max(D_1, D_2) = \max(a_1, c_1, a_2, c_2)$. However, one more attribute must be specified. In this study, instead of predicting minor axis growth, we predict the apparent density $\rho^{\mathrm{i}\prime}$ by introducing an intuitive model that considers the compaction of fluffy snowflakes. Consequently, the fractal dimension of the mass-dimension relationship of snow aggregates predicted by our model is close to 2 (see the green shade in Figs. 4 and 15), which agrees well with various previous studies (e.g., Brown and Francis, 1995; Heymsfield et al., 2010; Mitchell, 1996; Schmitt and Heymsfield, 2010).

However, the filling-in assumption is not valid for aggregation. Higuchi (1960) introduced a parameter called the separation ratio: $s := 2l/(D_1 + D_2)$, $s \in [0, 1]$, where $l$ is the horizontal distance between the centers of the two particles. For an aggregation between two planar ice particles, the resultant ice particle's maximum dimension can be evaluated by $D' = \max\{D_1, D_2, (1+s)(D_1 + D_2)/2\}$. Our model corresponds to the special case of $s = 0$, but it has been reported that $s \approx 0.5$–0.6 for two planar crystals and dendrites (Higuchi, 1960; Kajikawa and Heymsfield, 1989; Kajikawa et al., 2002), and $s \approx 0.9$ for spatial dendrites (Kajikawa et al., 2002). In contrast, $s = 0$ for columnar ice crystals can be justified from Kajikawa (1995)'s observation that two needles of similar sizes tend to attach with their centers close ($s \approx 0$) and a right angle between their polar axes (crossed adhesion). Notably, the cross adhesion displacement gives the largest possible volume $V_{\mathrm{max}}$, which we used to calculate the apparent density $\rho^{\mathrm{i}\prime}$ of the resultant ice particle by interpolation.

Another issue of the filling-in assumption is that it gradually makes snow aggregates quasi-spherical (see the green shades in Figs. 5 and 16). Measurements indicate that snow aggregates have an average aspect ratio of 0.6 (e.g., Korolev and Isaac, 2003) or smaller (Jiang et al., 2017).

Introducing the separation ratio $s$ in our model is straightforward and could improve our model's accuracy. In general, this tends to reduce the mass-dimension relationship's fractal dimension, and their aspect ratio. Locatelli and Hobbs (1974) reported

that aggregates of dendrites and aggregates of unrimed side planes had fractal dimensions of $1.4$ (plotted in Figs. 4 and 15), which is smaller than 2.

In our model, the apparent density $\rho^{i\prime}$ after aggregation is predicted by the formula given in Eq. (62). It is natural to assume that there is a lower limit of apparent density; however, this is a crude expression of the idea and requires further validation and improvement. Also note that a contact angle model was used in Chen and Lamb (1994b) and Hashino and Tripoli (2011a) to determine the resultant ice particle.

Several numerical models can create detailed 3D structures of snow aggregates consisting of primary ice crystals (e.g., Westbrook et al., 2004a, b; Maruyama and Fujiyoshi, 2005; Schmitt and Heymsfield, 2010). We can refine our aggregation outcome model by using the results of those more microscopic models that resolve snow aggregate structures. For example, Przybylo et al. (2019) and Dunnavan et al. (2019) intensively studied the geometry of aggregates using such numerical models.

### 9.3.9   Spontaneous/collisional breakup

Several mechanisms can induce the spontaneous/collisional breakup of hydrometeors. However, we did not consider any of them in the present study. In particular, rime splintering (Findeisen and Findeisen, 1943; Hallett and Mossop, 1974), and the collisional breakup of ice particles (Vardiman, 1978) are critical in mixed-phase clouds, as these processes are thought to be responsible for the large excess in the observed number concentration of ice particles to the number concentration of IN aerosol particles (e.g., Field et al., 2017).

First, a particle-based numerical algorithm for calculating spontaneous/collisional breakup processes has not yet been established. A simple strategy is to add more super-particles to the system when a breakup event occurs, but this could be computationally inefficient.

Mathematical models of spontaneous/collisional breakup processes are available from various studies. For the spontaneous breakup of rain droplets $> 6.5\,\mathrm{mm}$, we can use the mathematical model from Kamra et al. (1991). For the collisional breakup of droplets, the models compiled and compared in Prat et al. (2012) can be used. For the shedding of excess melt water, Phillips et al. (2007)'s model can be used. For rime splintering, the model summarized in Sec. 16.1.6 of Pruppacher and Klett (1997) can be used. Readers may also refer to Field et al. (2017) and the references cited therein. For the collisional breakup of ice particles, Phillips et al. (2017)'s model can be used.

### 9.3.10   Sub-grid scale turbulence

The grid size we tested for evaluating the model ranged from $31.25\,\mathrm{m}$ to $250\,\mathrm{m}$, and only flows that are larger than the chosen grid size can be resolved. A substantial portion of turbulence kinetic energy is accumulated in large scales, and small scale turbulence is mostly driven by large scale motions; therefore, SGS turbulence is of secondary importance to the phenomena. Nevertheless, SGS turbulence does affect moist air flow and atmospheric particle behavior. SGS turbulence should be appropriately incorporated to improve the model's grid convergence.

The Smagorinsky-Lilly model (Smagorinsky, 1963; Lilly, 1962; Brown et al., 1994; Scotti et al., 1993), which is already available in SCALE-SDM, can be used for the diffusion of moist air by SGS turbulence. However, we did not use it in this study because the model is designed for 3D turbulence.

SGS turbulence can enhance particle collision, which can be incorporated by using the collision kernels proposed in Wang et al. (2008), Onishi and Seifert (2016), and Chen et al. (2018). Particle velocity fluctuations due to SGS turbulence can be modeled as an Ornstein-Uhlenbeck process (e.g., Pope, 1994; Schilling et al., 1996; Grabowski and Abade, 2017). The fluctuation of supersaturation through eddy-hopping and entrainment can be considered by introducing a new stochastic attribute (Grabowski and Abade, 2017; Abade et al., 2018) or by applying the Linear Eddy model to particles (Hoffmann et al., 2019).

## 10   Conclusions

Using SDM, we constructed a detailed numerical model of mixed-phase clouds based on a kinetic description, and subsequently demonstrated that a large-eddy simulation of a cumulonimbus that predicts ice particle morphology without assuming ice categories or mass-dimension relationships is possible. Our results strongly support the particle-based modeling methodology's efficacy for simulating mixed-phase clouds.

In our model, ice particles are approximated by porous spheroids. The elementary cloud microphysics processes that the model considers include advection and sedimentation; immersion/condensation and homogeneous freezing; melting; condensation and evaporation including the activation and deactivation of CCNs; deposition and sublimation; and coalescence, riming, and aggregation. Moist air fluid dynamics is described using the compressible Navier–Stokes equation.

Our model successfully simulated the life cycle of a cumulonimbus, and the predicted mass-dimension and velocity-dimension relationships were comparable with existing formulas. Numerical convergence was achieved at a super-particle number concentration as low as $128\,/\mathrm{cell}$, which consumed 30 times more computational time than a typical two-moment bulk model. We then fixed several issues of the original model and developed two updated versions: SCALE-SDM 0.2.5-2.2.1 (fix of the odd ice particle creation) and SCALE-SDM 0.2.5-2.2.2 (fix of the underestimated columnar ice terminal velocity).

A more detailed evaluation of the model to explore the applicability of the new approach is an essential step forward. Our results strongly indicate that ice particle morphology can be predicted more accurately by further developing particle-based models. However, from this study, we cannot quantify the extent to which the refined representation of mixed-phase cloud microphysics could improve the predictability of mixed-phase clouds' macroscopic properties. Such proficiency can be addressed by conducting a thorough comparison with observations and other models.

In addition, further sophistication of the model is necessary. As discussed in Sec. 9.3, various elementary processes must be incorporated or refined in the model. In particular, rime splintering and the collisional breakup of ice particles are critical because these processes are thought to be responsible for secondary ice production. Therefore, establishing an accurate and efficient particle-based algorithm for spontaneous/collisional breakup is also crucial.

Particle-based model accuracy is more subject to cloud microphysics uncertainties than numerical errors. Therefore, a quantitative understanding of elementary cloud microphysics processes is becoming increasingly important. More laboratory, ob-

servational, and theoretical studies to advance our knowledge of cloud microphysics are desired in the future (Morrison et al., 2020). Additionally, we can go into a more microscopic description of cloud microphysics than kinetic description, i.e., to explicitly resolve droplet and ice particle shapes and deterministically consider their collisions (e.g., Demange et al., 2017; Wang and Ji, 2000; Westbrook et al., 2004a, b; Maruyama and Fujiyoshi, 2005; Schmitt and Heymsfield, 2010; Mazloomi
5  Moqaddam et al., 2015). Such model studies would also be useful for refining kinetic descriptions.

Our model's computational cost is at least one or two orders of magnitude larger than that of bulk models. To further accelerate calculation, the use of SGS models discussed in Sec. 9.3.10 is crucial. Further reduction of the computational cost could also be achieved by using the Twomey super-droplet methodology described in Grabowski et al. (2018); however, it is vital to introduce dynamic load balancing. The acceleration achieved by those improvements might be insufficient to allow
10  using particle-based cloud microphysics models in weather or climate models. Studies to construct a high-fidelity bulk model or another form of macroscopic cloud microphysics model must also be pursued (e.g., Noh et al., 2018; Morrison et al., 2020).

## Appendix A: List of symbols

Table A1 summarizes important variables used in this study.

**Table A1.** List of symbols

| Symbol | Description |
| --- | --- |
| $\boldsymbol{a}, \boldsymbol{a}_i$ | attributes of a particle |
| $a, a_i$ | equatorial radius of an ice particle |
| $a$ | coefficient of curvature term of Köhler curve |
| $A_i$ | projected area of a particle perpendicular to flow direction |
| $A_i^{\mathrm{cc}}$ | area of circumcircle of $A_i$ |
| $A_i^{\mathrm{ce}}$ | area of circumscribed ellipse of $A_i$ |
| $A_{\mathrm{g}}$ | geometric cross-sectional area |
| $A^{\mathrm{insol}}$ | surface area of an insoluble substance |
| $b$ | coefficient of solute term of Köhler curve |
| $b_1, b_2$ | constant for ventilation coefficients |
| $c, c_i$ | polar radius of an ice particle |
| $c_{\mathrm{pd}}, c_{\mathrm{pv}}, c_{\mathrm{p}}$ | isobaric specific heat of dry air, water vapor, and moist air; $c_{\mathrm{p}} := q_{\mathrm{d}} c_{\mathrm{pd}} + q_{\mathrm{v}} c_{\mathrm{pv}}$ |
| $c^{\mathrm{sulf}}, c^{\mathrm{dust}}$ | initial number concentration of ammonium bisulfate aerosol particles and mineral dust particles |
| $c^{\mathrm{SP}}$ | initial number concentration of super-particles |
| $C$ | electric capacitance of a spheroid |
| $C_{\mathrm{SC}}$ | Cunningham slip correction factor |
| $d^{\mathrm{dust}}$ | mineral dust particle diameter |
| $d_i$ | particle characteristic length |
| $D_i$ | particle maximum dimension |
| $D_{\mathrm{v}}$ | diffusivity of water vapor in air |
| $e, e_i$ | vapor pressure and ambient vapor pressure |
| $e_{\mathrm{s}}^{\mathrm{w}}, e_{\mathrm{s}}^{\mathrm{i}}$ | saturation vapor pressure over planar liquid water surface, over planar ice surface |
| $e_{\mathrm{s}i}^{\mathrm{w,eff}}$ | effective saturation vapor pressure with respect to droplet surface |
| $E_{\mathrm{coal}}, E_{\mathrm{rime}}, E_{\mathrm{agg}}$ | collection efficiencies of collision-coalescence, -riming, and -aggregation |

| | |
|---|---|
| $E_{\text{coal}}^{\text{collis}}, E_{\text{coal}}^{\text{coal}}$ | collision and coalescence efficiencies of coalescence; $E_{\text{coal}} = E_{\text{coal}}^{\text{collis}} E_{\text{coal}}^{\text{coal}}$ |
| $E_{\text{agg}}^{\text{collis}}, E_{\text{agg}}^{\text{stick}}$ | collision and sticking efficiencies of aggregation; $E_{\text{agg}} = E_{\text{agg}}^{\text{collis}} E_{\text{agg}}^{\text{stick}}$ |
| $\bar{f}_{\text{vnt}}, f_{\text{vnt}}$ | ventilation coefficients for mass growth rate, and axis growth rate |
| $\boldsymbol{F}_i^{\text{drg}}$ | drag force from moist air on a particle |
| $F_{\text{k}}^{\text{i}}, F_{\text{d}}^{\text{i}}, F_{\text{k}}^{\text{w}}, F_{\text{d}}^{\text{w}}$ | thermodynamic terms of a particle's diffusional growth |
| $g$ | Earth's gravity |
| $\boldsymbol{G}, \boldsymbol{G}_i, \boldsymbol{G}_{lmn}$ | state of moist air, state of ambient moist air, state of moist air at grid point $(l, m, n)$ |
| $i, j, k$ | index of particles or super-particles |
| $i_n^{\text{fz}}, i_n^{\text{mlt}}, i_n^{\text{rime}}$ | indices of the $n$-th frozen droplet, melted ice particle, and rimed droplet |
| $I_{\text{r}}(t), I_{\text{s}}(t)$ | set of all particle indices at time $t$, set of all super-particle indices |
| $I_\alpha$ | degree of a solute's ionic dissociation |
| $k$ | thermal conductivity of moist air, or viscous shape factor for $v_{\text{Böhm}}^\infty$ |
| $K, K_{\text{coal}}, K_{\text{rime}}, K_{\text{agg}}$ | collision-coalescence, -riming, and -aggregation kernels |
| $L_{\text{v}}, L_{\text{s}}\ L_{\text{f}}$ | latent heat of vaporization, latent heat of sublimation, and latent heat of fusion |
| $m, m_i$ | particle mass |
| $m^*$ | normalized ice particle mass |
| $m_{\text{min}}^{\text{i}}$ | arbitrary small mass |
| $m^{\text{rime}}, m_i^{\text{rime}}$ | ice particle rime mass |
| $m_\alpha^{\text{sol}}, m_{\alpha i}^{\text{sol}}$ | mass of a soluble substance contained in a particle; $\alpha = 1, \ldots, N^{\text{sol}}$ |
| $m_\alpha^{\text{insol}}, m_{\alpha i}^{\text{insol}}$ | mass of an insoluble substance contained in a particle; $\alpha = 1, \ldots, N^{\text{insol}}$ |
| $M_\alpha^{\text{sol}}$ | molecular weight of a solute |
| $n(\boldsymbol{a}, \boldsymbol{x}, t)$ | particle distribution function |
| $n^{\text{sulf}}(\log r_{\text{dry}}^{\text{sulf}}, T^{\text{fz}})$ | initial distribution function of ammonium bisulfate particles |

| | |
|---|---|
| $n^{\mathrm{mono}}, n_i^{\mathrm{mono}}$ | number of monomers of an ice particle |
| $n_{\mathrm{S}}(T)$ | ice nucleation active surface site (INAS) density |
| $N_{\mathrm{r}}(t), N_{\mathrm{s}}(t)$ | total number of particles at time $t$, total number of super-particles at time $t$ |
| $N_{\mathrm{r}}^{\mathrm{wp}}, N_{\mathrm{s}}^{\mathrm{wp}}$ | total number of particles accumulated over the whole period, total number of accumulated super-particles |
| $N_{\mathrm{mFr}}$ | mixed Froude number |
| $N_{\mathrm{Sc}}$ | Schmidt number |
| $N^{\mathrm{insol}}, N^{\mathrm{sol}}$ | number of insoluble substances, number of soluble substances |
| $N_{\mathrm{Re}i}^{\mathrm{i}}, N_{\mathrm{Re}i}^{\mathrm{clm}}, N_{\mathrm{Re}i}^{\mathrm{w}}$ | Reynolds number of an ice particle, of an ice particle based on the column width, and of a droplet |
| $N_{\mathrm{St}}^{\mathrm{i/w}}, N_{\mathrm{St}}^{\mathrm{w/i}}$ | Stokes impaction parameter when a droplet collects an ice particle and when an ice particle collects a droplet |
| $N^{\mathrm{sulf}}(r_{\mathrm{dry}}^{\mathrm{sulf}})$ | accumulated number of particles smaller than $r_{\mathrm{dry}}^{\mathrm{sulf}}$ per unit volume of air at $t = 0$ |
| $p$ | probability density |
| $p^{\mathrm{i/w}}, p^{\mathrm{w/i}}$ | $p^{\mathrm{i/w}} := r_j^{\mathrm{i}}/r_k, p^{\mathrm{w/i}} := r_k/r_j^{\mathrm{i}}$ |
| $P$ | probability |
| $P_{\mathrm{INia}}$ | probability that a mineral dust particle is IN inactive; $P_{\mathrm{INia}} := P(T^{\mathrm{fz}} \le -38\,^{\circ}\mathrm{C})$ |
| $P_{\mathrm{INia}}^{\mathrm{SP}}$ | fraction of super-particles used for IN inactive mineral dust particles |
| $P, P_i$ | pressure, ambient pressure |
| $P_0$ | reference pressure; $P_0 = 1000$ hPa |
| $P_{jk}$ | probability of collision-coalescence, -riming, and -aggregation |
| $q_i, q_i^{\mathrm{cc}}, q_i^{\mathrm{ce}}$ | area ratio, area ratio with respect to circumcircle, and area ratio with respect to circumscribed ellipse; $q_i^{\mathrm{cc}} := A_i/A_i^{\mathrm{cc}}$, $q_i^{\mathrm{ce}} := A_i/A_i^{\mathrm{ce}}$ |

| | |
|---|---|
| $q_{\mathrm{v}}, q_{\mathrm{d}}$ | specific humidity and mass of dry air per unit mass of moist air; $q_{\mathrm{v}} := \rho_{\mathrm{v}}/\rho$, $q_{\mathrm{d}} := \rho_{\mathrm{d}}/\rho$ |
| $r, r_i$ | radius of the volume-equivalent sphere of liquid water in a particle |
| $r_i^{\mathrm{i}}$ | radius of the volume-equivalent sphere of an ice particle; $r_i^{\mathrm{i}} := (a_i^2 c_i)^{1/3}$ |
| $r_{\mathrm{dry}}^{\mathrm{sulf}}$ | dry radius of the ammonium bisulfate component |
| $R_{\mathrm{d}}, R_{\mathrm{v}}, R$ | gas constants of dry air, vapor, and moist air; $R := q_{\mathrm{d}} R_{\mathrm{d}} + q_{\mathrm{v}} R_{\mathrm{v}}$ |
| $s$ | power-law exponent of area-dimension relationship |
| $s_{\mathrm{v}}, s_{\mathrm{s}}, s_{\mathrm{f}}$ | source terms by vaporization, sublimation, and fusion |
| $S_i^{\mathrm{w}}, S_i^{\mathrm{i}}$ | ambient saturation ratio over liquid water, over ice; $S_i^{\mathrm{w}} := e_i/e_{\mathrm{s}}^{\mathrm{w}}$, $S_i^{\mathrm{i}} := e_i/e_{\mathrm{s}}^{\mathrm{i}}$ |
| $t$ | time |
| $\Delta t$, $\Delta t_{\mathrm{adv}}$, $\Delta t_{\mathrm{fz/mlt}}$, $\Delta t_{\mathrm{cnd/evp}}$, $\Delta t_{\mathrm{dep/sbl}}$, $\Delta t_{\mathrm{collis}}, \Delta t_{\mathrm{dyn}}$ | common time step, time steps for advection of particles; freezing and melting; condensation and evaporation; deposition and sublimation; collision-coalescence, -riming, and -aggregation; and fluid dynamics |
| $t_n^{\mathrm{fz}}, t_n^{\mathrm{mlt}}, t_n^{\mathrm{rime}}$ | times of the $n$-th freezing event, melting event, and riming event |
| $T, T_i$ | temperature, ambient temperature |
| $T^{\mathrm{fz}}, T_i^{\mathrm{fz}}$ | particle freezing temperature |
| $T_{\mathrm{min}}^{\mathrm{fz}}, T_{\mathrm{max}}^{\mathrm{fz}}$ | $T_{\mathrm{min}}^{\mathrm{fz}} := -36\,^{\circ}\mathrm{C}$, $T_{\mathrm{max}}^{\mathrm{fz}} := -12\,^{\circ}\mathrm{C}$ |
| $T_i^{\mathrm{sfc}}$ | particle surface temperature |
| $\boldsymbol{U}, \boldsymbol{U}_i$ | wind velocity, ambient wind velocity; $\boldsymbol{U} = (U, V, W)$ |
| $\boldsymbol{v}, \boldsymbol{v}_i$ | particle velocity |
| $v_{\mathrm{imp}}$ | impact velocity |
| $v_i^{\infty}$ | particle terminal velocity |
| $V, V_i$ | ice particle apparent volume |

| | |
|---|---|
| $V_{\max}$ | largest possible volume |
| $\Delta V$ | well-mixed volume |
| $\boldsymbol{x}, \boldsymbol{x}_i$ | particle position |
| $\Delta x, \Delta y, \Delta z$ | grid size |
| $X$ | $N_{\mathrm{Sc}}^{1/3}(N_{\mathrm{Re}i}^{\mathrm{i}})^{1/2}$, or Davies (Best) number for $v_{\mathrm{B\ddot{o}hm}}^{\infty}$ |
| $Y, Y^{\downarrow}$ | $Y := -r_k v_{\mathrm{imp}}/T_j^{\mathrm{sfc}}$, $Y^{\downarrow} := \min(Y, 3.5)$ |
| $\hat{\boldsymbol{z}}$ | unit vector in the $z$ axis direction |
| $\alpha, \beta$ | index of aerosol substances |
| $\beta$ | power-law exponent of mass-dimension relationship, or auxiliary parameter for $v_{\mathrm{B\ddot{o}hm}}^{\infty}$ |
| $\gamma$ | constant for ventilation coefficients, coefficient of the artificial hyperdiffusion term, or auxiliary parameter for $v_{\mathrm{B\ddot{o}hm}}^{\infty}$ |
| $\Gamma(T), \Gamma^{*}$ | inherent growth ratio, effective inherent growth ratio; $\Gamma^{*} := \Gamma(T) f_{\mathrm{vnt}}$ |
| $\Gamma(\phi)$ | a function for $v_{\mathrm{B\ddot{o}hm}}^{\infty}$ |
| $\delta^{d}(\boldsymbol{x})$ | $d$-dimensional Dirac's delta function |
| $\theta$ | potential temperature of moist air; $\theta := T/\Pi$ |
| $\kappa$ | power exponent relating porosity to projected area |
| $\mu$ | dynamic viscosity of moist air |
| $\xi_i$ | super-particle multiplicity |
| $\Pi$ | Exner function of moist air; $\Pi := (P/P_0)^{R/c_{\mathrm{P}}}$ |
| $\rho, \rho_i$ | density of moist air, density of ambient moist air; $\rho := \rho_{\mathrm{d}} + \rho_{\mathrm{v}}$ |
| $\rho_{\mathrm{d}}$ | density of dry air |
| $\rho_{\mathrm{dep}}, \rho_{\mathrm{rime}}, \rho_{\mathrm{sbl}}$ | deposition, rime, and sublimation densities |
| $\rho_{\mathrm{v}}, \rho_{\mathrm{v}i}$ | vapor density, ambient vapor density |
| $\rho^{\mathrm{i}}, \rho_i^{\mathrm{i}}$ | ice particle apparent density |
| $\rho_{\mathrm{crt}}^{\mathrm{i}}$ | limiting value of the apparent density |
| $\bar{\rho}_{jk}^{\mathrm{i}}$ | volume weighted average density |
| $\rho_{jk}^{\mathrm{i,min}}, \rho_{jk}^{\mathrm{i,max}}$ | minimum and maximum possible apparent density |

| | |
|---|---|
| $\rho_{\mathrm{true}}^{\mathrm{i}}$ | ice crystal true density |
| $\rho_{\mathrm{v}i}^{\mathrm{sfc}}$ | vapor density at a particle surface |
| $\rho^{\mathrm{w}}$ | density of liquid water |
| $\phi, \phi_i$ | ice particle aspect ratio; $\phi := c/a$ |
| $\partial \cdot / \partial t|_{\mathrm{cm}}$ | coupling term from cloud microphysics to fluid dynamics of moist air |
| $'$ | prime denotes a resultant particle |

## Appendix B: List of abbreviations

Table B1 summarizes important abbreviations that are used in this study.

**Table B1.** List of abbreviations

| Abbreviations | Full form |
| --- | --- |
| A72 | Auer (1972) |
| ARM | atmospheric radiation measurement |
| BG74 | Beard and Grover (1974) |
| CCN | cloud condensation nuclei |
| CFL | Courant-Friedrichs-Lewy |
| CL94 | Chen and Lamb (1994a) |
| CRYSTAL-FACE | Cirrus Regional Study of Tropical Anvils and Cirrus Layers-Florida Area Cirrus Experiment |
| EM17 | Erfani and Mitchell (2017) |
| H02 | Heymsfield et al. (2002) |
| H72 | Heymsfield (1972) |
| HK87 | Heymsfield and Kajikawa (1987) |
| HP85 | Heymsfield and Pflaum (1985) |
| IN | ice nucleation |
| INAS | ice nucleation active site |
| K89 | Kajikawa (1989) |
| KH83 | Knight and Heymsfield (1983) |
| LH74 | Locatelli and Hobbs (1974) |
| M90 | Mitchell et al. (1990) |
| M96 | Mitchell (1996) |
| SC85 | Starr and Cox (1985) |
| SDM | super-droplet method |
| SGS | sub-grid scale |
| W08 | Westbrook et al. (2008) |

*Code and data availability.* The source code of SCALE-SDM 0.2.5-2.2.0, -2.2.1, and -2.2.2 are available from https://doi.org/10.5281/zenodo.3483650. All the data used for this study can be reproduced by following the instructions included in the above repository. The data are also deposited in local storage at the University of Hyogo in Kobe, Japan, and are available from the corresponding author upon request.

*Supplement.* The supplement related to this article is available online at: https://doi.org/10.5281/zenodo.3478207.

*Author contributions.* All the authors designed the model and numerical experiments. SS developed the model code and performed the simulations. SS prepared the manuscript with contributions from all co-authors.

*Competing interests.* The authors declare that they have no conflicts of interest.

*Acknowledgements.* S. Shima is grateful to Wojciech W. Grabowski, Sylwester Arabas, Dennis Niedermeier, Miklós Szakáll, Will H.
5   Cantrell, Yutaka Tobo, and Hugh Morrison for informative discussions. This research partly used the computational resources of the K computer provided by the RIKEN Center for Computational Science (R-CCS) through the HPCI System Research Project (Project ID: hp150153) and the Computing Resources for Enhancement (Project ID: ra001010), and FX10 provided by Kyushu University through the HPCI System Research Project (Project ID: hp140094, hp160132). This work was supported by JSPS KAKENHI Grant Number 26286089; MEXT KAKENHI Grant Number 18H04448; the joint research program of the Institute for Space-Earth Environmental Research, Nagoya
10   University; the Center for Cooperative Work on Computational Science, University of Hyogo; and the Department of HPC Support, Research Organization for Information Science & Technology (RIST) under the Optimization Support Program of the HPCI system. The authors would like to thank Enago (www.enago.jp) for the English language review.

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
