# Peer review of "Predicting the morphology of ice particles in deep convection using the super-droplet method: development and evaluation of SCALE-SDM 0.2.5-2.2.0, -2.2.1, and -2.2.2"

_Geoscientific Model Development, 2019_

## Referee Comment (RC1) · Anonymous Referee #1 · 16 Jan 2020

**Review of "Predicting the morphology of ice particles in deep convection using the super-droplet method: development and evaluation of SCALE-SDM 0.2.5-2.2.0/2.2.1" by Shima et al. (gmd-2019-294)**

The manuscript describes the foundation and implementation of a new mixed-phase cloud model based on the superdroplet approach, in which individual Lagrangian particles represent ensembles of real aerosols and hydrometeors (e.g., cloud droplets, ice crystals, graupel). In first applications, dimension-mass and dimension-velocity are compared to literature values, and the sensitivity of the model to three numerical parameters (number of Lagrangian particles, time step, grid spacing) is tested.

First of all, I like to highlight the author's work on the formulation and implementation of their model, which is the first fully Lagrangian mixed-phase model in the literature. Furthermore, I like to emphasize their thorough model description. However, this thoroughness makes the manuscript lengthy, and in some places even repetitive. Therefore, the manuscript feels more like a reference work than a scientific article. Accordingly, besides requesting some rewriting for tightening the manuscript, my only major concern is the lack of comparisons with other mixed-phase models or observational data as outlined below. After my concerns are addressed, I can fully support the manuscript's publication in Geoscientific Model Development.

**Major Comments**

*Comparison with other models or observational data:* While the comparison of dimension-mass and dimension-velocity with literature values is already a first step in the right direction, I miss a thorough comparison with other modeling approaches or observational data. Mixed-phase microphysics is a highly complex subject due to a large number of (partially unknown) processes and their (highly uncertain) representation. Comparing different models and integrating observations that include information of particle habits is therefore essential to confirm the applicability of this new approach, and to identify missing or inappropriately represented processes. However, this is probably not within the scope of the study but will be a mandatory part of the further development and institutionalization of this modeling approach.

*Combining sections for tightening the manuscript:* The structure of the manuscript is very clear, with dedicated sections for the description of the model equations (Sec. 4), their numerical implementation (Sec. 5), and potential changes and additions to these equations (Sec. 9.2). Similarly, modeling results are described in Secs. 7 and 8 and discussed in Secs. 9.1 and 9.3, respectively. While this (more traditional) separation has been used for a long time in scientific writing, more recent publications tend to combine these sections, especially results and their interpretation/discussion, which might increase readability and understanding.

While reading Sec. 4.1, which describes the basic microphysical processes and equations used in the model, I was missing a more rigorous discussion of the choices made, i.e., which processes are included, which are neglected, and the reason for this. Of course, these points are not missing, but they are primarily stated in Sec. 9.2 - 29 pages later. Therefore, I suggest combining Sec. 4.1 with Sec. 9.2.

Similarly, the results of Sec. 7 are discussed in Sec. 9.1, and the results of Sec. 8 are discussed in Sec. 9.3. Again with a large gap that interrupts individual lines of thought. Therefore, I suggest combining Sec. 7 with Sec. 9.1 and Sec. 8 with Sec. 9.3 to increase readability.

Note that I am not sure if a combination of the aforementioned sections can be successful. However, I feel that some tightening of the manuscript might help the reader to grasp the main ideas of the manuscript.

**Minor Comments**

P. 1, Il. 7 – 8: The manuscript does not really show that the results capture the characteristics of a real cumulonimbus since it lacks a comparison with observations. This statement is also made on p. 7, l. 2, p. 64, l. 23.

P. 4, I. 20: The truth (despite some rare analytical solutions) is usually not even known.

P. 4, Il. 25 - 28: This problem has been described nicely in a paper by Stevens and Lenschow (2001), which I suggest to cite.

P. 5, I. 3: I would write about "a macroscopic description of cloud microphysics" to avoid confusion with other macroscopic properties of clouds, e.g., cloud morphology.

P. 5, II. 5 – 6: I do not agree that "bulk models do not have a rigorous theoretical foundation". Their theory — although it is based on approximate or idealized droplet size distributions — condenses a lot of our knowledge on microphysical processes.

P. 5, l. 11: I recommend to state explicitly that "aspect ratio" refers to the ratio of the ice crystal axes.

P. 5, Il. 13 – 16: Diffusional growth, spontaneous breakup, and collisional breakup are also an inherent parts of liquid-phase microphysics.

P. 5, II. 23 – 25: If the development of bin and bulk models started in the 1950s, why do you only cite articles from  $\ge$  2015?

P. 6, II. 6 – 8: "[C]urse of dimensionality" needs to be explained in more detail or left out.

P. 6, l. 34 – p. 7, l. 1: You do not resolve fluid dynamics "fully explicitly" since you have a grid spacing that is much larger than the Kolmogorov lengthscale.

P. 7, Il. 1 - 2: You should address that two-dimensional turbulence is different from threedimensional turbulence.

P. 7, II. 8 – 9: A particle attribute is added to a bin model?

P. 7, II. 23 – 26: This paragraph feels like a repetition of p. 7, II. 1 - 4.

P. 8, Il. 3 – 7: It feels arbitrary that the particle position x and the attributes a are treated separately.

P. 8, Il. 21 – 22: These lines feel like they belong to Section 5.

P. 9, I. 1: Where does this equation come from? Is there a reference?

P. 12, II. 9 – 13: How do the same  $(\beta, s) = (2.22, 1.32)$  result in different  $\kappa = 0.375$  and 0.300?

P. 12, l. 19: While condensation freezing requires that the ambient water vapor is supersaturated with respect to liquid water, this is not necessary for immersion freezing.

Eq. (9): This equation is an approximation.

P. 14, II. 11 – 12: Where does the approximation  $C \approx (2a_i + c_i)/3$  come from?

P. 16, II. 14 - 16, Eq. (29): The text and the equation do not agree. While the text states that the rime mass fraction does not change during sublimation, the equation states that it does not change during deposition.

P. 16, ll. 25 – 26: Dziekan and Pawlowska (2017) wrote about this. I suggest citing them here.

P. 17, Il. 8 – 9: I strongly believe that your collision-coalescence representation also captures selfcollection, i.e., the collision and coalescence of two raindrops.

P. 17, II. 10 - 15: The collection efficiency is usually the product of the collision efficiency and the coalescence efficiency. The effects described here are only a part of the processes constituting the collision efficiency, e.g., I miss a discussion of the so-called wake effect that increases the collision efficiency of large droplets due to a reverse flow in a large droplet's wake. Moreover, by not explicitly

considering the coalescence efficiency, it is assumed to be unity. However, it can be significantly smaller than unity, reflecting that smaller droplets, although they collide with a larger droplet, might not coalesce, i.e., they may just bounce off. I suggest commenting on this simplification.

P. 17, II. 26 – 27: To what does "latter case" refer to in the last sentence?

Eq. (55): I believe  $T_s$  is not defined.

P. 24, II. 21 – 23: How do we know that these degrees of freedom are unnecessary? This is a highly interesting question. However, simulating a small number of superparticles instead of all real particles is usually a result of limited computing resources, and not a deliberate decision on how many degrees of freedom are necessary.

P. 25, II. 1 - 6, Eq. (82): This is a helpful equation and an interesting switch of perspective on superparticles, multiplicity, and probability. Intuitively, I might agree that both lines of the equation are equal. However, I am wondering if it is possible to derive the second line from the first in a mathematically exact way.

P. 28, II. 6 – 7: Equation (7) is stiff even without the curvature and solute terms.

P. 28, ll. 15 - 17: It might be necessary to state that the applied collision-coalescence algorithm is linearized, which circumvents the quadratic nature of the collision-coalescence process.

P. 28, I. 31: This has already been said on p. 26, II. 17 – 18.

P. 31, II. 15 – 16: Does this boundary condition result in a significant loss of (almost) weightless aerosols? Or does is only affect precipitating particles?

P. 31, II. 17 - 23: This heating is not applied to the surface but the air above. Does this heating start at the beginning of the simulation? The timing of the heating might have an important impact on the degree of turbulence in the simulation.

P. 32, II. 1 - 3: The vertical grid spacing is quite large. It is well known that a too large vertical grid spacing reduces the maximum supersaturation at cloud base (e.g., Morrison and Grabowski 2008). This, in turn, affects the activation of cloud droplets, leading to a smaller number of cloud droplets. Do you see a substantial change in the number of cloud droplets in the sensitivity studies conducted in Section 8.2?

P. 32, Eqs. (91): I am struggling to derive (91) from (84) and (86). Please add some more comments. The same applies to Eqs. (92) and (93).

Pp. 33 – 34: You state that "a 10-member ensemble of simulations" is calculated "by changing the pseudo-random number sequence" in on p. 33, l. 9. This statement is repeated four times on p. 33, l. 18, p. 33, l. 29, p. 34, l. 11, and p. 34, l. 13.

P. 34, II. 1 - 11: The tested microphysical timesteps are quite small compared to the literature that states timesteps that are typically between 1 and 10 s (especially in box calculations). Therefore, it might be worthwhile to add one or two additional simulations with even longer timesteps to explore the entire parameter space (e.g., DTx4 and DTx8).

Fig. 2: The caption is missing. I assume that this is a formatting error.

P. 43, II. 11 - 13, Sec. 9.1: Could some of the unrealistic hailstones and graupel particles be explained by the lack of an appropriate shedding formulation that would break these particles after some timesteps?

P. 44, Il. 8 – 9: I disagree. Figure 9 shows that a higher number of superparticles increases the fluctuation/standard deviation in precipitation.

P. 63, Il. 27 – 28: The typical citation for the phase relaxation timescale is Squires (1952).

P. 64, l. 8 - 10: The general estimate of computational costs and its comparison with a Eulerian bin model is legit. However, I feel uncomfortable with the estimate of the collision calculation. The

collision calculation is a quadratic problem in both Eulerian bin and Lagrangian schemes. The linearization of this problem applied in the Lagrangian cloud model might also have its caveats, and I believe that a similar implementation into a Eulerian bin model is possible. Accordingly, the last step from  $10^5 - 100^5$  to  $10^{10} - 100^{10}$  feels unjustified.

**Technical Comments**

P. 12, l. 23: I assume that "initiated" is better than "updated" here.

- P. 13, l. 14: Use a lower case "w" to start this line.
- P. 24, l. 4: Remove one "planets".
- P. 27, Il. 3 4: I believe it is a "predictor-corrector scheme" and not a "predictor-collector scheme".
- P. 35, l. 16: Replace "amount" with "number".
- P. 38, I. 3, I. 6: Replace "segments" with "slopes".

**References**

Stevens, B. and Lenschow, D.H., 2001. Observations, experiments, and large eddy simulation. *Bulletin of the American Meteorological Society*, *82*(2), pp. 283-294.

Dziekan, P. and Pawlowska, H., 2017. Stochastic coalescence in Lagrangian cloud microphysics. *Atmospheric Chemistry and Physics*, *17*(22), pp. 13509-13520.

Morrison, H. and Grabowski, W.W., 2008. Modeling supersaturation and subgrid-scale mixing with two-moment bulk warm microphysics. *Journal of the Atmospheric Sciences*, *65*(3), pp. 792-812.

Squires, P., 1952. The growth of cloud drops by condensation. I. General characteristics. *Australian Journal of Chemistry*, 5(1), pp. 59-86

---

## Referee Comment (RC2) · Anonymous Referee #2 · 21 Jan 2020

The manuscript describes a new Lagrangian particle-based microphysics model, its numerical implementation and first results for a 2D convective cloud. To my knowledge this is the first (fairly) complete implementation of the super-particle approach for mixed-phase clouds. In that sense the contribution is highly appreciated and is potentially a landmark paper for the cloud modeling community. The manuscript is impressive in many ways. First of all, because the model succeeds to predict the typical mass-dimension and velocity-dimension relationships of ice particles in the atmosphere based on physically reasonable and fairly simple assumptions. Second, it

proves that the Lagrangian particle approach is computationally feasible and affordable for mixed-phase convective clouds. Third, it outlines the necessary future development and research to make further progress in understanding the properties of ice particles in mixed-phase clouds.

The manuscript provides a detailed and clear model description of the microphysical assumptions and equations and their implementation. This is very valuable for the community and in the best spirit of GMD that models should be well documented and their assumptions and limitations should be openly discussed. Even more, also the model development process is documented in the sense that the step from version 2.2.0 to 2.2.1 is described in the manuscript. This is very enlightening, because the base version 2.2.0 has some quite noticeable deficiencies, which are removed in version 2.2.1 by making some fairly small adjustments and fixes to the microphysical equations. I think it should be highly appreciated that the authors present their work in this way instead of simply desribing the newest version 2.2.1.

As a results the manuscript is a mix of a model documentation, a progress report and a research plan. Hence, it lacks the conciseness and structure that one would expect from a more focused scientific paper. In my opinion this is not a problem at all, especially not in a journal like GMD. Therefore I would recommend to keep the structure pretty much as it is now and make only some minor changes to improve readability.

Major comments:

- I was quite confused when reading section 2 from 2.1 to 2.7, because I did not understand how you can ensure mass conservation with this set of prognostic attributes. It only became clear when I read section 2.8 and understood that there are no partially melted wet ice particles (yet) in the model. I would strongly recommend to move that statement from section 2.8 to section 2.1 that particles are either liquid (and fully described by radius $r$) or ice (and described by major and minor axes $a$ and $c$ and the density $\rho^i$).

[Figure]

- From Figure 8 and 19 I would conclude that snow (aggregates) is falling too fast in SCALE-SDM, i.e. the green data point to not coincide with the empirical relations for aggregates. Can you explain this bias in the fallspeed of snow? I think this should be discussed in the paper.

- Maybe related to that: Wouldn't it be more accurate to use an ellipse instead of the circumcircle for the area in Boehms formula (section 4.1.3, page 11, line 20)? Do you take into account the turbulence correction for large Reynolds numbers in Boehms equations? The latter is actually necessary to limit the fall speed of large aggregates and match the observed terminal fall speed of aggregates.

Minor comments:

- page 5, line 5-7: I agree that a rigorous theory for bulk models is still lacking, but it would nevertheless be appropriate to reference the review by Beheng (2010). This paper gives an overview of the steps that have been made towards such a theoretical foundation, at least for liquid clouds and rain.

Beheng, K.D. (2010). The Evolution of Raindrop Spectra: A Review of Microphysical Essentials. In Rainfall: State of the Science (eds F.Y. Testik and M. Gebremichael), Geophysical Monograph Series, Wiley.

- page 6, line 4: 'approximated by a histogram', here I would recommend to replace 'histogram' by 'finite volumes or finite differences'.

- page 5, line 8: 'breakdown of the Smoluchowski equation'. Not all readers might be familiar with the notion of the breakdown of the Smoluchowski equation. A reference other than Smoluchowski (1916) or an additional sentence would be helpful.

- page 9, section 2.7: It should be mentioned that the assumption that particles move at their terminal fall velocity is an approximation. In the framework of a Lagrangian particle model this can quite easily be improved by considering the adjustment towards the new terminal fall velocity, e.g., after a collision event (see e.g. Naumann and Seifert

2015).

- page 13, section 4.1.6: When I first read this paragraph I was surprised that the ventilation is missing and is not even mentioned. It would be good to mention this approximation already here and not only later in section 9.2.4.

- page 14, eq. (13): Why is the minimum mass $m_{min}^i$ necessary in this equation? Is this because homogeneously frozen droplets may not contain any insoluble aerosol mass and then you would eventually have a super-droplet with zero mass? Does that $m_{min}^i$-particle not grow immediately when it is advected into cold, ice-supersaturated conditions and produce unrealistic ice? It does remember its freezing temperature, but it is already ice and would therefore grow immediately when the environment is supersaturated with respect to ice. I don't understand how this is implemented.

- page 15, eq. (21): Why is it necessary to impose this explicit limit to water saturation? If water droplets are present, then the supersaturation should be limited to due the rapid condensational growth. If no water droplets are present and no CCN can be activated, then the limit to water saturation might be unphysical.

- page 15 and 16: For depositional growth it is assumed that particle are spherical for D smaller than 10 microns (top of page 15), but for sublimation it is assumed that particles become spherical only when smaller than 1 micron. Why this asymmetry/hysteresis?

- page 16, line 14: 'rime mass fraction does not change during sublimation'. According to equation (29) rime mass fraction does not change during deposition ($dm > 0$) and only change during sublimation ($dm < 0$). Do you mean 'rime mass fraction does only change during sublimation'.

- page 17, line 16: 'remove k from the system'. Do you remove the particle because you have not yet introduced the multiplicity in those equations? Isn't it confusing to give here a Monte-Carlo algorithm without multiplicity, which is (as I assume) not used in SCALE-SDM. Maybe it should be emphasized (again) that this is the underlying

theoretical model, but not the numerical implementation.

- page 21, line 16: Why $c_j + min(a_k, c_k)$? Shouldn't it be $c_j + max(a_k, c_k)$ for the longest possible minor axis?

- page 24, line 4: 'other planets planets'. Two times 'planets'.

- page 28, section 5.5.5: Would it be possible to discuss the time step of the Monte Carlo scheme in some more detail? Or is this basically the same argument as in Shima et al.(2009) on page 1313?

- page 27, line 3: 'predictor-collector', maybe 'predictor-corrector'?

- page 44, line 11-12: 'Figure 10 clearly indicates that the super-particle number concentration must be larger than 128/cell'. This is not obvious to me. From Figure 10 I would conclude that 64/cell or even 32/cell is actually fine. Can you explain how you determined the value of 128/cell.

- page 60, line 14: 'approximating the particle is spherical' -> 'as spherical'

- page 60 and elsewhere: I find collision-riming and collision-aggregation awkward wording. Riming and aggregation are always due to collisions. Hence, the prefix 'collision' is not necessary.

- page 60, line 25: First sentence of 9.2.7 'We assume that collision-riming's collection efficiency'. Should this read aggregation instead of riming?

- page 62, line 9: 'Seifert et al. (2005)'s model'. This is actually the Low and List (1982) breakup model combined with Beard and Ochs (1995) for small drops. Seifert et al. (2005) did not add anything new to the physics of the breakup process.

- page 62, line 13-15: I would recommend to delete the two sentences starting with 'On average,...'. This is very questionable, has not been shown in the paper and would, in my opinion, be just a compensation of errors. Such a compensating effect is not a good reason to ignore breakup processes.

---

## Author Comment (AC1) · 24 May 2020

The comment was uploaded in the form of a supplement:
https://www.geosci-model-dev-discuss.net/gmd-2019-294/gmd-2019-294-AC1-supplement.zip

---

## Author Response (AR1)

**Authors' response regarding the first revision**

Dear Editor(s),

Thank you for handling our manuscript. We also would like to thank the two reviewers for carefully reading the manuscript, and for their informative and insightful comments.

The manuscript has been revised in response to the comments raised by the referees. Additional changes are also made by the authors to improve the manuscript. The most substantial changes are summarized as follows:

- Suggested by anonymous referee #1, in order to extend the numerical convergence test regarding time steps, we have added two more ensembles DTx5 and DTx10. We found that there is no significant difference between DTx5 and CTRL, but DTx10 was diverged at around t=1200s due to a numerical instability. From these results, we added a remark that time steps five to ten times as large as CTRL could suffice.
- As anonymous referee #2 indicated, we found that Böhm's formula for ice particle terminal velocity was not properly implemented in our model. The fall speeds of columnar ice particles were underestimated, but we confirmed that this flaw causes only a minor impact on this study. Details of our assessment are now presented in Sec. 9.2 "Fix of underestimated terminal velocities of columnar ice particles". Here, we corrected the problem and created a new version SCALE-SDM 0.2.5-2.2.2. We then conducted the typical CTRL simulation, and confirmed that this correction does not change the results significantly.
- To increase the readability, we merged Sec. 8 (results of numerical convergence test) and Sec. 9.3 (discussion of Sec.8).

An itemized response to all comments raised by the referees is provided below. Explanation of the additional changes follows. All the changes made to the manuscript are detailed in the difference file diff.html.

We hope that you find our responses satisfactory and that the manuscript is acceptable for publication in Geoscientific Model Development.

Sincerely,
Shin-ichiro Shima

**Reply to the first reviewer**

Thank you for carefully reading the manuscript. We appreciate your informative and insightful comments. Below, we provided an itemized response to all the comments raised, with the original comments presented in blue. Please also see the revised manuscript and the difference file diff.html, which we will submit separately. All the changes made to the manuscript are detailed in diff.html.

**Major Comments**

1-1) Comparison with other models or observational data: While the comparison of dimension-mass and dimension-velocity with literature values is already a first step in the right direction, I miss a thorough comparison with other modeling approaches or observational data. Mixed-phase microphysics is a highly complex subject due to a large number of (partially unknown) processes and their (highly uncertain) representation. Comparing different models and integrating observations that include information of particle habits is therefore essential to confirm the applicability of this new approach, and to identify missing or inappropriately represented processes. However, this is probably not within the scope of the study but will be a mandatory part of the further development and institutionalization of this modeling approach.

We agree that detailed comparison with other models and observations is desired, but we would like to leave it for future studies. We have stressed this point by adding the following remarks to the conclusions.

> A more detailed evaluation of the model to explore the applicability of the new approach is an essential step forward. Our results strongly indicate that ice particle morphology can be predicted more accurately by further developing particle-based models. However, from this study, we cannot quantify the extent to which the refined representation of mixed-phase cloud microphysics could improve the predictability of mixed-phase clouds' macroscopic properties. Such proficiency can be addressed by conducting a thorough comparison with observations and other models.

1-2) Combining sections for tightening the manuscript: The structure of the manuscript is very clear, with dedicated sections for the description of the model equations (Sec. 4), their numerical implementation (Sec. 5), and potential changes and additions to these equations (Sec. 9.2). Similarly, modeling results are described in Secs. 7 and 8 and discussed in Secs. 9.1 and 9.3, respectively. While this (more traditional) separation has been used for a long time in scientific writing, more recent publications tend to combine these sections, especially results and their interpretation/discussion, which might increase readability and understanding.

While reading Sec. 4.1, which describes the basic microphysical processes and equations used in the model, I was missing a more rigorous discussion of the choices made, i.e., which processes are included, which are neglected, and the reason for this. Of course, these points are not missing, but they are primarily stated in Sec. 9.2 — 29 pages later. Therefore, I suggest combining Sec. 4.1 with Sec. 9.2.

Similarly, the results of Sec. 7 are discussed in Sec. 9.1, and the results of Sec. 8 are discussed in Sec. 9.3. Again with a large gap that interrupts individual lines of thought. Therefore, I suggest combining Sec. 7 with Sec. 9.1 and Sec. 8 with Sec. 9.3 to increase readability.

Note that I am not sure if a combination of the aforementioned sections can be successful. However, I feel that some tightening of the manuscript might help the reader to grasp the main ideas of the manuscript.

We prioritized the global structure of the manuscript, but we admit that this diminished the local flow to some extent. After careful consideration, we decided to change the structure slightly.

Firstly, we consider that Sec. 4.1 (model description) and Sec. 9.2 of the original manuscript (possible sophistication) (Sec 9.3 of the revised manuscript) should not be merged. This study's primary objective is to assess particle-based modeling methodology's capability to simulate mixed-phase clouds. Advancing our scientific understanding of mixed-phase cloud microphysics is beyond the scope of this study. Therefore, we believe it is better to keep Sec. 4.1 as concise as possible.

Secondly, we believe Sec. 9.1 (fix of odd particles) should be located together with Sec. 9.2 of the original manuscript (possible sophistication), but after the numerical convergence is assessed (Sec. 8).

Finally, we agree that the merge of Sec. 8 (numerical convergence test) and Sec. 9.3 of the original manuscript (discussion of Sec.8) increases readability.

As a result, the following is the table of contents of the revised manuscript. (Note that Sec. 9.2 has been added. We found a bug in our terminal velocity formula implementation and this issue is discussed in Sec. 9.2.)

Following the policy of GMD, we have added PDF bookmarks and removed the table of contents. To inform the overall structure of the manuscript to the readers, the following sentence is added to the end of Sec. 1

Note that a comprehensive table of contents is provided as PDF bookmarks.

A guide to the readers is added to the end of the first paragraph of Sec. 4 "Time evolution equations of mixed-phase clouds".

Our model is detailed; however, it still falls short in completely describing mixed-phase cloud microphysics. To keep the model description concise, discussions on the shortcomings and how to overcome them are left for Sec. 9.

**Minor Comments**

1-3)    P. 1, ll. 7 – 8: The manuscript does not really show that the results capture the characteristics of a real cumulonimbus since it lacks a comparison with observations. This statement is also made on p. 7, l. 2, p. 64, l. 23.

To clarify our argument, we have revised the manuscript as follows.

(Abstract)
old< To evaluate the model's performance, a 2D large-eddy simulation of a cumulonimbus was conducted, and the results well capture characteristics of a real cumulonimbus.

new> To evaluate the model's performance, a 2D large-eddy simulation of a cumulonimbus was conducted, and the life cycle of a cumulonimbus typically observed in nature was

successfully reproduced.

(P. 4, ll. 18-20 of the revised manuscript)
old< To evaluate our model's performance, we conduct a two-dimensional (2D) simulation of an isolated cumulonimbus, and find that the results well capture characteristics of a real cumulonimbus.

new> To evaluate our model's performance, we conduct a two-dimensional (2D) simulation of an isolated cumulonimbus, and find that our model well reproduces the life cycle of a cumulonimbus typically observed in nature.

(The first sentence of Sec. 9 "Improvement of the model")
old< Results of the typical realization of CTRL presented in Sec. 7 show that our model well captures characteristics of a real cumulonimbus.

new> Results of the typical realization of CTRL presented in Sec. 7 show that the life cycle of a cumulonimbus was successfully simulated and the predicted mass- and velocity-dimension relationships agree fairly well with the existing formulas based on laboratory measurements and observations.

(P. 67, l. 21)
old< Our model captured cumulonimbus characteristics well, ...

new> Our model successfully simulated the life cycle of a cumulonimbus, ...

To reinforce our statement, we have added the following discussions to Sec. 7.2 "Spatial structure of the cloud, water path, and precipitation amount"

(4th and 5th paragraphs of Sec. 7.2)
… At the same time, we also observed that convective cores near the homogeneous freezing level ($z \approx 9.3\,km$) containing high liquid water content were sustained until around $t = 5000\,s$. For example, at $t = 3300\,s$, we observed a liquid water content of $2.1\,g\,m^{-3}$ at $(x, z) = (21.8\,km, 9.8\,km)$, where $T = -37.5\,°C$. The existence of such a high supercooled liquid water content down to the homogeneous freezing limit $-38\,°C$ are frequently observed in deep convective clouds (Rosenfeld and Woodley, 2000). … The maximum updraft and downdraft speeds were $39.0\,ms^{-1}$ and $21.9\,ms^{-1}$, which were observed at $(t, x, z) = (2340\,s, 12.8\,km, 11.1\,km)$ and $(1620\,s, 9.5\,km, 4.1\,km)$, respectively.

Our model successfully simulated the life cycle of a cumulonimbus typically observed in nature (see, e.g., Chap. 8 of Cotton et al., 2010). …

1-4)    P. 4, l. 20: The truth (despite some rare analytical solutions) is usually not even known.

We agree that the true solution of cloud simulation is not known, but this fact is irrelevant to our statement here. Assume that the current cloud microphysics models are accurate enough. Then, they should produce similar results (from which we hopefully infer that they are a good approximation of the truth). However, the spread of current models is big. Therefore, we can conclude that there is a big uncertainty in current cloud microphysics models.

1-5)    P. 4, ll. 25 – 28: This problem has been described nicely in a paper by Stevens and Lenschow (2001), which I suggest to cite.

Thank you for letting us know about this interesting paper. We have added the following sentence to the end of the paragraph:

(P. 2, l. 9)
This general philosophy of simulation is well documented, e.g., in Stevens and Lenschow (2001).

1-6)    P. 5, l. 3: I would write about "a macroscopic description of cloud microphysics" to avoid confusion with other macroscopic properties of clouds, e.g., cloud morphology.

Revised as suggested.

1-7)    P. 5, ll. 5 – 6: I do not agree that "bulk models do not have a rigorous theoretical foundation". Their theory — although it is based on approximate or idealized droplet size distributions — condenses a lot of our knowledge on microphysical processes.

To clarify our argument, we have rephrased the part as follows.

(P. 2, ll. 14-21)
old< … They solve a mathematical model that is closed in lower moments of the distribution function of cloud droplets, rain droplets, and ice particle categories (e.g., total mass and total number of particles). Currently, bulk models do not have a rigorous theoretical foundation and must rely on empirical parameterizations. A more bottom-up approach to construct more accurate and reliable numerical models would thus be desired.

new> … They solve a mathematical model that is closed in the lower moments of the distribution function of cloud droplets, rain droplets, and ice particle categories (e.g., mass and number mixing ratios). The basic premise of bulk models is that the distribution function can be determined by the lower moments, but such a universal relationship is unknown. In other words, in bulk models, to predict the time evolution of a chosen set of moments, their time derivatives are approximated by some functions of the moments being predicted, but this is not generally possible (see, e.g., Beheng, 2010). It would be also informative to note the analogy and difference between the Navier–Stokes equation and bulk models (Morrison et al., 2020), which highlights the difficulty in deriving bulk models. Therefore, for cloud microphysics, a more bottom-up approach to construct more accurate and reliable numerical models would be desired.

1-8)    P. 5, l. 11: I recommend to state explicitly that "aspect ratio" refers to the ratio of the ice crystal axes.

We replaced the "aspect ratio" here by

ratio of the ice crystal's minor axis to the major axis (hereafter called "aspect ratio"),

1-9)    P. 5, ll. 13 – 16: Diffusional growth, spontaneous breakup, and collisional breakup are also an inherent parts of liquid-phase microphysics.

That is right. To be more accurate, we revised the sentence as follows:

(P. 2, ll. 27-29)
old< Mixed-phase cloud microphysics are far more complicated than those of liquid-phase clouds, with various ice crystal formation mechanisms, diffusional growth, diverse ice particle morphologies, ice melting and shedding, spontaneous breakup, riming and wet growth, aggregation, collisional breakup, and rime splintering at play

new> Mixed-phase cloud microphysics are far more complicated than those of liquid-phase clouds, with various ice crystal formation mechanisms, diffusional growth by deposition/sublimation, diverse ice particle morphologies, ice melting and shedding, riming and wet growth, aggregation, spontaneous/collisional breakup of ice particles, and rime splintering at play

1-10)   P. 5, ll. 23 – 25: If the development of bin and bulk models started in the 1950s, why do you only cite articles from ≥ 2015?

We have added the earliest works of bin models.

(P. 3, ll. 4-6)
The development of bin schemes started independently of bulk models in the 1950s (e.g., Mason and Ramanadham, 1954; Hardy, 1963; Srivastava, 1967). For a review, see, e.g., Khain et al. (2015), Khain and Pinsky (2018), Grabowski et al. (2019), and Morrison et al. (2020).

1-11)   P. 6, ll. 6 – 8: "[C]urse of dimensionality" needs to be explained in more detail or left out.

We removed it.

1-12)   P. 6, l. 34 – p. 7, l. 1: You do not resolve fluid dynamics "fully explicitly" since you have a grid spacing that is much larger than the Kolmogorov length scale.

We understand your point, but here, the term "fully explicitly" was used in a different sense; we intended to explain that a forward (i.e., explicit) temporal integration scheme is employed for both horizontal and vertical directions. To clarify this point, we rephrased the sentence as follows:

(P. 4, ll. 16-18)
old< Mixed-phase cloud microphysics are solved using the SDM and the fluid dynamics of moist air is solved fully explicitly using a finite volume method with an Arakawa-C staggered grid.

new> Mixed-phase cloud microphysics is solved using the SDM. The fluid dynamics of moist air is solved by adopting a forward temporal integration scheme to both horizontal and vertical directions using a finite volume method with an Arakawa-C staggered grid.

The term "fully explicitly" was used in two other locations, which we also rephrased or removed.

1-13)   P. 7, ll. 1 – 2: You should address that two-dimensional turbulence is different from three dimensional turbulence.

We have added the following sentence to the last paragraph of Sec. 7.2 "Spatial structure of the cloud, water path, and precipitation amount":

(P. 36, ll. 2-3)
At the same time, our results are limited because the simulation was conducted in 2D; the turbulence characteristics are different in 2D and 3D.

1-14)   P. 7, ll. 8 – 9: A particle attribute is added to a bin model?

Misumi et al. (2010) added one more dimension, ice volume, to the bin component. They did not use the term "attribute", but "bin component" and "particle attribute" are equivalent.

1-15)   P. 7, ll. 23 – 26: This paragraph feels like a repetition of p. 7, ll. 1 – 4.

This paragraph aims to highlight the novelty of this study compared to previous closely relevant studies, which we explained in the preceding paragraph. We admit that the second sentence is out of context and keep only the first sentence.

1-16)   P. 8, ll. 3 – 7: It feels arbitrary that the particle position x and the attributes a are treated separately.

Collision-coalescence/riming/aggregation kernel $K$ in Eq. (31) is a function of attributes $\vec{a}$, but not a function of particle position $\vec{x}$. Therefore, there is a reason to distinguish $\vec{a}$ and $\vec{x}$.

**1-17)** P. 8, ll. 21 – 22: These lines feel like they belong to Section 5.

Here, we are explaining how the freezing temperature attribute changes through coalescence, riming, or aggregation. We presume that you are indicating not Sec. 5 but Sec. 4. (Section 5 is devoted to numerical schemes and implementation, hence it is not suitable.) Following your suggestion, we moved the explanation to the Sec. 4.1.9 "Coalescence between two droplets".

> (P. 15, ll. 28-30)
>
> Here, we assumed that the resultant particle's $T_j^{\text{fz}\prime}$ is given by $\max(T_j^{\text{fz}}, T_k^{\text{fz}})$, i.e., the higher freezing temperature of the two constituent particles. We also assume that the same applies to riming and aggregation.

**1-18)** P. 9, l. 1: Where does this equation come from? Is there a reference?

By the definition of $P$ and $p$,

$$P(T^{\text{fz}} > T) = \int_T^{0\,°C} p(T')dT',$$

from which we can derive the first equality of Eq. (1). Substituting the relation

$$P(T^{\text{fz}} > T) = 1 - \exp[-A^{\text{insol}}n_{\text{S}}(T)], \qquad \text{R1-1}$$

(which is introduced just before Eq. (1)), we can derive the second equality of Eq. (1). Equation R1-1 is derived, e.g., in Eq. (5) of Niedermeier et al. (2015). For more detail of the derivation, see my note 150829.heterogeneous_freezing_and_Poisson.pdf. Here, between $S$, $\lambda(T)$, $\Lambda(T)$ in Eqs. (22) – (24) and the variables used in the manuscript, the following relations hold,

$$S = A^{\text{insol}}, \quad \lambda(T) = -\frac{dn_{\text{S}}}{dT}, \quad \Lambda(T) = A^{\text{insol}}n_{\text{S}}(T).$$

**1-19)** P. 12, ll. 9 – 13: How do the same $(\beta, s)$ = (2.22,1.32) result in different $\kappa$ = 0.375 and 0.300?

Thank you. We noticed that the $(\beta, s)$ in both cases were wrong. $(\beta, s)$ of CRYSTAL-FACE was corrected to (2.22,1.30), and that of ARM was corrected to (2.20, 1.25).

**1-20)** P. 12, l. 19: While condensation freezing requires that the ambient water vapor is supersaturated with respect to liquid water, this is not necessary for immersion freezing.

We agree. A proposition below is added to Sec. 9.2.1 "Ice nucleation pathways".

> (P. 61, ll. 11-13)
> Note that our requirement that the ambient water vapor must be supersaturated over liquid water would be too restrictive for immersion freezing. Even under an unsaturated condition, it is reasonable to allow immersion freezing if the droplet is sufficiently large, for instance, larger than 1μm in radius.

**1-21)** Eq. (9): This equation is an approximation.

In the revised manuscript, we explicitly mention that the equation is an approximation.

**1-22)** P. 14, ll. 11 – 12: Where does the approximation $C \approx (2a_i + c_i)/3$ come from?

$C$ is given by

$$
C(a,c) = \begin{cases} a\epsilon/\sin^{-1}\epsilon, & \text{if } \phi < 1 \text{ (oblate)}; \\ a = c, & \text{if } \phi = 1 \text{ (spherical)}; \\ c\epsilon/\log[(1+\epsilon)\phi], & \text{if } \phi > 1 \text{ (prolate)}. \end{cases}
$$

Here, ε is the eccentricity

$$
\epsilon = \begin{cases} \sqrt{1-\phi^2}, & \text{if } \phi < 1 \text{ (oblate)}; \\ 0, & \text{if } \phi = 1 \text{ (spherical)}; \\ \sqrt{1-(1/\phi)^2}, & \text{if } \phi > 1 \text{ (prolate)}. \end{cases}
$$

If we expand *C/a* in powers of ε, we can derive $C \approx (2a+c)/3$.

1-23) P. 16, ll. 14 – 16, Eq. (29): The text and the equation do not agree. While the text states that the rime mass fraction does not change during sublimation, the equation states that it does not change during deposition.

The definition of rime mass fraction is $m_{\mathrm{i}}^{\mathrm{rime}}/m_{\mathrm{i}}$, hence both the text and the equation are correct. To avoid confusion, we have clarified the definition of rime mass fraction.

1-24) P. 16, ll. 25 – 26: Dziekan and Pawlowska (2017) wrote about this. I suggest citing them here.

The size of well-mixed volume is discussed also in Shima et al. (2009) (see §3 of Sec. 2.1.4. "Stochastic coalescence of droplets"). We have cited both references in the revised manuscript

1-25) P. 17, ll. 8 – 9: I strongly believe that your collision-coalescence representation also captures selfcollection, i.e., the collision and coalescence of two raindrops.

That is right. We have added selfcollection to the list.

1-26) P. 17, ll. 10 – 15: The collection efficiency is usually the product of the collision efficiency and the coalescence efficiency. The effects described here are only a part of the processes constituting the collision efficiency, e.g., I miss a discussion of the so-called wake effect that increases the collision efficiency of large droplets due to a reverse flow in a large droplet's wake. Moreover, by not explicitly considering the coalescence efficiency, it is assumed to be unity. However, it can be significantly smaller than unity, reflecting that smaller droplets, although they collide with a larger droplet, might not coalesce, i.e., they may just bounce off. I suggest commenting on this simplification.

We admit our explanation was not precise enough. The corresponding part was revised as follows:

(P. 15, ll. 14-21)

…, which can be decomposed into $E_{\mathrm{coal}} = E_{\mathrm{coal}}^{\mathrm{collis}} E_{\mathrm{coal}}^{\mathrm{coal}}$. Here, collision efficiency $E_{\mathrm{coal}}^{\mathrm{collis}}$ considers the effect that a smaller droplet is swept aside by the flow around a larger droplet, or a droplet being caught in the wake of a similarly sized droplet collides on the downstream side. We adopt the collision efficiency used in Seeßlberg et al. (1996) and Bott (1998). Here, Davis (1972) and Jonas (1972) are used for small droplets, and Hall (1980) for larger droplets, with modifications to the collector droplet radius range $70\,\mu m - 300\,\mu m$ to incorporate the wake effect suggested by Lin and Lee (1975). Not all the collisions end up with coalescence. Rebound or breakup (fragmentation) could also occur. Coalescence efficiency $E_{\mathrm{coal}}^{\mathrm{coal}}$ represents the fraction of collisions that result in permanent coalescence. In this study, we assume $E_{\mathrm{coal}}^{\mathrm{coal}} = 1$ for simplicity.

In addition, to discuss a possible refinement, we added a new section as follows.

9.2.6 Coalescence

For the collision efficiency of collision-coalescence $E_{\text{coal}}^{\text{collis}}$, we used a modified table of Hall (1980) proposed in Seeßlberg et al. (1996) and Bott (1998). However, the table of Pinsky et al. (2001) is more comprehensive and reliable. It is based on numerical results, but supported by the laboratory experiments of Vohl et al. (2007). Another option is to use the formula of Böhm (1992b, 1999, 2004). It is interesting to note that Böhm 's formula (1992b, 1999) predicts that the collision-coalescence kernel $K_{\text{coal}}$ does not vanish for equal size droplets due to wake capture effect, but caution has to be taken because his theory has an error Böhm (2004).

We assumed that the coalescence efficiency is unity, $E_{\text{coal}}^{\text{coal}} = 1$, for simplicity, but it can be much smaller than 1 for large droplets. Straub et al. (2010) proposed a simple formula based on their numerical results. The formula of Seifert et al. (2005) can also be used, which compiles the formulas of Low and List (1982) and Beard and Ochs (1995).

1-27) P. 17, ll. 26 – 27: To what does "latter case" refer to in the last sentence?

"Latter case" refers to "the collection of small ice particles by a larger droplet".

1-28) Eq. (55): I believe $T$s is not defined.

Sorry, it is not $T_{\text{s}}$, but $T_j^{\text{sfc}}$ (the surface temperature of ice particle $j$).

1-29) P. 24, ll. 21 – 23: How do we know that these degrees of freedom are unnecessary? This is a highly interesting question. However, simulating a small number of superparticles instead of all real particles is usually a result of limited computing resources, and not a deliberate decision on how many degrees of freedom are necessary.

We admit that the concept of "unnecessary degrees of freedom" is abstract, and identifying them is not easy. Our explanation in the original manuscript might have been abrupt and misleading, so we rephrased the part as follows:

> (The first paragraph of Sec 5.2 "Super-particles and real particles")
> old< There are many particles in the atmosphere, thus it is practically impossible to follow all of them in a numerical model. Therefore, we reduce unnecessary degrees of freedom by approximating the population of real particles $\{\{\vec{x}_i(t), \vec{a}_i(t)\}, i = 1, 2, \ldots, N_{\text{r}}^{\text{wp}}\}$ using a population of super-particles: $\{\{\xi_i(t), \vec{x}_i(t), \vec{a}_i(t)\}, i = 1, 2, \ldots, N_{\text{s}}^{\text{wp}}\}$
>
> new> There are many particles in the atmosphere, thus it is practically impossible to follow all of them in a numerical model. However, it is reasonable to assume that only the collective properties of particle population are relevant to predict the behavior of clouds, because clouds are insensitive to each individual particle. Therefore, let us approximate the population of real particles $\{\{\vec{x}_i(t), \vec{a}_i(t)\}, i = 1, 2, \ldots, N_{\text{r}}^{\text{wp}}\}$ by a population of super-particles: $\{\{\xi_i(t), \vec{x}_i(t), \vec{a}_i(t)\}, i = 1, 2, \ldots, N_{\text{s}}^{\text{wp}}\}$

Now, let us provide an answer to your question. First of all, SDM has the "exact limit"; if all the multiplicities are equal to 1, the super-particle population becomes equivalent to the real particle population, hence there is no approximation. As the number of super-particles approaches to the number of real particles, i.e., $N_{\text{s}}^{\text{wp}} \to N_{\text{r}}^{\text{wp}}$, the result of SDM converges to the true solution of the real particle system.

Theoretically, we can justify our choice of $N_{\text{s}}^{\text{wp}}$, if we compare the results with the particle-by-particle (i.e., $N_{\text{s}}^{\text{wp}} = N_{\text{r}}^{\text{wp}}$) simulation results, as you pointed out already. If the deviation is small enough, we can safely conclude that our choice of $N_{\text{s}}^{\text{wp}}$ is sufficient, and that the remaining

degrees of freedom are unnecessary.

However, it is practically impossible to conduct particle-by-particle simulation for clouds, due to the limitation of computer power. Instead, we increase $N_{\mathrm{s}}^{\mathrm{wp}}$ until the results become not sensitive to the change of $N_{\mathrm{s}}^{\mathrm{wp}}$. From which we infer that our $N_{\mathrm{s}}^{\mathrm{wp}}$ is large enough, and that the results we obtained are a good approximate solution of the particle-by-particle solution, but of course this is not a rigorous proof.

Recently, several groups are applying the SDM and other Lagrangian particle-based schemes to simulate cloud chamber experiments. In this case, we can compare the laboratory measurements, particle-by-particle DNS, and SDM. Let us see what will be concluded from these studies.

1-30)   P. 25, ll. 1 – 6, Eq. (82): This is a helpful equation and an interesting switch of perspective on superparticles, multiplicity, and probability. Intuitively, I might agree that both lines of the equation are equal. However, I am wondering if it is possible to derive the second line from the first in a mathematically exact way.

Let us regard the super-particles $\{\{\xi_i(t), \vec{x}_i(t), \vec{a}_i(t)\}, i = 1, 2, \ldots, N_{\mathrm{s}}^{\mathrm{wp}}\}$ as independent and identically distributed (i.i.d.) samples obeying the probability density $p(\xi, \vec{a}, \vec{x}, t)$. Then,

$$
\begin{aligned}
n(\vec{a}, \vec{x}, t) &= \left\langle \sum_{i \in I_{\mathrm{s}}(t)} \xi_i(t) \delta^d(\vec{a} - \vec{a}_i(t)) \delta^3(\vec{x} - \vec{x}_i(t)) \right\rangle \\
&= \sum_{i \in I_{\mathrm{s}}(t)} \left[ \sum_{\xi_i=1}^{\infty} \int d^d a_i \int d^3 x_i \, p(\xi_i, \vec{a}_i. \vec{x}_i, t) \xi_i(t) \delta^d(\vec{a} - \vec{a}_i(t)) \delta^3(\vec{x} - \vec{x}_i(t)) \right] \\
&= N_{\mathrm{s}}(t) \sum_{\xi=1}^{\infty} \xi p(\xi, \vec{a}, \vec{x}, t).
\end{aligned}
$$

1-31)   P. 28, ll. 6 – 7: Equation (7) is stiff even without the curvature and solute terms.

A system that behaves like $dx/dt = -kx$, $k \gg 1$, is called stiff. Equation (7) without the curvature and solute terms has the form $dr^2/dt = \mathrm{const.}$, hence, it is not stiff.

1-32)   P. 28, ll. 15 – 17: It might be necessary to state that the applied collision-coalescence algorithm is linearized, which circumvents the quadratic nature of the collision-coalescence process.

We have added the following sentence to the paragraph.

> (P. 26, ll. 26-29)
> The computational cost of this algorithm is proportional to the number of super-particles $O(N_{\mathrm{s}})$, which is achieved by an efficient collision candidate pair number reduction technique. An additional advantage of this technique is the parallelizability of computation; each super-particle belongs to only one candidate pair, and hence, dependencies are eliminated.

1-33)   P. 28, l. 31: This has already been said on p. 26, ll. 17 – 18.

We revised the first explanation (p. 26, ll. 17 – 18 in the original manuscript) as follows:

> (P. 24, l. 32)
> old< Let $\Delta t_{\mathrm{dyn}}$ be the time step for moist air fluid dynamics, which must fulfill the Courant-Friedrichs-Lewy (CFL) condition of acoustic waves.

new> Let $\Delta t_{\mathrm{dyn}}$ be the time step for moist air fluid dynamics.

1-34)   P. 31, ll. 15 – 16: Does this boundary condition result in a significant loss of (almost) weightless aerosols? Or does is only affect precipitating particles?

It almost only affects precipitating particles. As explained in Sec. 5.5.1, we employed a predictor-corrector scheme with the "simple linear interpolation" of wind velocities from the face grid following Grabowski et al. (2018), which preserves the divergence of fluid flow.

1-35)   P. 31, ll. 17 – 23: This heating is not applied to the surface but the air above. Does this heating start at the beginning of the simulation? The timing of the heating might have an important impact on the degree of turbulence in the simulation.

The title of the section is revised to "Near-surface heating".

It is clarified that the heating is applied from the beginning:

> (P. 31, l. 13)
> Convective cloud development is triggered by a $20\,\mathrm{min}$ heating started from the beginning within a $10\,\mathrm{km}$ wide region ...

We also admit that our setup is not appropriate to create a realistic cloud turbulence observed in nature. The following sentence pointing out this issue is added to the end of Sec. 7.2 "Spatial structure of the cloud, water path, and precipitation amount":

> (P. 36, ll. 3-5)
> Furthermore, the convection was initiated from a stratified, non-turbulent atmosphere; however, this is unrealistic. Following Lasher-Trapp et al. (2005), imposing a spin-up period to develop turbulence in the boundary layer before initiating the deep convection would be desirable.

1-36)   P. 32, ll. 1 – 3: The vertical grid spacing is quite large. It is well known that a too large vertical grid spacing reduces the maximum supersaturation at cloud base (e.g., Morrison and Grabowski 2008). This, in turn, affects the activation of cloud droplets, leading to a smaller number of cloud droplets. Do you see a substantial change in the number of cloud droplets in the sensitivity studies conducted in Section 8.2?

Figure R1-1 compares the time evolution of the total number of activated droplets per unit $y$ length. Here, droplets larger than $1\,\mu\mathrm{m}$ in radius are defined as activated droplets. One member is selected from each ensemble. From the fact that the curves in Fig. R1-1 (a) between the period 1200--2000 s are located close to each other, we conclude that the number of activated CCNs were only weakly affected by the grid size at least for the range we tested. In our model, the deactivation/activation of CCNs are predicted explicitly. The impact of grid size would have been suppressed to some extent buffered by deactivation/activation process, as discussed in Hoffmann (2016) and Grabowski et al. (2018).

[Figure]

[Figure]

[Figure]

Figure R1-1. Time evolution of the number of activated droplets ($r > 1\,\mu\mathrm{m}$) per unit $y$ length. (a) Comparison among DX ensembles. (b) NSP ensembles. (c) DT ensembles. One member is selected from each ensemble.

1-37)   P. 32, Eqs. (91): I am struggling to derive (91) from (84) and (86). Please add some more comments. The same applies to Eqs. (92) and (93).

We revised the explanation as follows.

(P. 32, ll. 9-16)
From Eqs. (85) and (96), the super-particle's multiplicity is then given by

$$
\xi_i = \frac{n^{\mathrm{sulf}}(\log r^{\mathrm{sulf}}_{\mathrm{dry},i}, T^{\mathrm{fz}}_i)}{N_{\mathrm{s}}(0)/2} \frac{V_{\mathrm{domain}} \log(r^{\mathrm{sulf}}_{\mathrm{dry,max}}/r^{\mathrm{sulf}}_{\mathrm{dry,min}})}{\delta(T^{\mathrm{fz}} - (-38\,^{\circ}\mathrm{C}))}
$$
$$
= \frac{dN^{\mathrm{sulf}}}{d\log r^{\mathrm{sulf}}_{\mathrm{dry}}} \left(\log r^{\mathrm{sulf}}_{\mathrm{dry},i}\right) \frac{\log(r^{\mathrm{sulf}}_{\mathrm{dry,max}}/r^{\mathrm{sulf}}_{\mathrm{dry,min}})}{c^{\mathrm{SP}}/2},
$$

where $V_{\mathrm{domain}}$ is the total volume of the domain, $n$ and $p$ in Eq. (85) in this case are given by

$$
n = n^{\mathrm{sulf}}(\log r^{\mathrm{sulf}}_{\mathrm{dry},i}, T^{\mathrm{fz}}_i),
$$
$$
p = \frac{\delta(T^{\mathrm{fz}} - (-38\,^{\circ}\mathrm{C}))}{V_{\mathrm{domain}} \log(r^{\mathrm{sulf}}_{\mathrm{dry,max}}/r^{\mathrm{sulf}}_{\mathrm{dry,min}})},
$$

and $N_{\mathrm{s}}(0)$ in Eq. (85) is replaced by $N_{\mathrm{s}}(0)/2$ because we use half of the super-particles for pure ammonium bisulfate aerosol particles.

Eqs. (104) and (105) (Eqs. (92) and (93) in the original manuscript) can be derived similarly.

1-38)   Pp. 33 – 34: You state that "a 10-member ensemble of simulations" is calculated "by changing the pseudo-random number sequence" in on p. 33, l. 9. This statement is repeated four times on p. 33, l. 18, p. 33, l. 29, p. 34, l. 11, and p. 34, l. 13.

All of them except the first one have been deleted.

1-39)  P. 34, ll. 1 – 11: The tested microphysical timesteps are quite small compared to the literature that states timesteps that are typically between 1 and 10 s (especially in box calculations). Therefore, it might be worthwhile to add one or two additional simulations with even longer timesteps to explore the entire parameter space (e.g., DTx4 and DTx8).

Following your suggestion, we have conducted DTx5 and DTx10. (DTx4 and DTx8 were avoided to unify the data output timing.) We found that there is no significant difference between DTx5 and CTRL. However, DTx10 diverged at around $t$=1200s due to a numerical instability. Therefore, we conclude that time steps five to ten times as large as CTRL could suffice. We have revised the manuscript accordingly.

1-40)  Fig. 2: The caption is missing. I assume that this is a formatting error.

We have fixed the problem by adjusting the size of the figure.

1-41)  P. 43, ll. 11 – 13, Sec. 9.1: Could some of the unrealistic hailstones and graupel particles be explained by the lack of an appropriate shedding formulation that would break these particles after some timesteps?

Our correction (107) (Eq. (95) in the original manuscript) to the rime density formula of Heymsfield and Pflaum (1985) accounts for the wet growth of hailstones and graupel. Wet growth of hailstones may involve shedding, but shedding itself plays a secondary role in the long hailstone problem discussed in Sec. 9.1.

1-42)  P. 44, ll. 8 – 9: I disagree. Figure 9 shows that a higher number of superparticles increases the fluctuation/standard deviation in precipitation.

We admit that there exists a weak trend that the standard deviation of precipitation increases with increasing super-particle number. We have added the following remark to point this out.

> (P. 44, ll. 29-32)
> old< However, Figs. 9 and 10 show that the fluctuation is not sensitive to the initial super-particle number concentration $c^{\mathrm{SP}}$. This indicates that fluctuations in all simulations are dominated by atmospheric turbulence.
>
> new> However, Figs. 9 and 10 show that the fluctuation is not sensitive to the initial super-particle number concentration $c^{\mathrm{SP}}$. This indicates that fluctuations in all simulations are mostly dominated by atmospheric turbulence. One might note that the fluctuations are slightly increasing as $c^{\mathrm{SP}}$ increases. This suggests that the super-particle number affects the turbulence characteristics; however, we leave that for further investigation in future work.

1-43)  P. 63, ll. 27 – 28: The typical citation for the phase relaxation timescale is Squires (1952).

Thank you for the information. We have replaced the reference.

1-44)  P. 64, l. 8 – 10: The general estimate of computational costs and its comparison with a Eulerian bin model is legit. However, I feel uncomfortable with the estimate of the collision calculation. The collision calculation is a quadratic problem in both Eulerian bin and Lagrangian schemes. The linearization of this problem applied in the Lagrangian cloud model might also have its caveats, and I believe that a similar implementation into a Eulerian bin model is possible. Accordingly, the last step from $10^5 - 100^5$ to $10^{10} - 100^{10}$ feels unjustified.

Firstly, let us justify the "linear sampling" of collision candidate pairs used in SDM. Of course, a quadratic algorithm is more straightforward, and already available for Lagrangian particle-based

schemes (e.g., Unterstrasser 2017). However, in order to make it computationally more efficient, Shima et al. (2009) developed a collision candidate pair number reduction technique, which enabled a computational cost proportional to the super-particle number. Dziekan and Pawlowska (2017) confirmed that "linear sampling" of SDM can provide a correct solution.

There is a reason why the linear sampling technique works well. Consider a grid cell with $N_s$ number of super-particles. Let $N_r$ be the number of real particles in this cell. Typical multiplicity can be evaluated as $\xi = N_r/N_s$. Let $P_r$ be the typical collision probability of a real particle pair. Then, the typical collision probability of a super-particle pair before introducing the linear sampling can be evaluated as $P_s = \xi P_r$. Then, the expected number of super-particle collisions $N_s^{\mathrm{collis}}$ can be evaluated as

$$
\begin{aligned}
N_s^{\mathrm{collis}} &\approx \frac{N_s(N_s - 1)}{2} P_s \\
&= \frac{N_s - 1}{2} N_r P_r,
\end{aligned}
$$

which is proportional to the super-particle number $N_s$. Therefore, it is reasonable to examine $O(N_s)$ number of candidate pairs instead of all the combinations. Note that the equation at the end of p. 1313 in Shima et al. (2009), which indicates $\Delta t_{\mathrm{collis}}$ can be chosen independently to $N_s$ and cell volume $\Delta V$, is another consequence of the property $N_s^{\mathrm{collis}} = O(N_s)$. Note also that this fact constitutes the basis of the Direct Simulation Monte Carlo algorithm for rarefied gas flow (Bird, 1994).

Secondly, on the computational cost of Eulerian bin models. Though quadratic algorithms are being used in almost all bin models, we do admit that such a pair number reduction technique similar to SDM can be applied also to bin models. Indeed, Sato et al. (2009) developed a procedure for bin models, and confirmed that it works efficiently.

All in all, to make a fair comparison, the discussion has been revised as follows.

> (P. 50, ll. 29-34)
> For the binary collision calculation, most bin models assess all the combinations of bins. In this case, the computational cost scales with the square of the number of bins, i.e., $10^{10}$--$100^{10}$. However, we can reduce the cost of bin models by introducing a collision pair number reduction technique similar to that of SDM (Sato et al., 2009). Therefore, if we enhance the efficiency by using this algorithm, the computational cost of bin models scales linearly with the number of bins, i.e., $10^5$--$100^5$. However, this is still much larger than 100, i.e., the computational cost of SDM.

**Technical Comments**

1-45)  P. 12, l. 23: I assume that "initiated" is better than "updated" here.
1-46)  P. 13, l. 14: Use a lower case "w" to start this line.
1-47)  P. 24, l. 4: Remove one "planets".
1-48)  P. 27, ll. 3 – 4: I believe it is a "predictor-corrector scheme" and not a "predictor-collector scheme".
1-49)  P. 35, l. 16: Replace "amount" with "number".
1-50)  P. 38, l. 3, l. 6: Replace "segments" with "slopes".

All the above comments are reflected in the revised manuscript.

**References**

Beheng, K.D. (2010). The Evolution of Raindrop Spectra: A Review of Microphysical Essentials. In Rainfall: State of the Science (eds F.Y. Testik and M. Gebremichael). doi:10.1029/2010GM000957

Bird GA. 1994. Molecular gas dynamics and the direct simulation of gas flows. Clarendon Press: Oxford.

Dziekan, P. and Pawlowska, H.: Stochastic coalescence in Lagrangian cloud microphysics, Atmospheric Chemistry and Physics, 17, 13509–13520, https://doi.org/10.5194/acp-17-13509-2017, 2017.

Grabowski, W. W., Dziekan, P., and Pawlowska, H.: Lagrangian condensation microphysics with Twomey CCN activation, Geosci. Model Dev., 11, 103–120, https://doi.org/10.5194/gmd-11-103-2018, 2018.

Hoffmann, F., 2016: The Effect of Spurious Cloud Edge Supersaturations in Lagrangian Cloud Models: An Analytical and Numerical Study. Mon. Wea. Rev., 144, 107–118, https://doi.org/10.1175/MWR-D-15-0234.1

Lasher-Trapp, S.G., Cooper, W.A. and Blyth, A.M. (2005), Broadening of droplet size distributions from entrainment and mixing in a cumulus cloud. Q.J.R. Meteorol. Soc., 131: 195-220. https://doi.org/10.1256/qj.03.199.

Morrison, H., van Lier-Walqui, M., Fridlind, A. M., Grabowski, W.W., Harrington, J. Y., Hoose, C., Korolev, A., Kumjian, M. R., Milbrandt, J. A., Pawlowsk, H., Posselt, D. J., Prat, O. P., Reimel, K. J., Shima, S.-I., van Diedenhoven, B., and Xue, L.: Confronting the challenge of modeling cloud and precipitation microphysics, under review at Journal of Advances in Modeling Earth Systems, 2020.

Niedermeier, D., Augustin-Bauditz, S., Hartmann, S., Wex, H., Ignatius, K., and Stratmann, F. (2015), Can we define an asymptotic value for the ice active surface site density for heterogeneous ice nucleation?. J. Geophys. Res. Atmos., 120, 5036– 5046. doi: 10.1002/2014JD022814.

Sato, Y., Nakajima, T., Suzuki, K., and Iguchi, T. ( 2009), Application of a Monte Carlo integration method to collision and coagulation growth processes of hydrometeors in a bin-type model, J. Geophys. Res., 114, D09215, doi:10.1029/2008JD011247.

Unterstrasser, S., Hoffmann, F., and Lerch, M.: Collection/aggregation algorithms in Lagrangian cloud microphysical models: Rigorous evaluation in box model simulations, Geoscientific Model Development, 10, 1521–1548, https://doi.org/10.5194/gmd-10-1521-2017, 2017.

**Reply to the second reviewer**

Thank you for carefully reading the manuscript. We appreciate your informative and insightful comments. Below, we provided an itemized response to all the comments raised, with the original comments presented in blue. Please also see the revised manuscript and the difference file diff.html, which we will submit separately. All the changes made to the manuscript are detailed in diff.html.

**Major Comments**

2-1)    I was quite confused when reading section 2 from 2.1 to 2.7, because I did not understand how you can ensure mass conservation with this set of prognostic attributes. It only became clear when I read section 2.8 and understood that there are no partially melted wet ice particles (yet) in the model. I would strongly recommend to move that statement from section 2.8 to section 2.1 that particles are either liquid (and fully described by radius r) or ice (and described by major and minor axes *a* and *c* and the density $\varrho_i$).

Following your suggestion, we moved the paragraph from Sec. 2.8 to Sec. 2.1 with some modifications.

> (P. 5, ll. 24-29)
> In this study, for simplicity, partially frozen/melted particles are not considered. We assume that each particle completely freezes or melts instantaneously (see Secs. 4.1.4 and 4.1.5).
> Therefore, either the equivalent droplet radius $r$ or ice particle attributes $\{a, c, \rho^i\}$ are always zero in our model. Furthermore, we assume that all particles contain soluble substances and are always deliquescent even when the humidity is low (see Sec. 4.1.6). Further, as a crude representation of ``pre-activation'', we do not allow the complete sublimation of an ice particle (see Sec. 4.1.7). Therefore, $r$ and $\{a, c, \rho^i\}$ cannot be simultaneously zero.

2-2)    From Figure 8 and 19 I would conclude that snow (aggregates) is falling too fast in SCALE-SDM, i.e. the green data point to not coincide with the empirical relations for aggregates. Can you explain this bias in the fallspeed of snow? I think this should be discussed in the paper.

The bias can be explained by the air density dependence of fall speed. In Figs. 8 and 19, the green slopes for snow aggregates represent the formulas of Locatelli and Hobbs (1974) (LH74 in short) and Heymsfield et al. (2002) (H02 in short). LH74's formulas are for data measured between altitudes of 750 and 1500 m above sea level, hence the density is approximately 1.1 kg m$^{-3}$. H02's formula is for temperature and pressure of -10 °C and 500 hPa, hence the density is approximately 0.66 kg m$^{-3}$. In our simulation, most of the snow aggregates exist in the anvil cloud, where the density is approximately 0.38 kg m$^{-3}$. Khvorostyanov and Curry (2002) estimated that the terminal velocities of large ice particles scale with the ambient density to the power of -1/2. Figure R2-1 below was created by incorporating this density dependence to Fig. 19. That is, we multiplied the LH74's formulas for aggregates by a factor of $(0.38\,\mathrm{kgm}^{-3}/1.1\,\mathrm{kgm}^{-3})^{-1/2} \approx 1.70$, and the formula of H02 for aggregates by a factor of $(0.38\,\mathrm{kgm}^{-3}/0.66\,\mathrm{kgm}^{-3})^{-1/2} \approx 1.32$. Now the agreement between our model results and the formulas is much better.

To clarify this point, the above discussion is added to Sec. 7.3 "Ice particle morphology and fall speeds"

[Figure]

Figure R2-1. Same as Fig. 19 but with snow aggregates formulas (green slopes) adjusted to an air density of 0.38 kg m$^{-3}$.

 Maybe related to that: Wouldn't it be more accurate to use an ellipse instead of the circumcircle for the area in Boehms formula (section 4.1.3, page 11, line 20)? Do you take into account the turbulence correction for large Reynolds numbers in Boehms equations? The latter is actually necessary to limit the fall speed of large aggregates and match the observed terminal fall speed of aggregates.

As explained in our reply to Comment 2-2, the fall speeds of snow aggregates in our model compare well with other formulas if the air density difference is considered. Regarding the turbulence correction, yes, it is incorporated in our model (see Eq. R2-1 below). However, we learned that circumscribed ellipse instead of circumcircle has to be used in Böhm's formula. We also learned that the characteristic length in Böhm's formula is not given by the maximum dimension. Nevertheless, based on the assessment presented below, we confirmed that these corrections do not change the behavior of the cloud significantly, and hence, this flaw causes only a minor impact on this study.

Noting that area ratio $q_i \leq 1$ always holds in our model, Böhm (1989,1992,1999)'s formula $v_{\text{Böhm}}^\infty(m_i, \phi_i, d_i, q_i; \rho_i, T_i)$ can be summarized as follows:

$$X = \frac{8 m_i g \rho}{\pi \mu^2 \max(\phi_i, 1) q_i^{1/4}},$$

$$X' = X \frac{1 + (X/X_0)^2}{1 + 1.6(X/X_0)^2},$$   R2-1

$$X_0 = 2.8 \times 10^6, \quad \text{for ice particles,}$$

$$k = \min\left\{ \max\left(0.82 + 0.18\phi_i, 0.85\right), \left(0.37 + \frac{0.63}{\sqrt{\phi_i}}\right), \right.$$

$$\left. \frac{1.33}{\max(\log\phi_i, 0) + 1.19} \right\},$$

$$\Gamma = \max\left\{1, \min\left(1.98, 3.76 - 8.41\phi_i + 9.18\phi_i^2 - 3.53\phi_i^3\right)\right\},$$

$$C_{\mathrm{DP}} = \max\left(0.292k\Gamma, 0.492 - 0.200/\sqrt{\phi_i}\right),$$

$$C_{\mathrm{DO}} = 4.5k^2 \max\left(\phi_i, 1\right),$$

$$\beta = \left[1 + \frac{C_{\mathrm{DP}}}{6k}\left(\frac{X'}{C_{\mathrm{DP}}}\right)^{1/2}\right]^{1/2} - 1,$$

$$\gamma = \frac{C_{\mathrm{DO}} - C_{\mathrm{DP}}}{4C_{\mathrm{DP}}},$$

$$N_{\mathrm{Re}} = \frac{6k}{C_{\mathrm{DP}}}\beta^2\left[1 + \frac{2\beta e^{-\beta\gamma}}{(2+\beta)(1+\beta)}\right],$$

$$v_{\mathrm{B\ddot{o}hm}}^{\infty} = \frac{\mu N_{\mathrm{Re}}}{\rho d_i}.$$

In SCALE-SDM 0.2.5-2.2.0/2.2.1, we assumed that the characteristic length $d_i$ is given by the maximum dimension $D_i = 2\max\left(a_i, c_i\right)$, and area ratio $q_i$ is given by the the area ratio with respect to the circumcircle $q_i^{\mathrm{cc}} = A_i/A_i^{\mathrm{cc}}$, but we learned this is not correct. In Böhm's theory, they are defined by

$$d_i = 2a_i, \quad q_i = q_i^{\mathrm{ce}} = A_i/A_i^{\mathrm{ce}}, \quad\quad\quad\quad \text{R2-2}$$

i.e., for columnar particles, minor axis is used for the characteristic length $d_i$, and the area ratio with respect to the circumscribed ellipse is used for $q_i$. Figure 1 in Böhm (1989) suggests $q_i = q_i^{\mathrm{ce}}$. It is not clearly specified, but from the second equality of Eq. 17 in Böhm (1992), we can confirm that $d_i = 2a_i$.

For planar ice particles ($\phi_i < 1$), $v_{\mathrm{B\ddot{o}hm}}^{\infty}(d_i = 2a_i, q_i = q_i^{\mathrm{ce}})$ and $v_{\mathrm{B\ddot{o}hm}}^{\infty}(d_i = D_i, q_i = q_i^{\mathrm{cc}})$ yield the same results, because $2a_i = D_i$ and $q_i^{\mathrm{ce}} = q_i^{\mathrm{cc}}$ hold for $\phi_i < 1$. However, for columnar ice particles ($\phi_i > 1$), $v_{\mathrm{B\ddot{o}hm}}^{\infty}(D_i, q_i^{\mathrm{cc}})$ always underestimates the fall velocity. From the above equations, we can derive $v_{\mathrm{B\ddot{o}hm}}^{\infty}(2a_i, q_i^{\mathrm{ce}})/v_{\mathrm{B\ddot{o}hm}}^{\infty}(D_i, q_i^{\mathrm{cc}}) = \phi_i^{3/4}$ for $X \ll 1$, and $v_{\mathrm{B\ddot{o}hm}}^{\infty}(2a_i, q_i^{\mathrm{ce}})/v_{\mathrm{B\ddot{o}hm}}^{\infty}(D_i, q_i^{\mathrm{cc}}) = \phi_i^{7/8}$ for $X \gg 1$. Therefore, if $\phi_i = 2$, the ratio $v_{\mathrm{B\ddot{o}hm}}^{\infty}(2a_i, q_i^{\mathrm{ce}})/v_{\mathrm{B\ddot{o}hm}}^{\infty}(D_i, q_i^{\mathrm{cc}})$ is in the rage of 1.68–1.83; if $\phi_i = 10$, the range is 5.62–7.50; if $\phi_i = 20$, it is 9.46–13.75. From Fig. R2-2 we can confirm that Böhm's original definition $v_{\mathrm{B\ddot{o}hm}}^{\infty}(2a_i, q_i^{\mathrm{ce}})$ agrees well with the formulas of Westbrook (2008), and Heymsfield and Westbrook (2010).

[Figure]

Figure R2-2. Comparison of terminal velocity formulas for long ice particles with aspect ratio $\phi = 10$. Westbrook (2008)'s formula is applicable only to small ice particles. Böhm (1992)'s formula with the correct $d$ and $q$ agrees well with other formulas.

Therefore, the correction R2-2 generally increases the fall speed of columnar ice particles, and the increase factor is larger for longer particles. Then, through the ventilation effects (13) and (17), the diffusional growth of columnar ice particles is enhanced. Due to this mechanism, we observed a creation of very long ice particles with aspect ratio $\phi > 100$ if we incorporate the correction R2-2 to SCALE-SDM 0.2.5-2.2.1. However, this is unrealistic. The maximum aspect ratio reported is approximately 30 in Auer and Veal (1970) (see Fig. 12 therein), and 15.77 in Um et al. (2015). In nature, such an extreme shape ice particle would be shattered spontaneously or by collision, but for the moment, we fix this issue in an ad-hoc way; we do not allow an ice particle to grow by diffusion slenderer than $\phi = 40$ by imposing a limiter to the effective inherent growth ratio $\Gamma^*$ as follows.

$$\Gamma^* = 1 \quad \text{for } dm_i \geq 0 \ \wedge \ \phi_i > 40. \qquad\qquad \text{R2-3}$$

We incorporated the corrections R2-2 and R2-3 into SCALE-SDM 0.2.5-2.2.1 to create a revision, SCALE-SDM 0.2.5-2.2.2. To assess the impact of these corrections, we conducted the same simulation as the typical realization of CTRL using the new model. We observed that the precipitation was developed a few minutes faster, but the total precipitation amount was almost the same as the previous versions (Fig. R2-3). Figure R2-4 compares the time evolution of water paths. Here, a noticeable decrease of graupel water path can be observed, which is attributed to the faster fall speed of columnar graupel particles (i.e., densely rimed columns). This in turn increased the rain water path. The time evolution of other hydrometeor water paths (cloud, cloud ice, and snow) were almost unchanged. Ice particle morphology distributions resemble closely to the previous results except the vanishment of cloud ice particles with relatively slow terminal velocities (Figs. R2-5 -- R2-8. See also Movies 13--16 in the Supplement). The corrections do not alter the spatial structure of the cloud either (Movie 12 in the Supplement).

[Figure]

[Figure]

Figure R2-3. Changes in accumulated precipitation amounts before and after corrections. The long dashed, solid, and short dashed lines represent the SCALE-SDM 0.2.5-2.2.0, -2.2.1, and -2.2.2, respectively.

Figure R2-4. Changes in the domain-averaged water path before and after corrections. The long dashed, solid, and short dashed lines represent the SCALE-SDM 0.2.5-2.2.0, -2.2.1, and -2.2.2, respectively.

Based on the above discussion, we have made various revisions to the manuscript. Major changes are summarized as follows.

Title of the manuscript is slightly modified:
old< Predicting the morphology of ice particles in deep convection using the super-droplet method: development and evaluation of SCALE-SDM 0.2.5-2.2.0/2.2.1
* * *
new> Predicting the morphology of ice particles in deep convection using the super-droplet method: development and evaluation of SCALE-SDM 0.2.5-2.2.0, -2.2.1, and -2.2.2

In Sec. 4.1.3 "Ice particle terminal velocity", the second paragraph is added to inform the readers that $d_i = 2a_i$ and $q_i = q_i^{\text{ce}}$ are the correct definition.

In Figs. 20 and 21, the results of SCALE-SDM 0.2.5-2.2.2 are now included.

Section 9.2 "Fix of ice particle terminal velocity implementation" is added. Here, the impact of the corrections $d_i = 2a_i$ and $q_i = q_i^{\text{ce}}$ on this study is assessed in detail.

SCALE-SDM 0.2.5-2.2.2 is released on the software repository.

List of symbols is updated.

[Figure]

Figure R2-5. Same as Fig. 5 but shows results from SCALE-SDM 0.2.5-2.2.2. See also Movie 13 in the Supplement.

Figure R2-6. Same as Fig. 6 but shows results from SCALE-SDM 0.2.5-2.2.2. See also Movie 14 in the Supplement.

[Figure]

Figure R2-7. Same as Fig. 7 but shows results from SCALE-SDM 0.2.5-2.2.2. See also Movie 15 in the Supplement.

Figure R2-8. Same as Fig. 8 but shows results from SCALE-SDM 0.2.5-2.2.2. See also Movie 16 in the Supplement.

**Minor Comments**

2-4) page 5, line 5-7: I agree that a rigorous theory for bulk models is still lacking, but it would nevertheless be appropriate to reference the review by Beheng (2010). This paper gives an overview of the steps that have been made towards such a theoretical foundation, at least for liquid clouds and rain.

To clarify our argument, we have rephrased the part as follows.

> (P. 2, ll. 14-21)
> old< … They solve a mathematical model that is closed in lower moments of the distribution function of cloud droplets, rain droplets, and ice particle categories (e.g., total mass and total number of particles). Currently, bulk models do not have a rigorous theoretical foundation and must rely on empirical parameterizations. A more bottom-up approach to construct more accurate and reliable numerical models would thus be desired.
>
> new> … They solve a mathematical model that is closed in the lower moments of the distribution function of cloud droplets, rain droplets, and ice particle categories (e.g., mass and number mixing ratios). The basic premise of bulk models is that the distribution function can be determined by the lower moments, but such a universal relationship is unknown. In other words, in bulk models, to predict the time evolution of a chosen set of moments, their time derivatives are approximated by some functions of the moments being predicted, but this is not generally possible (see, e.g., Beheng, 2010). It would be also informative to note the analogy and difference between the Navier--Stokes equation and bulk models (Morrison et al., 2020), which highlights the difficulty in deriving bulk models. Therefore, for cloud microphysics, a more bottom-up approach to construct more accurate and reliable numerical models would be desired.

2-5)  page 6, line 4: 'approximated by a histogram', here I would recommend to replace 'histogram' by 'finite volumes or finite differences'.

We agree that 'histogram' would be awkward as an explanation of a numerical scheme, but it has an affinity to 'bin'. Therefore, we rephrased the sentence as follows.

> (P. 3, ll. 19-21)
> old< Bin schemes adopt an Eulerian approach and the particle distribution function is approximated by a histogram.
>
> new> Bin schemes adopt an Eulerian approach and the particle distribution function is approximated using a finite number of control volumes (histogram).

2-6)  page 5, line 8: 'breakdown of the Smoluchowski equation'. Not all readers might be familiar with the notion of the breakdown of the Smoluchowski equation. A reference other than Smoluchowski (1916) or an additional sentence would be helpful.

We have added Alfonso and Raga (2017), and Dziekan and Pawlowska (2017).

2-7)  page 9, section 2.7: It should be mentioned that the assumption that particles move at their terminal fall velocity is an approximation. In the framework of a Lagrangian particle model this can quite easily be improved by considering the adjustment towards the new terminal fall velocity, e.g., after a collision event (see e.g. Naumann and Seifert 2015).

We clarified that it is a simplification. Sec. 2.7 "Velocity" is modified as follows.

> (P. 7, l. 18)
> old< We consider that each particle is always moving at its terminal velocity.
>
> new> We approximate that each particle is always moving at its terminal velocity.

New paragraph is added to Sec. 4.1.1 "Advection and sedimentation"

> (P. 8, ll. 25-27)
> In this study, we assume that terminal velocity is always achieved instantaneously; however,

this is a simplification. The relaxation time of large droplets is a few seconds (Fig. 3 of Wang and Pruppacher (1977)). The acceleration of particles can be considered by explicitly solving the motion equation (see, e.g., Naumann and Seifert (2015)).

2-8)   page 13, section 4.1.6: When I first read this paragraph I was surprised that the ventilation is missing and is not even mentioned. It would be good to mention this approximation already here and not only later in section 9.2.4.

We have added the following explanation to the paragraph.

(P. 11, ll. 15-18)
The growth of a droplet by condensation/evaporation is governed by Eqs. (8)-(10) in our model. When a droplet or an ice particle falls through the air, the flow around it enhances the diffusional growth, a phenomenon known as the ventilation effect. It does not essentially affect the growth of droplets smaller than $50\,\mu m$ in radius (see Sec. 13.2.3 of Pruppacher and Klett (1997)). Therefore, for simplicity, we do not consider the ventilation effect on droplets in this study. ...

2-9)   page 14, eq. (13): Why is the minimum mass $m^{\mathrm{i}}_{\mathrm{min}}$ necessary in this equation? Is this because homogeneously frozen droplets may not contain any insoluble aerosol mass and then you would eventually have a super-droplet with zero mass? Does that $m^{\mathrm{i}}_{\mathrm{min}}$-particle not grow immediately when it is advected into cold, ice-supersaturated conditions and produce unrealistic ice? It does remember its freezing temperature, but it is already ice and would therefore grow immediately when the environment is supersaturated with respect to ice. I don't understand how this is implemented.

This is a crude expression of pre-activation. Next to Eq.(14) we have added the following explanation.

(P. 12, ll. 18-21)
This is a crude representation of pre-activation (see, e.g., Marcolli, 2017, for a review). Each particle keeps the memory of ice activation until the ambient temperature rises above $0\,^{\circ}\mathrm{C}$; A particle with $m^{\mathrm{i}}_{\mathrm{min}}$ ice grows immediately after the ambient air is supersaturated over ice irrespective of its freezing temperature $T^{\mathrm{fz}}_{i}$.

We have also added the following discussion to Sec. 9.3.1 "Ice nucleation pathways"

(P. 61, ll. 28-34)
A crude model of pre-activation is incorporated in our model by inhibiting complete sublimation (see Eq. (14) and the explanation follows). Pre-activation denotes ``the capability of particles or materials to nucleate ice at lower relative humidities or higher temperatures compared to their intrinsic ice nucleation efficiency after having experienced an ice nucleation event or low temperature before'' (Marcolli, 2017). Intensive sophistication based on laboratory studies is required; however, particle-based models are suitable for exploring the atmospheric relevance of pre-activation. Conversely, one might want to switch off pre-activation in our model, which is possible by resetting the particles as deliquescent aerosol particles when complete sublimation occurs.

2-10)   page 15, eq. (21): Why is it necessary to impose this explicit limit to water saturation? If water droplets are present, then the supersaturation should be limited to due the rapid condensational growth. If no water droplets are present and no CCN can be activated, then the limit to water saturation might be unphysical.

First of all, note that the limit to water saturation only applies to the deposition density formula of Chen and Lamb (1994a) given in Eq. (20). It is not clarified in Chen and Lamb (1994a), but Miller and Young (1979) suggested to use the same deposition density at and above water saturation. Maybe they assumed this simply because no data was available above water saturation, but in

order to avoid the use of an unrealistically low deposition density, we followed their suggestion.

2-11)  page 15 and 16: For depositional growth it is assumed that particle are spherical for *D* smaller than 10 microns (top of page 15), but for sublimation it is assumed that particles become spherical only when smaller than 1 micron. Why this asymmetry/hysteresis?

In SCALE-SDM 0.2.5-2.2.0, $\Gamma(T) = 1$ for $D < 10\,\mu m$ applies to both deposition and sublimation. In SCALE-SDM 0.2.5-2.2.1 and -2.2.2, $\Gamma = 1$ is always assumed for sublimation. $\Gamma = 1$ just preserves the aspect ratio during sublimation/deposition (if ventilation effect is ignored), hence the creation of very small planar or columnar ice particles can happen, which occurs particularly when they sublimate. Therefore, we decided to reset the shape of an ice particle as spherical when it is very small. Radius of minor axis smaller than $1\,\mu m$ is the criteria we introduced. We have to admit this is not based on a rigorous physical consideration, but it would be justified because $1\,\mu m$ is roughly the boundary between the continuum and kinetic regimes. The specific value of the criteria would not be very important; we can expect that almost all submicron sized, sublimating ice particles will sublimate completely almost instantaneously. Still, it is worth mentioning that the spherical resetting we introduced is beneficial for numerical simulation; if $\phi_i = 1$, Eq. (11) without the ventilation effect reduces to a simple form $dm_i^{2/3}/dt = \mathrm{const.}$

2-12)  page 16, line 14: 'rime mass fraction does not change during sublimation'. According to equation (29) rime mass fraction does not change during deposition (dm > 0) and only change during sublimation (dm < 0). Do you mean 'rime mass fraction does only change during sublimation'.

The definition of rime mass fraction is $m_i^{\mathrm{rime}}/m_i$, hence both the text and the equation are correct. To avoid confusion, we have clarified the definition of rime mass fraction.

2-13)  page 17, line 16: 'remove k from the system'. Do you remove the particle because you have not yet introduced the multiplicity in those equations? Isn't it confusing to give here a Monte-Carlo algorithm without multiplicity, which is (as I assume) not used in SCALE-SDM. Maybe it should be emphasized (again) that this is the underlying theoretical model, but not the numerical implementation.

To clarify and emphasize that the section is devoted to the description of the underlying theoretical model, we have added the following paragraph at the end of the subsection.

> (End of Sec. 4.1.9 "Coalescence between two droplets")
> Let us emphasize that the stochastic model introduced in this section describes the underlying mathematical model of coalescence process, not the Monte Carlo algorithm of SDM that solves the stochastic process numerically. In the preceding paragraph, droplet $k$ was removed from the system because both $j$ and $k$ are real particles. On the contrary, in the SDM, the number of super-particles is (almost always) conserved through coalescence (Shima et al., 2009).

2-14)  page 21, line 16: Why $c_j$+min($a_k$, $c_k$)? Shouldn't it be $c_j$+max($a_k$, $c_k$) for the longest possible minor axis?

Even when a pair of ice particles stick together and construct an aggregate with the maximum possible volume, we still assume that these ice particles are falling with their maximum dimension perpendicular to the flow direction. Probably it rarely happens that planar or columnar ice particles rotate vertically and stick together at a right angle, like the shape of "T".

2-15)  page 24, line 4: 'other planets planets'. Two times 'planets'.

We have fixed the typo.

2-16)   page 28, section 5.5.5: Would it be possible to discuss the time step of the Monte Carlo scheme in some more detail? Or is this basically the same argument as in Shima et al.(2009) on page 1313?

The time step can be determined from the same argument, but in the revised manuscript we put it in a slightly different way to provide a precise physical interpretation. See the second and third paragraphs of Sec. 5.5.5 "Coalescence, riming, and aggregation".

2-17)   page 27, line 3: 'predictor-collector', maybe 'predictor-corrector'?

We have fixed the typo.

2-18)   page 44, line 11-12: 'Figure 10 clearly indicates that the super-particle number concentration must be larger than 128/cell'. This is not obvious to me. From Figure 10 I would conclude that 64/cell or even 32/cell is actually fine. Can you explain how you determined the value of 128/cell.

We jumped to the conclusion, but we admit it is not obvious. To provide a quantitative basis, we have conducted a statistical hypothesis test, the result of which is summarized in Table R2-1.

The equality of variances and averages are tested by F-test and T-test, respectively. "2-512" indicates that the column corresponds to the test between NSP002 and NSP512. The same applies to other column headers. CWP, …, and SWP represent the maximum water path of each hydrometeor type plotted in Fig. 10. "prec" represents the accumulated precipitation amount plotted in Fig. 9. The number in each cell represents the p-value, i.e., the probability that the actual difference is greater than the observed difference under the null hypothesis that the variances or averages of the two ensembles are equal. Yellow and green indicate that there exists a significant difference with a confidence level of 99% and 95%, respectively. Blue indicates that the equality cannot be rejected.

Our F-test could not detect a significant difference in variances in most of the cases. From the T-test, we confirmed that the numerical convergence of CWP is slow, which can be observed also in Fig. 10. This is closely related to the onset of warm rain through coalescence; From Fig. 10, we can find that the maximum of cloud water path coincides with the emergence of rainwater. Therefore, a small shift of the warm rain onset time changes the maximum value, but it does not have a big impact on the overall properties of the simulated cloud. Indeed, with a few exceptions, the maximum water paths of all the other hydrometeor types do not show a significant difference if super-particle number concentration is larger than 64 or 128/cell.

All in all, we may conclude that numerical convergence with respect to super-particle number is fairly well achieved at 128/cell, but 64/cell would be also acceptable.

Based on the above discussion, we have revised the manuscript as follows.

> (P. 45, ll. 1-11)
> Figure 9 indicates that the accumulated precipitation amount is less sensitive to the super-particle number. However, Fig. 10 reveals that the initial super-particle number concentration $c^{\mathrm{SP}}$ affects the maximum water path statistics. The numerical convergence of maximum cloud water path is noticeably slow. This is closely related to the onset of warm rain through coalescence. From Fig. 3, we determine that the maximum of the cloud water path coincides with the emergence of rainwater. Therefore, a small shift of the warm rain onset time changes the maximum cloud water path; however, it does not have a considerable impact on the overall properties of the simulated cloud. The maximum water paths of all the other hydrometeor types do not show a significant difference if $c^{\mathrm{SP}}$ is larger than 64 or 128/cell (see also Table R2-1 of authors' response to anonymous referee #2). When the number of super-particles was too low, more rain droplets were produced because of an erroneous enhancement of collision-coalescence that suppressed the amount of cloud droplets, cloud ice particles, and graupel particles.

To summarize, we may conclude that numerical convergence regarding the super-particle number is fairly well achieved at NSP128 (CTRL), i.e., $c^{\mathrm{SP}} = 128/cell$.

F-test (H0: two variances are equal)

| | 2-512 | 4-512 | 8-512 | 16-512 | 32-512 | 64-512 | 128-512 | 256-512 |
|------|--------|--------|--------|--------|--------|--------|---------|---------|
| CWP | 0.0080 | 0.0690 | 0.3131 | 0.7480 | 0.7210 | 0.7772 | 0.5606 | 0.5758 |
| RWP | 0.1061 | 0.3265 | 0.3044 | 0.5608 | 0.2523 | 0.3157 | 0.8333 | 0.1567 |
| CIWP | 0.3214 | 0.8093 | 0.7933 | 0.8478 | 0.7260 | 0.2142 | 0.0446 | 0.0368 |
| GWP | 0.0457 | 0.1412 | 0.0502 | 0.8361 | 0.5501 | 0.0034 | 0.6177 | 0.0839 |
| SWP | 0.0425 | 0.0157 | 0.2712 | 0.6814 | 0.2791 | 0.4521 | 0.7480 | 0.6340 |
| prec | 0.0229 | 0.3057 | 0.0145 | 0.0507 | 0.0897 | 0.1556 | 0.6828 | 0.2113 |

F-test (H0: two variances are equal)

| | 2-256 | 4-256 | 8-256 | 16-256 | 32-256 | 64-256 | 128-256 | 512-256 |
|------|--------|--------|--------|--------|--------|--------|---------|---------|
| CWP | 0.0019 | 0.0205 | 0.1225 | 0.8108 | 0.3620 | 0.7812 | 0.2580 | 0.5758 |
| RWP | 0.8313 | 0.6512 | 0.6852 | 0.3932 | 0.7764 | 0.6675 | 0.2242 | 0.1567 |
| CIWP | 0.0034 | 0.0613 | 0.0640 | 0.0553 | 0.0166 | 0.3638 | 0.9285 | 0.0368 |
| GWP | 0.7671 | 0.7828 | 0.8007 | 0.1245 | 0.2447 | 0.1702 | 0.2082 | 0.0839 |
| SWP | 0.1112 | 0.0456 | 0.5264 | 0.3779 | 0.1243 | 0.2240 | 0.4272 | 0.6340 |
| prec | 0.2682 | 0.8150 | 0.1955 | 0.4518 | 0.6356 | 0.8581 | 0.3934 | 0.2113 |

T-test (H0: two averages are equal)

| | 2-512 | 4-512 | 8-512 | 16-512 | 32-512 | 64-512 | 128-512 | 256-512 |
|------|--------|--------|--------|--------|--------|--------|---------|---------|
| CWP | 8.E-28 | 3.E-26 | 5.E-24 | 8.E-21 | 6.E-19 | 2.E-13 | 4.E-09 | 0.0258 |
| RWP | 4.E-09 | 3.E-08 | 2.E-06 | 0.0007 | 0.0370 | 0.5484 | 0.7741 | 0.5956 |
| CIWP | 3.E-11 | 3.E-09 | 2.E-08 | 4.E-06 | 4.E-05 | 0.0120 | 0.3656 | 0.1916 |
| GWP | 6.E-12 | 4.E-10 | 6.E-09 | 4.E-05 | 0.0024 | 0.0011 | 0.0623 | 0.0297 |
| SWP | 9.E-09 | 2.E-07 | 3.E-08 | 9.E-08 | 0.0195 | 0.6384 | 0.4855 | 0.4595 |
| prec | 0.1901 | 0.9715 | 0.9218 | 0.1673 | 0.9969 | 0.6853 | 0.2584 | 0.2232 |

T-test (H0: two averages are equal)

| | 2-256 | 4-256 | 8-256 | 16-256 | 32-256 | 64-256 | 128-256 | 512-256 |
|------|--------|--------|--------|--------|--------|--------|---------|---------|
| CWP | 3.E-26 | 1.E-24 | 1.E-22 | 1.E-19 | 2.E-17 | 9.E-12 | 3.E-06 | 0.0258 |
| RWP | 8.E-12 | 2.E-10 | 1.E-08 | 2.E-05 | 0.0017 | 0.1561 | 0.3664 | 0.5956 |
| CIWP | 4.E-06 | 4.E-05 | 0.0002 | 0.0151 | 0.0890 | 0.4621 | 0.7240 | 0.1916 |
| GWP | 7.E-14 | 3.E-11 | 6.E-10 | 0.0004 | 0.0725 | 0.0422 | 0.9301 | 0.0297 |
| SWP | 7.E-09 | 1.E-07 | 2.E-08 | 9.E-08 | 0.0057 | 0.2244 | 0.1683 | 0.4595 |
| prec | 0.9852 | 0.1224 | 0.0858 | 0.0013 | 0.0929 | 0.2658 | 0.9686 | 0.2232 |

Table R2-1. F-test and T-test for statistically testing the equality of variances and averages, respectively. "2-512" indicates that the column corresponds to the test between NSP002 and NSP512. The same applies to other column headers. CWP, …, and SWP represent the maximum water path of each hydrometeor type plotted in Fig. 10. "prec" represents the accumulated precipitation amount plotted in Fig. 9. The number in each cell represents the p-value, i.e., the

probability that the actual difference is greater than the observed difference under the null hypothesis that the variances or averages of the two ensembles are equal. Yellow and green indicate that there exists a significant difference with a confidence level of 99% and 95%, respectively. Blue indicates that the equality cannot be rejected.

2-19)   page 60, line 14: 'approximating the particle is spherical' -> 'as spherical'

We have fixed the typo.

2-20)   page 60 and elsewhere: I find collision-riming and collision-aggregation awkward wording. Riming and aggregation are always due to collisions. Hence, the prefix 'collision' is not necessary.

Good idea. Coalescence also always accompanies collision. We removed "collision-" from the manuscript unless otherwise it is misleading.

2-21)   page 60, line 25: First sentence of 9.2.7 'We assume that collision-riming's collection efficiency'. Should this read aggregation instead of riming?

We have fixed the typo.

2-22)   page 62, line 9: 'Seifert et al. (2005)'s model'. This is actually the Low and List (1982) breakup model combined with Beard and Ochs (1995) for small drops. Seifert et al. (2005) did not add anything new to the physics of the breakup process.

We decided to cite Prat et al. (2012) to introduce breakup models. They tested several combinations of existing models, such as Low and List (1982), Seifert et al. (2005) (compilation of Low and List (1982) and Beard and Ochs (1995)), Testik et al. (2011), and McFarquhar (2004).

2-23)   page 62, line 13-15: I would recommend to delete the two sentences starting with 'On average,...'. This is very questionable, has not been shown in the paper and would, in my opinion, be just a compensation of errors. Such a compensating effect is not a good reason to ignore breakup processes.

We have deleted the two sentences. We admit that the thought experiment assessing the impact is too simplified and misleading.

**References**

Alfonso, L. and Raga, G. B.: The impact of fluctuations and correlations in droplet growth by collision–coalescence revisited – Part 1: Numerical calculation of post-gel droplet size distribution, Atmos. Chem. Phys., 17, 6895–6905, https://doi.org/10.5194/acp-17-6895-2017, 2017.

Auer, A.H. and D.L. Veal, 1970: The Dimension of Ice Crystals in Natural Clouds. J. Atmos. Sci., 27, 919–926, https://doi.org/10.1175/1520-0469(1970)027<0919:TDOICI>2.0.CO;2

Böhm, H.P., 1989: A General Equation for the Terminal Fall Speed of Solid Hydrometeors. J. Atmos. Sci., 46, 2419–2427, https://doi.org/10.1175/1520-0469(1989)046<2419:AGEFTT>2.0.CO;2

Böhm, J. P.: A general hydrodynamic theory for mixed-phase microphysics. Part I: drag and fall speed of hydrometeors, Atmospheric Research, 27, 253–274, https://doi.org/10.1016/0169-8095(92)90035-9, 1992.

Böhm, J. P.: Revision and clarification of 'a general hydrodynamic theory for mixed-phase microphysics', Atmospheric Research, 52, 167–176, https://doi.org/10.1016/S0169-8095(99)00033-2, 1999.

Dziekan, P. and Pawlowska, H.: Stochastic coalescence in Lagrangian cloud microphysics, Atmos. Chem. Phys., 17, 13509–13520, https://doi.org/10.5194/acp-17-13509-2017, 2017.

Heymsfield, A. J., Lewis, S., Bansemer, A., Iaquinta, J., Miloshevich, L. M., Kajikawa, M., Twohy, C., and Poellot, M. R.: A general approach for deriving the properties of cirrus and stratiform ice cloud particles, Journal of the Atmospheric Sciences, 59, 3–29, https://doi.org/10.1175/1520-0469(2002)059<0003:AGAFDT>2.0.CO;2, 2002.

Heymsfield, A.J. and C.D. Westbrook, 2010: Advances in the Estimation of Ice Particle Fall Speeds Using Laboratory and Field Measurements. J. Atmos. Sci., 67, 2469–2482, https://doi.org/10.1175/2010JAS3379.1

Khvorostyanov, V.I. and J.A. Curry, 2002: Terminal Velocities of Droplets and Crystals: Power Laws with Continuous Parameters over the Size Spectrum. J. Atmos. Sci., 59, 1872–1884, https://doi.org/10.1175/1520-0469(2002)059<1872:TVODAC>2.0.CO;2

Locatelli, J. D. and Hobbs, P. V.: Fall speeds and masses of solid precipitation particles, Journal of Geophysical Research, 79, 2185–2197, https://doi.org/10.1029/jc079i015p02185, 1974.

Morrison, H., van Lier-Walqui, M., Fridlind, A. M., Grabowski, W.W., Harrington, J. Y., Hoose, C., Korolev, A., Kumjian, M. R., Milbrandt, J. A., Pawlowsk, H., Posselt, D. J., Prat, O. P., Reimel, K. J., Shima, S.-I., van Diedenhoven, B., and Xue, L.: Confronting the challenge of modeling cloud and precipitation microphysics, under review at Journal of Advances in Modeling Earth Systems, 2020.

Um, J., McFarquhar, G. M., Hong, Y. P., Lee, S.-S., Jung, C. H., Lawson, R. P., and Mo, Q.: Dimensions and aspect ratios of natural ice crystals, Atmos. Chem. Phys., 15, 3933–3956, https://doi.org/10.5194/acp-15-3933-2015, 2015.

**Additional changes not required by the referees**

Other than the revisions made in response to the referee comments, there are various changes in the manuscript made by the authors. Major ones of them are listed below, with the reasons why we revised them.

Following the policy of GMD, we added PDF bookmarks and removed the table of contents. Accordingly, the following sentence is added to the end of Sec. 1.

Note that a comprehensive table of contents is provided as PDF bookmarks.

---

## Referee Report (RR1)

**Review of "Predicting the morphology of ice particles in deep convection using the super-droplet method: development and evaluation of SCALE-SDM 0.2.5-2.2.0, -2.2.1, and 2.2.2" by Shima et al. (gmd-2019-294)**

The revised manuscript addresses most of my previous concerns and is almost ready to be published. I enjoyed reading it, and I only have very minor suggestions, which the authors may consider.

**Minor Comments**

P. 8, II. 25 – 27: It might also be worthwhile to state that the relaxation of very small particles to the surrounding fluid is so fast that (2) needs to be solved with a very small timestep, which is certainly not in the spirit of a computationally efficient model. See, e.g., Chen et al. (2018).

P. 9, II. 9 – 17: Why do you state the wrong  $d_i$  and  $q_i$  in Eq. (4) and give a warning in the following text? It might be clearer to state the correct  $d_i$  and  $q_i$  in (4) and then state that the wrong values are used in the presented study.

P. 12, I. 15: I suspect this is only the case for the numerical solution of (11). One can see that for  $m_i \rightarrow 0 \implies C \rightarrow 0$  and hence  $dm_i/dt \rightarrow 0$ , which prevents negative  $m_i$  for a (probably impossible) analytical solution.

P. 34, Il. 29 – 30: How do you decide if a droplet is activated or not?

P. 50, l. 11: Why is the freezing/melting timescale restricted by the CFL criterion? It is not directly apparent why a microphysical timestep is restricted by a fluid-dynamical criterion.

P. 50, ll. 22 – 23: You may want to cite Árnason and Brown (1971), who showed nicely that the model timestep for condensation/evaporation needs to be smaller than the phase relaxation timescale.

P. 62, II. 29 - 30: I agree that evaporation delays the melting process, but how does it delay the "melting onset"? I assume that before the melting onset, the considered particles consist of pure ice, and hence only sublimation might cool the particle.

**Technical Comments**

P. 2, I. 26: "composition", not "compositions"

P. 5, I. 15: Although "Appendixes" is technically correct, I suggest using the more common "Appendices".

P. 12, I. 6: I suggest adding "particle-averaged" before "ventilation coefficient".

P. 17, I. 5: For clarity, add "real" before "particles".

Fig. 1: This figure looks more like a table. Consider changing the caption.

Figs. 5, 6, 16, 17: I suggesting removing the (meaningless) empty brackets "[]" from the labels on the abscissa.

**References**

Árnason, G., & Brown Jr, P. S. (1971). Growth of cloud droplets by condensation: A problem in computational stability. *Journal of the Atmospheric Sciences*, 28(1), 72-77.

Chen, S., Yau, M. K., & Bartello, P. (2018). Turbulence effects of collision efficiency and broadening of droplet size distribution in cumulus clouds. *Journal of the Atmospheric Sciences*, 75(1), 203-217.

---

## Author Response (AR2)

**Authors' response regarding the second revision**

Dear Editor(s),

Thank you for handling our manuscript. We also would like to thank the anonymous reviewer and the topical editor Simon Unterstrasser for their thorough reading of this long manuscript. Following their incisive and constructive suggestions, we have placed the finishing touches on the manuscript.

An itemized response to all the comments raised by the referee and the topical editor Simon Unterstrasser is provided below. A marked-up manuscript showing all the changes follows.

We hope that you find our responses satisfactory and that the manuscript is acceptable for publication in Geoscientific Model Development.

Sincerely,
Shin-ichiro Shima

**Reply to the first reviewer**

We appreciate your in-depth reading of the revised manuscript and insightful feedback. Below, we provided an itemized response to all the comments raised, with the original comments presented in blue. A marked-up manuscript showing all the changes is attached at the end of this document.

**Minor Comments**

 It might also be worthwhile to state that the relaxation of very small particles to the surrounding fluid is so fast that (2) needs to be solved with a very small timestep, which is certainly not in the spirit of a computationally efficient model. See, e.g., Chen et al. (2018).

We agree that this is important information to the readers. We have revised the part as follows.

(P. 8, ll. 25--29 of the revised manuscript)
old< The relaxation time of large droplets is a few seconds (Fig. 3 of Wang and Pruppacher (1977)). The acceleration of particles can be considered by explicitly solving the motion equation (see, e.g., Naumann and Seifert, 2015).

new> For example, the relaxation time of large droplets is a few seconds (Fig. 3 of Wang and Pruppacher, 1977) though that of micrometer-sized droplets is approximately $10^{-5}\,s$ (see, e.g., Eq. (1) of Chen et al., 2018, and the discussion that follows). The acceleration of particles can be considered by explicitly solving the motion equation (see, e.g., Naumann and Seifert, 2015), but extremely small time steps would be required for small particles.

 Why do you state the wrong di and qi in Eq. (4) and give a warning in the following text? It might be clearer to state the correct di and qi in (4) and then state that the wrong values are used in the presented study.

Following your suggestion, we rephrased the part as follows.

(P. 9, ll. 9--19)
old< In this study, we consider that $d_i$ and $q_i$ are given by

$$d_i = D_i := 2\max(a_i, c_i), \quad q_i = q_i^{\mathrm{cc}} := A_i/A_i^{\mathrm{cc}},$$

where $D_i$ is the maximum dimension, $q_i^{\mathrm{cc}}$ is the area ratio regarding circumscircle, $A_i$ is the projected area perpendicular to the flow direction, and $A_i^{\mathrm{cc}}$ is the area of the circumcircle of $A_i$, i.e., the area of the smallest circle that completely contains $A_i$.

Here, the readers must be warned that our choices of $d_i$ and $q_i$ specified in Eq. (4) are incorrect. In Böhm's theory, $d_i$ is defiend by $2a_i$, and $q_i$ is defined by the area ratio regarding circumscribed ellipse $q_i^{\mathrm{ce}} := A_i/A_i^{\mathrm{ce}}$. Consequently, Eq. (4) underestimates the fall speeds of columnar ice particles. Nevertheless, based on the assessment detailed in Sec. 9.2, we confirmed that this difference does not change the results of our simulation significantly,and hence, we conclude that this flaw causes only a minor impact on this study.
* * *
new> In Böhm's theory, $d_i$ is defined by $2a_i$, and $q_i$ is defined by the area ratio regarding circumscribed ellipse $q_i^{\mathrm{ce}} := A_i/A_i^{\mathrm{ce}}$, where $A_i$ is the projected area perpendicular to the flow direction, and $A_i^{\mathrm{ce}}$ is the area of the circumscribed ellipse of $A_i$, i.e., the area of the smallest ellipse that completely contains $A_i$.

However, in this study, we start from a slightly different definition of $d_i$ and $q_i$, which we

adopted mistakenly:

$$d_i = D_i := 2\max(a_i, c_i), \quad q_i = q_i^{\mathrm{cc}} := A_i/A_i^{\mathrm{cc}},$$

where $D_i$ is the maximum dimension, $q_i^{\mathrm{cc}}$ is the area ratio regarding circumcircle, and $A_i^{\mathrm{cc}}$ is the area of the circumcircle of $A_i$, i.e., the area of the smallest circle that completely contains $A_i$.

Consequently, Eq. (4) underestimates the fall speeds of columnar ice particles. Nevertheless, based on the assessment detailed in Sec. 9.2, we will confirm that this difference does not change the results of our simulation significantly, and hence, we conclude that this flaw causes only a minor impact on this study. We also note that in Sec. 9.2 we will develop and release a fixed version of the model, SCALE-SDM 0.2.5-2.2.2.

3-3)  P. 12, l. 15: I suspect this is only the case for the numerical solution of (11). One can see that for $m_i \to 0 \Rightarrow C \to 0$ and hence $dm/dt \to 0$, which prevents negative $m_i$ for a (probably impossible) analytical solution.

You are right. $m_i$ becomes 0 in finite time, but never becomes negative. (If the ice particle is spherical and small, $dm_i/dt \propto -m_i^{1/3}$ holds when it sublimates. Then, $m_i(t) = a[(m_i(0)/a)^{2/3} - t]^{3/2}$ with some constant $a$ is the analytic solution. This is also a direct consequence of $dr_i^2/dt = \mathrm{const.}$ for spherical and small ice particles.) Therefore, we revised the part as follows.

> (P. 12, ll. 18--19)
> old< Note that $m_i$ in Eq. (11) could become negative through sublimation over a finite time. Therefore, we impose a limiter to $dm_i$ as follows:
>
> new> Note that $m_i$ in Eq. (11) can become zero through sublimation over a finite time. However, in this study, we prohibit complete sublimation, and instead, we impose a limiter to $dm_i$ as follows:

3-4)  P. 34, ll. 29 – 30: How do you decide if a droplet is activated or not?

In this manuscript, we do not distinguish cloud droplets and deliquescent aerosol particles. In Fig. R1-1 of our previous reply, droplets larger than $1\,\mu\mathrm{m}$ in radius are defined as activated droplets. A more precise definition can be considered by using the critical radius $r_{\mathrm{crt}} := (3b/a)^{1/2}$, which corresponds to the maximum of the Köhler curve. Then, we can distinguish giant CCN and activated droplets.

3-5)  P. 50, l. 11: Why is the freezing/melting timescale restricted by the CFL criterion? It is not directly
apparent why a microphysical timestep is restricted by a fluid-dynamical criterion.

Imagine that we increase the freezing/melting timestep $\Delta t_{\mathrm{fz/mlt}}$ larger than the CFL condition of wind velocity. Then, the latent heat of freezing/melting will be released/absorbed at a grid cell different from the original one, which probably impairs the accuracy of buoyancy calculation. We have not tested it before, but we therefore consider that it is reasonable to restrict $\Delta t_{\mathrm{fz/mlt}}$ by the CFL condition of wind velocity.

3-6)  P. 50, ll. 22 – 23: You may want to cite Árnason and Brown (1971), who showed nicely that the model timestep for condensation/evaporation needs to be smaller than the phase relaxation timescale.

Thank you for the information. This is a very good paper. We have added the following sentence to

the paragraph.

> (P. 51, ll. 1--2)
> Otherwise, numerical instability occurs (Árnason and Brown, 1971).

3-7)    P. 62, ll. 29 – 30: I agree that evaporation delays the melting process, but how does it delay the "melting onset"? I assume that before the melting onset, the considered particles consist of pure ice, and hence only sublimation might cool the particle.

As you pointed out, an ice particle can be colder than the ambient air due to sublimation. Figure 4 of Rasmussen and Pruppacher (1982) indicates that melting could start at +4°C at a relative humidity of 50%. See also Eq. (14) of Rasmussen and Pruppacher (1982), and accompanying discussions.

**Technical Comments**

3-8)    P. 2, l. 26: "composition", not "compositions"
3-9)    P. 5, l. 15: Although "Appendixes" is technically correct, I suggest using the more common "Appendices".
3-10)  P. 12, l. 6: I suggest adding "particle-averaged" before "ventilation coefficient".
3-11)  P. 17, l. 5: For clarity, add "real" before "particles".
(We assumed this is a comment for p. 27, not p. 17)
3-12)  Fig. 1: This figure looks more like a table. Consider changing the caption.

All the above comments are reflected in the revised manuscript.

3-13)  Figs. 5, 6, 16, 17: I suggesting removing the (meaningless) empty brackets "[]" from the labels on the abscissa.

We leave them unchanged. They are empty but have the role to inform the readers that the quantities are unitless.

**References**

Rasmussen, R. and Pruppacher, H. R.: A wind tunnel and theoretical study of the melting behavior of atmospheric ice particles. I: a wind tunnel study of frozen drops of radius less than 500 micrometers., Journal of the Atmospheric Sciences, 39, 152–158, https://doi.org/10.1175/1520-0469(1982)039<0152:AWTATS>2.0.CO;2, 1982.

**Reply to the topical editor Simon Unterstrasser**

Thank you for reading through the revised manuscript carefully. Below, we provided an itemized response to all the comments raised, with the original comments presented in blue. A marked-up manuscript showing all the changes is attached at the end of this document.

**Minor and Technical Comments**

4-1)   p.8, l.1: introduce the name of the physical quantity in front of q_d

We do not know if $q_d$ has a name commonly used. Instead, we have added the meaning of $q_d$.

(P. 8, l. 1)
mass of dry air per unit mass of moist air $q_\mathrm{d} := \rho_\mathrm{d}/\rho$,

4-2)   p.9.: circumScircle=circumcircle; defiend = defined

We have corrected the typos.

4-3)   p.9., l.18: You may explicitly state that this flaw is removed in the newer version 2.2.2.

We have added the following sentence to the end of the paragraph.

(P. 9, ll. 18--19)
We also note that in Sec. 9.2 we will develop and release a fixed version of the model, SCALE-SDM 0.2.5-2.2.2.

4-4)   p.10, l.15 and p.13, l.2: I am always picky about the expression "cold temperature".

We have replaced the two "colder" by "lower".

4-5)   p.12, l.20: Ice crystals with mass m_min^i re rather small. Don't you include a Kelvin correction, such that ice crystals wouldn't for too small supersaturation w.r.t ice?

We assumed that a pre-activated particle grows immediately when the ambient air is supersaturated over ice, but we admit that this is a crude simplification of the pre-activation phenomenon. The ice is considered to be preserved in nanoscale pores of solid particles. There is no doubt that the Kelvin effect is playing an important role, but modeling of the phenomenon is not straightforward. We leave further sophistication for future studies.

4-6)   p.18, Eqs. 55-57: I appreciate the way, the units are handled. (No changes needed.)

Thank you. I always think that we should include units to variables and constants to make equations clearer.

4-7)   p.23,l.1: Is G_lmn defined at the center of the grid cell? "at each grid point" is not specific enough in my sense.

We admit that the exact definition of the grid indices $lmn$ are not introduced in this manuscript. We consider it is not necessary because $lmn$ is used only in a symbolic sense in this manuscript, but for clarification, we revised the part as follows.

(The end of Sec. 5.1)
old< To simplify the notation, we use $\vec{G}_{lmn}$ to denote the status of moist air at each grid point.

new> To simplify the notation, we use $\vec{G}_{lmn}$ to denote the status of moist air at each point on the center grid and the face grid.

4-8)     p.23, l.7: I guess you miss to say that the sets are called I_r(t) and I_s(t).

Note that $I_{\mathrm{r}}(t) \neq \{\{\vec{x}_i(t), \vec{a}_i(t)\}, i = 1, 2, \ldots, N_{\mathrm{r}}^{\mathrm{wp}}\}$. As explained in Sec. 4.1, $\{\{\vec{x}_i(t), \vec{a}_i(t)\}, i = 1, 2, \ldots, N_{\mathrm{r}}^{\mathrm{wp}}\}$ is the set of all the particles accumulated over the whole period. On the other hand, $I_{\mathrm{r}}(t)$ is the set of particle indices existing in the domain at time $t$. (Therefore, $I_{\mathrm{r}}(t) \subset \{i = 1, 2, \ldots, N_{\mathrm{r}}^{\mathrm{wp}}\}$ holds.) Similarly, $I_{\mathrm{s}}(t)$ is the set of super-particle indices existing in the domain at time $t$, which is defined at the end of Sec. 5.2.

4-9)     p.23, Eq.83 & 83: With Dirac's delta, you evaluate n at a discrete point. There can only be one droplet at this location and I do not understand how a concentration is a reasonable quantity. Concentrations can only defined fro a continuum in my personal opinion.

Dirac's delta is not continuous, but after the ensemble average (denoted by <...>), we can expect $n(\vec{a}, \vec{x}, t)$ becomes a continuous function.

4-10)    p.25, l.5: Remove dot after "Figure"

We have corrected the typo.

4-11)    p.25, l.13: Could you reformulate "For consistency of wind velocity field divergence, .."? I am having troubles understanding it.

We have rephrased the part as follows.

(1st paragraph of Sec. 5.5.1 "Advection and sedimentation")
old< For consistency of wind velocity field divergence, we use a predictor-corrector scheme with the "simple linear interpolation" of wind velocities from the face grid following Grabowski et al. (2018). … We then interpolate $\vec{U}_{lmn}$ to the super-particle position using the simple linear scheme of Grabowski et al. (2018).

new> So that we can predict the particle number concentration accurately, we use the predictor-corrector scheme with the "simple linear interpolation" of wind velocities from the face grid following Grabowski et al. (2018). … We then interpolate $\vec{U}_{lmn}$ to the super-particle position using the simple linear scheme of Grabowski et al. (2018), which ensures that the wind velocity divergence over any subgrid volume becomes exactly the same as that over the grid cell volume.

4-12)    p.26, l.7: "is much shorter than THAT OF other processes"

Revised as suggested.

4-13)    p.26, l.18: From the way it is written, it is not clear, whether or not you assume sphericitiy and neglect the ventilation effect. Please rephrase.

We clarified the meaning as follows.

(P. 26, ll. 25--28)
old< Then, if the ice particle is spherical and if we ignore the ventilation effect, the r.h.s. of the resultant equation does not depend on $m$. Therefore, we adopt the forward Euler scheme to solve the time evolution equation of $m^{2/3}$.

new> Then, in a situation when the ice particle is spherical and, at the same time, so small that the ventilation effect can be ignored, then the equation reduces to $dm^{2/3}/dt = \text{const.}$, i.e., the r.h.s. does not depend on $m$. Inspired by this fact, we adopt the forward Euler scheme to solve the time evolution equation of $m^{2/3}$ even when the ice particle is not spherical or small.

**4-14)**  p.30, l.27: Add "Eq." in front of (95).

We rephrased the sentence as follows.

(P. 31, ll. 12--13)
old< We assume the same size distribution of internally mixed ammonium bisulfate as that of the pure ammonium bisulfate (95).

new> We assume that the size distribution of internally mixed ammonium bisulfate is the same as that of the pure ammonium bisulfate given by Eq. (95).

**4-15)**  p.39, l.32: You may better say: "The ice particle WHOSE POSITION is denoted by ...". This issue appears several times.

We leave this unchanged. The symbols are displayed not only in Fig. 2, but also in Figs. 5-8. Therefore, the symbols do not only represent their physical locations, but also the attributes of the odd ice particles. Further, we already asked a professional English editing service twice (see the acknowledgement), but they did not see any issue here.

**4-16)**  Fig.6 and several others: I would say: "The figure is the same as Fig.5, except for the (physical quantity on the) vertical axis" (remove at least "difference" in the end).

We simply removed "difference" from Figs. 5-7, 10, 11, 13, and 14.

**4-17)**  p.45, l.9: "suppress the amount of" sounds awkward. I would say, some process can be suppressed, but not a quantity.

We replaced "suppressed" by "reduced" as follows.

(P. 45, ll. 23--24)
old< When the number of super-particles was too low, more rain droplets were produced because of an erroneous enhancement of coalescence that suppressed the amount of cloud droplets, cloud ice particles, and graupel particles.

new> When the number of super-particles was too low, more rain droplets were produced because of an erroneous enhancement of coalescence that reduced the amount of cloud droplets, cloud ice particles, and graupel particles.

**4-18)**  p.63, l.7: Do you mean "heating rate"? Or is "warming rate" something else?

We followed the terminology used in Rasmussen and Pruppacher (1982).

**4-19)**  p.65, l.14: Is p defined as ratio collected over collector? Then p<1 makes sense.

p<0.1, not p<1, is the requirement of the Beard and Grover (1974)'s formula. Well, p is defined as the ratio of collected over collector, but still, it does not guarantee p<1, because we defined that the particle that falls faster is the collector (see Eq. (39) and the explanation that follows).

**4-20)**  p.65, l.29: Please reformulate. "Calculating the properties of the resultant ice crystal .."?

We think that the meaning of the original sentence is clear enough, but following your suggestion, we rephrased the sentence as follows.

(P. 66, l. 10)

[revised manuscript text omitted]

$$n = n^{\text{sulf}}(\log r^{\text{sulf}}_{\text{dry},i}, T^{\text{fz}}_i), \tag{102}$$

$$p = \frac{\delta(T^{\text{fz}} - (-38\,^\circ\text{C}))}{V_{\text{domain}} \log(r^{\text{sulf}}_{\text{dry,max}}/r^{\text{sulf}}_{\text{dry,min}})}, \tag{103}$$

and $N_{\text{s}}(0)$ in Eq. (85) is replaced by $N_{\text{s}}(0)/2$ because we use half of the super-particles for pure ammonium bisulfate aerosol

5   particles. The ammonium bisulfate mass is calculated from the dry radius $r^{\text{sulf}}_{\text{dry},i}$ as $m^{\text{sol}}_{1i} = (4\pi/3)\rho_{\text{(NH)}_4\text{HSO}_4}(r^{\text{sulf}}_{\text{dry},i})^3$, where $\rho_{\text{(NH)}_4\text{HSO}_4} = 1.78\,\text{g\,cm}^{-3}$. The soluble aerosol particle freezing temperature is $T^{\text{fz}}_i = -38\,^\circ\text{C}$.

For IN inactive mineral dust super-particles, we use $P^{\text{SP}}_{\text{INia}} = 0.05$. The mineral dust initially has the same size $d^{\text{dust}} = 1\,\mu\text{m}$. The dry radius $r^{\text{sulf}}_{\text{dry},i}$ is calculated using the same procedure as the pure ammonium bisulfate aerosol particles, i.e., for each super-particle we draw a random number uniformly in log-space from the interval $[r^{\text{sulf}}_{\text{dry,min}}, r^{\text{sulf}}_{\text{dry,max}}]$. The IN inactive mineral

10   dust freezing temperature is $T^{\text{fz}}_i = -38\,^\circ\text{C}$. From Eqs. (85) and (98), an IN inactive mineral dust super-particle's multiplicity is then given by

$$\xi_i = \frac{c^{\text{dust}}}{c^{\text{sulf}}} \frac{dN^{\text{sulf}}}{d\log r^{\text{sulf}}_{\text{dry}}} \left(\log r^{\text{sulf}}_{\text{dry},i}\right) \frac{\log(r^{\text{sulf}}_{\text{dry,max}}/r^{\text{sulf}}_{\text{dry,min}})}{c^{\text{SP}}/2} \frac{P_{\text{INia}}}{P^{\text{SP}}_{\text{INia}}}. \tag{104}$$

Finally, we consider IN active mineral dust internally mixed with ammonium bisulfate. The remaining super-particles, i.e., $(1 - P^{\text{SP}}_{\text{INia}})/2$, are used for this population. The initial diameter of the mineral dust initial is $d^{\text{dust}} = 1\,\mu\text{m}$, and the dry radius

15   $r^{\text{sulf}}_{\text{dry},i}$ is determined as in the other populations. We draw another random number uniformly from the interval $[T^{\text{fz}}_{\text{min}}, T^{\text{fz}}_{\text{max}}]$ and determine the freezing temperature $T^{\text{fz}}_i$. From Eqs. (85) and (98), an IN active mineral dust super-particle's multiplicity is then given by

$$\xi_i = \frac{c^{\text{dust}}}{c^{\text{sulf}}} \frac{dN^{\text{sulf}}}{d\log r^{\text{sulf}}_{\text{dry}}} \left(\log r^{\text{sulf}}_{\text{dry},i}\right) p(T^{\text{fz}}_i)$$
$$\frac{\log(r^{\text{sulf}}_{\text{dry,max}}/r^{\text{sulf}}_{\text{dry,min}})(T^{\text{fz}}_{\text{max}} - T^{\text{fz}}_{\text{min}})}{(c^{\text{SP}}/2)(1 - P^{\text{SP}}_{\text{INia}})}. \
[revised manuscript text omitted]